# Boosting Vision-Language Models with Transduction

**Maxime Zanella**[*]
UCLouvain, UMons

**Benoît Gérin**[*]
UCLouvain

**Ismail Ben Ayed**
ÉTS Montréal

Code: https://github.com/MaxZanella/transduction-for-vlms

## Abstract

Transduction is a powerful paradigm that leverages the structure of unlabeled data to boost predictive accuracy. We present TransCLIP, a novel and computationally efficient transductive approach designed for Vision-Language Models (VLMs). TransCLIP is applicable as a plug-and-play module on top of popular inductive zero- and few-shot models, consistently improving their performances. Our new objective function can be viewed as a regularized maximum-likelihood estimation, constrained by a KL divergence penalty that integrates the text-encoder knowledge and guides the transductive learning process. We further derive an iterative Block Majorize-Minimize (BMM) procedure for optimizing our objective, with guaranteed convergence and decoupled sample-assignment updates, yielding computationally efficient transduction for large-scale datasets. We report comprehensive evaluations, comparisons, and ablation studies that demonstrate: (i) Transduction can greatly enhance the generalization capabilities of inductive pretrained zero- and few-shot VLMs; (ii) TransCLIP substantially outperforms standard transductive few-shot learning methods relying solely on vision features, notably due to the KL-based language constraint.

## 1   Introduction

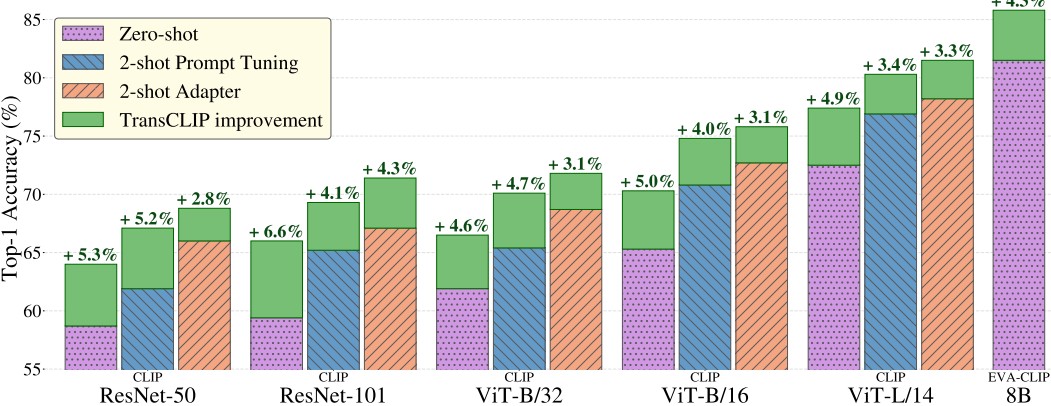

Figure 1: **TransCLIP** improves significantly the averaged top-1 accuracy on 11 datasets when used on top of inductive zero-shot **CLIP**, 2-shot **CoOp** prompt tuning and 2-shot **TaskRes** adapter for various encoder sizes.

---

[*]Equal contributions and corresponding authors. {maxime.zanella,benoit.gerin}@uclouvain.be

38th Conference on Neural Information Processing Systems (NeurIPS 2024).

Combining vision and language modalities can greatly enhance expressiveness and reduce ambiguities in the understanding and interpretation of our environment. This principle is central in the development of Vision-Language Models (VLMs), such as CLIP [50], which learns visual representations through natural-language supervision. In the pre-training phase, an input image $\mathbf{x}$ and associated text description $\mathbf{c}$ are encoded by separate vision and text encoders. This yields feature representations $\mathbf{f} = \theta_v(\mathbf{x})$ and $\mathbf{t} = \theta_t(\mathbf{c})$, which can be aligned by contrastive learning. Such a joint embedding space for the visual and textual modalities facilitates zero-shot recognition and yields powerful adaptation capabilities for a large variety of tasks. The recent literature on adapting VLMs has grown substantially, in both the zero-shot and few-shot learning settings [73, 72, 16, 70, 74, 25, 43]. However, so far, these techniques predominantly align with *induction*, i.e., inference for each test sample is performed independently from the other samples within the target dataset.

In contrast, *transduction* performs joint inference on all the test samples of a task, leveraging the statistics of the target unlabeled data [58, 27, 71]. In the context of standard vision-based classifiers, this has enabled transductive methods to outperform inductive-inference approaches as evidenced by benchmarks over large-scale datasets such as ImageNet [2].

Within the scope of deep learning, transduction has mainly been explored for few-shot learning to address the inherent challenges of training under limited supervision. This recent and quite abundant few-shot literature, e.g., [5, 13, 34, 37, 44, 76, 24, 75], among others, has focused on adopting standard vision-based pre-training models (such as ImageNet pre-training). However, as we will show in our experiments (Table 4), the direct application of existing transductive few-shot methods to VLMs yields poor performances, sometimes underperforming the inductive zero-shot predictions. This might explain why the transductive paradigm has been overlooked in zero-shot and few-shot learning for VLMs so far. The low performance of current transductive few-shot methods in the context of VLMs could be explained by the fact that the underlying objective functions do not account for the text knowledge. In this new multi-modal paradigm, additional supervision could be leveraged from the textual descriptions of the classes (prompts) [50], e.g., $\mathbf{c}_k =$ `a photo of a [kth class name]`, along with their corresponding representation $\mathbf{t}_k = \theta_t(\mathbf{c}_k)$ derived from the language encoder. We utilize the interleaved representation of text prompts and images with their cosine[2] similarity $\mathbf{f}^\top \mathbf{t}_k$, which yields text-based prediction $\hat{\mathbf{y}}_k$, thereby guiding our transductive optimization procedure with text-encoder knowledge. Our method optimizes a new objective function integrating a text-driven penalty. Optimization is carried out efficiently w.r.t the assignment variables associated with the unlabeled samples, which are then used as final predictions.

Adapting VLMs has recently attracted wide attention in the literature, predominantly focusing on inductive methods. Motivated by findings in NLP, which indicate that better prompt strategies could enhance performance [53, 26, 22], substantial efforts were directed towards prompt tuning [35] for VLMs, with CoOp [73] standing out as the pioneering work along this line. Following CoOp, prompt tuning has become the favorite strategy for adapting VLMs in a variety of contexts, including unsupervised [25, 43, 15, 41, 1] and few-shot [73, 72, 40, 12, 65, 6, 74, 8, 9, 29, 30, 67] learning. Meanwhile, there have been a few efforts towards computationally more efficient adapters [70, 48, 66]. Our transduction formulation aligns with this initiative. By operating solely on the output embeddings (i.e., in a black-box setting), TransCLIP is computationally efficient and does not make assumptions on the underlying encoder architectures. Still, our method is orthogonal to these design choices and could be applied atop any of the above-mentioned inductive approaches.

**Main contributions. (i)** We introduce a transductive formulation that enhances the zero-shot and few-shot generalization capabilities of VLMs by leveraging the structure of unlabeled data (Figure 1). Our new objective function can be viewed as a regularized maximum-likelihood estimation, constrained by a Kullback-Leibler (KL) divergence penalty integrating the text-encoder knowledge and guiding the transductive learning process. We further derive an iterative Block Majorize-Minimize (BMM) procedure for optimizing our objective, with guaranteed convergence and decoupled sample-assignment updates, yielding computationally efficient transduction for large-scale datasets, such as ImageNet. **(ii)** Our method can be used as a plug-and-play module on top of current inductive zero-shot models and few-shot learning methods, consistently boosting their performance. Also, **(iii)** our approach substantially outperforms recent transductive few-shot methods in the literature, notably due to the KL-based language supervision as a critical success factor.

---

[2]In VLMs, such as CLIP [50], both visual and text embeddings are normalized (i.e., are withing the unit hyper-sphere). Thus, the cosine similarity corresponds to the dot product.

## 2  Related Work

**Transduction for vision-only classifiers.**  The use of unlabeled test data at inference time has received attention lately in rapidly emerging subjects, such as few-shot learning and unsupervised test-time adaptation. Examples include adjusting batch normalization layer statistics [46] and minimizing the entropy of predictions [60], which can be supplemented by pseudo-labeling strategies [36]. In the few-shot literature solely based on vision models, transduction leverages both the few labeled samples and unlabeled test data, outperforming inductive methods [76, 5, 24, 37, 75].  One of the first works introducing transduction in vision-based few-shot learning proposes propagating the labels from the support (labeled) to the query (unlabeled) set with a meta-learned graph [39]. Building on this idea, another work proposes to iteratively augment the support set to improve label propagation [34]. LaplacianShot [76] also exploits the inherent structure of the data through a graph-Laplacian clustering, which discourages disparate class predictions for samples with close features, while matching each query set point to the nearest support prototype. Alternative approaches propose directly learning the class prototypes. For instance, Transductive Fine-Tuning (TF) [13] uses the prediction entropy on the query samples as a regularization term, while TIM and its variants [5, 59] employ the mutual information between the query samples and their predictions. BD-CSPN [37] refines the class prototypes by reducing the feature biases between the support set and the most confident query samples. An additional group of methods performs clustering in the feature space, for instance, by solving an optimal transport problem like PT-MAP [24], by projecting features into sub-spaces to facilitate clustering [75], or by revisiting the standard K-means with an additional partition-complexity regularizer to control the number of predicted classes [44].

**Zero- and few-shot learning in VLMs.**  Thanks to their extensive pre-training, VLMs exhibit stronger generalization capabilities than vision-only models but may also fail [50, 69, 57]. In response, substantial recent efforts have been directed towards using their general knowledge and adapting them on more specific tasks [63, 73, 70]. Arguably, the most popular strategy is prompt tuning [35], which is explored both in the unsupervised [43, 15, 41, 1] and few-shot [73, 72, 40, 12, 65, 6, 74, 8, 9, 29, 30] settings. The pioneering work, CoOp [73], updates input text-prompt tokens by leveraging the context provided by the few labeled samples (i.e., the support set). Building on this success, various strategies have been developed to enhance this approach, especially through additional regularization. For instance, ProGrad [74] guides the prompts towards the original hand-crafted ones by gradient projection. Prompt tuning has also been explored in the zero-shot setting, e.g., using the predictive confidence to generate pseudo-labels [25, 41]. Despite its popularity, prompt tuning remains tedious in terms of computations, due to the many back-propagations through the text encoder. This challenge is compounded in the recent developments, which introduce visual tokens [29, 30] alongside the text tokens. In contrast, there has been limited efforts so far in developing black-box methods [48, 17, 16, 66, 62], which only access the final embedding states. These methods often rely on the so-called adapters [23], like Tip-Adapter(-F) [70], which adds a classifier at the output of the vision encoder, in the form of a cache model involving the few-shot samples. Lately, a strong baseline based on Gaussian discriminant analysis clustering [62] demonstrates VLMs' adaptation abilities with a Gaussian hypothesis on the embedding space.

**Transductive inference in VLMs.**  Despite the growing interest in unsupervised, zero-shot and few-shot learning for VLMs, the transductive-inference paradigm has not been explored so far in this new multi-modal context, except for the very recent work in [45], which was deployed for small-size tasks ($\approx 10^2$ test samples). However, the method in [45] may not be computationally tractable for large-scale query sets, due to expensive inner loops for estimating the Dirichlet distribution's parameters. We provide a computationally efficient solution, which can scale up to large target datasets (such as ImageNet), while being easily amenable as a plug-and-play module on top of state-of-the-art inductive methods. It is worth mentioning that test-time adaptation methods also employ the transduction paradigm, but their settings are very different from those studied in this work. For instance, SwapPrompt [41] has been designed to make batch predictions on-the-fly, and has continual-learning mechanisms such as an exponential moving average prompt across batches. TPT [43] work on a single sample with many data augmentations to train one prompt per image. Both methods require access to model weights for training (i.e., do not operate in a black-box setting) and an expensive training procedure. We also note that prompt tuning does not scale well with the model size and is even impractical on very large models such as EVA-CLIP-8B [55]. We still report the performances of this class of methods in the Appendix (Table 9).

# 3 TransCLIP: Transduction for Vision-Language Models

In this section, we describe our objective function for transductive inference in vision-language models, and derive a block majorize-minimize (BMM) algorithm for minimizing it, with guaranteed convergence and decoupled sample-assignment updates. When dealing with a zero-shot classification problem based on a vision-language model, such as CLIP, and given a set of $K$ candidate classes, one creates textual descriptions, the so-called prompts [38], each corresponding to a class, e.g., $\mathbf{c}_k$ = a photo of a [kth class name], $k = 1, \ldots, K$. Let $\mathbf{t}_k = \theta_t(\mathbf{c}_k)$ denotes the corresponding normalized (unit hyper-sphere) embedding representation, with $\theta_t$ representing the language encoder. Similarly, each test image $\mathbf{x}_i$, $i = 1, \ldots, N$, is projected onto a normalized embedding space of the same dimension, using visual encoder $\theta_v$: $\mathbf{f}_i = \theta_v(\mathbf{x}_i)$. In the standard inductive zero-shot inference, classification of a given image $\mathbf{x}_i$ is done by evaluating the cosine similarity between these two encoded modalities and predicting the class corresponding to the most similar text embedding: $\hat{k} = \mathrm{argmax}_k \ \mathbf{f}_i^\top \mathbf{t}_k$. Furthermore, one can compute pseudo-labels corresponding to these zero-shot predictions by applying the softmax function with a temperature scaling[3] $\tau$, which yields the following probability-simplex vector for each sample:

$$\hat{\mathbf{y}}_i = (\hat{y}_{i,k})_{1 \leq k \leq K} \in \Delta_K; \quad \hat{y}_{i,k} = \frac{\exp(\tau \mathbf{f}_i^\top \mathbf{t}_k)}{\sum_j \exp(\tau \mathbf{f}_i^\top \mathbf{t}_j)} \tag{1}$$

where $\Delta_K$ denotes the probability simplex. Let $\mathcal{D} = \{i \in \mathbb{N} : 1 \leq i \leq N\} = \mathcal{S} \cup \mathcal{Q}$ denotes the samples indices of the target dataset, with $\mathcal{Q}$ the set of unlabeled *query* samples indices, i.e., those for which we want to make a prediction, and $\mathcal{S}$ the set of labeled *support* samples indices in the few-shot setting.

Note that, in the zero-shot setting, $\mathcal{S} = \emptyset$. We define a Gaussian Mixture Model-clustering (GMM) term in our objective function by modeling the likelihood of these target data as a balanced mixture of multivariate Gaussian distributions, each representing a class $k$ and parameterized by mean vector $\boldsymbol{\mu}_k$ and a diagonal covariance matrix $\boldsymbol{\Sigma}$:

$$p_{i,k} = \mathrm{Pr}(\mathbf{f}_i, k; \boldsymbol{\mu}_k, \boldsymbol{\Sigma}) \propto \det(\boldsymbol{\Sigma})^{-\frac{1}{2}} \exp\left(-\frac{1}{2}(\mathbf{f}_i - \boldsymbol{\mu}_k)^\top \boldsymbol{\Sigma}^{-1}(\mathbf{f}_i - \boldsymbol{\mu}_k)\right)$$

Notation $p_{i,k}$ is introduced here to simplify the equations in the sequel. Notice that, unlike standard GMMs, we deploy a common diagonal covariance matrix $\boldsymbol{\Sigma}$ across all classes. Interestingly, in our experiments, we found this simplifying choice improves the performance while reducing the computational load as there are substantially fewer parameters to learn. This is particularly the case when dealing with large numbers of classes as in large-scale target datasets such as ImageNet.

## 3.1 Proposed objective function

Our objective function depends on two types of variables: (i) Sample-to-class assignment variables within the probability simplex: $\mathbf{z}_i = (z_{i,k})_{1 \leq k \leq K} \in \Delta_K$, $i \in \mathcal{Q}$; and (ii) GMM parameters $\boldsymbol{\mu} = (\boldsymbol{\mu}_k)_{1 \leq k \leq K}$ and $\boldsymbol{\Sigma}$.

We propose to minimize the following objective, which integrates a GMM-clustering term, a Laplacian regularizer and a Kullback-Leibler (KL) divergence penalty encoding the text-encoder knowledge and guiding the transductive learning process:

$$\mathcal{L}_{\text{ZERO-SHOT}}(\mathbf{z}, \boldsymbol{\mu}, \boldsymbol{\Sigma}) = \underbrace{-\sum_{i \in \mathcal{Q}} \mathbf{z}_i^\top \log(\mathbf{p}_i)}_{\text{GMM clustering}} - \underbrace{\sum_{i \in \mathcal{D}} \sum_{j \in \mathcal{D}} w_{ij} \mathbf{z}_i^\top \mathbf{z}_j}_{\text{Laplacian reg.}} + \underbrace{\sum_{i \in \mathcal{Q}} \mathrm{KL}_\lambda(\mathbf{z}_i || \hat{\mathbf{y}}_i)}_{\text{Text knowledge}} \tag{2}$$

where $\mathbf{p}_i = (p_{i,k})_{1 \leq k \leq K} \in \Delta_K$ concatenates the GMM probabilities, $w_{ij}$ denotes some measure of affinity between visual embeddings $\mathbf{f}_i$ and $\mathbf{f}_j$, and the sample-wise parameterized[4] KL terms are given by:

$$\mathrm{KL}_\lambda(\mathbf{z}_i || \hat{\mathbf{y}}_i) = \mathbf{z}_i^\top \log \mathbf{z}_i - \lambda \mathbf{z}_i^\top \log \hat{\mathbf{y}}_i, \quad i \in \mathcal{Q}; \quad \lambda > 0 \tag{3}$$

In the following, we describe the effect of each term in our objective function in (2):

---

[3]Note that each CLIP version comes with a temperature scaling factor $\tau$, which is optimized along with the learnable parameters during pre-training.

[4]Notice that, for $\lambda = 1$, the expression in (3) corresponds to the KL divergence.

- **GMM-based clustering**: This unsupervised-learning term is akin to the GMM-based maximum-likelihood estimation objective in the standard EM algorithm [3]. By taking the negative logarithm, its minimization corresponds to maximizing the likelihood of the data. It can also be viewed as a probabilistic generalization of the K-means clustering objective [28]. Indeed, assuming $\boldsymbol{\Sigma}$ is the identity matrix reduces the first term in (2) to the K-means objective.

- **Laplacian regularization**: The second term in (2) is the Laplacian regularizer, widely used in the context of graph/spectral clustering [56] and semi-supervised learning [7]. This term encourages nearby samples in the visual-embedding space (i.e., pairs of samples with high affinity $w_{i,j}$) to have similar $\mathbf{z}$ assignments. In our case, we propose to build a positive semi-definite (PSD) affinity matrix based on the cosine similarities as $w_{ij} = \mathbf{f}_i^\top \mathbf{f}_j$ (Gram matrix). As we see below, this PSD condition is important to obtain a convergent Majorize-Minimize optimizer with decoupled (parallel) sample-wise updates for the $\mathbf{z}$-assignments, yielding a highly efficient transduction for large-scale target datasets (such as ImageNet).

- **Text-guided KL divergence:** This term is dedicated to vision-language models and, as we will see in our experiments (ablation studies in Tables 4 and 6), has a substantial effect on performance. It encourages the prediction not to deviate significantly from the zero-shot predictions, thereby providing text supervision to the other two unsupervised-learning terms. Furthermore, being convex over $\mathbf{z}_i$, $i \in \mathcal{Q}$, this term facilitates the optimization of the objective w.r.t the assignment variables.

## 3.2 Extension to the few-shot setting

Our zero-shot formulation naturally extends to the few-shot setting. We integrate supervision from the labeled-support samples, in the form of a cross-entropy, which corresponds to minimizing the following overall loss:

$$\mathcal{L}_{\text{FEW-SHOT}}(\mathbf{z}, \boldsymbol{\mu}, \boldsymbol{\Sigma}) = -\frac{\gamma}{|\mathcal{S}|} \sum_{i \in \mathcal{S}} \mathbf{z}_i^\top \log(\mathbf{p}_i) + \frac{1}{|\mathcal{Q}|} \mathcal{L}_{\text{ZERO-SHOT}}(\mathbf{z}, \boldsymbol{\mu}, \boldsymbol{\Sigma}) \qquad (4)$$

Note that, in the first term, the $\mathbf{z}_i$ are fixed, with $\mathbf{z}_i = \mathbf{y}_i$, $i \in \mathcal{S}$ and $\mathbf{y}_i$ the one-hot ground-truth label associated with the corresponding shot.

## 3.3 Block Majorize-Minimize (BMM) optimization

As our objective depends on three types of variables $(\mathbf{z}, \boldsymbol{\mu}, \boldsymbol{\Sigma})$, we proceed with a BMM procedure, alternating three sub-step optimizers. Each sub-step optimizes over one block of variables while the other two are fixed, ensuring the overall objective does not increase. Importantly, the obtained $\mathbf{z}$-updates (Eq. (5)) are decoupled, yielding computationally efficient transduction for large-scale datasets. Also, our overall procedure is guaranteed to converge (Theorem 1).

**Majorize-Minimize (MM) with respect to the z-block**   When $\boldsymbol{\mu}$ and $\boldsymbol{\Sigma}$ are fixed, both the GMM- and KL-based terms are convex w.r.t $\mathbf{z}_i$. However, the Laplacian term is concave[5] (for PSD matrix $\boldsymbol{W}$). Therefore, we proceed with inner iterations, each minimizing a linear and tight upper bound, the so-called majorizing function in the MM-optimization literature [33, 21, 31], which guarantees the overall objective does not increase. To obtain the tight linear bound, let us write the Laplacian term conveniently in the following matrix form: $\mathbf{z}^\top \boldsymbol{\Psi} \mathbf{z}$, with $\boldsymbol{\Psi} = -\boldsymbol{W} \otimes \boldsymbol{I}$, where $\otimes$ denotes the Kronecker product and $\boldsymbol{I}$ is the $N \times N$ identity matrix. Note that $\boldsymbol{\Psi}$ is negative semi-definite for a positive semi-definite $\boldsymbol{W}$. Therefore, $\mathbf{z}^\top \boldsymbol{\Psi} \mathbf{z}$ is a concave function with respect to $\mathbf{z}$, and its first-order approximation at current solution $\mathbf{z}^l$ ($l$ being the iteration index) gives the following tight[6] upper bound on the Laplacian term:

$$\mathbf{z}^\top \boldsymbol{\Psi} \mathbf{z} \leq (\mathbf{z}^l)^\top \boldsymbol{\Psi} \mathbf{z}^l + (\boldsymbol{\Psi} \mathbf{z}^l)^\top (\mathbf{z} - \mathbf{z}^l)$$

Replacing the quadratic Laplacian term by this linear bound yields a majorizing function on our overall objective. Importantly, this majorizing function is a sum of decoupled objectives, each

---

[5]This makes the overall sub-problem non-convex and there is no closed-form solution.

[6]"Tight" means that the upper bound is equal to the original objective at the current solution $\mathbf{z}^l$.

corresponding to one assignment variable $\mathbf{z}_i$, yielding a highly efficient optimizer for large-scale target datasets. Indeed, using simplex constraints $\mathbf{z}_i \in \Delta_K, i \in \mathcal{Q}$, and solving the Karush-Kuhn-Tucker (KKT) conditions independently for each $\mathbf{z}_i$, we obtain the following decoupled update rules for the $\mathbf{z}$-block:

$$\mathbf{z}_i^{(l+1)} = \frac{\hat{\mathbf{y}}_i^\lambda \odot \exp(\log(\mathbf{p}_i) + \sum_{j \in \mathcal{D}} w_{ij} \mathbf{z}_j^{(l)})}{(\hat{\mathbf{y}}_i^\lambda \odot \exp(\log(\mathbf{p}_i) + \sum_{j \in \mathcal{D}} w_{ij} \mathbf{z}_j^{(l)}))^\top \mathbb{1}_K} \tag{5}$$

**Closed-form updates of $\boldsymbol{\mu}$ and $\boldsymbol{\Sigma}$**    When both $\mathbf{z}$ and $\boldsymbol{\Sigma}$ are fixed, our objective in (4) is convex. It can be minimized by setting its gradient w.r.t each $\boldsymbol{\mu}_k$ to zero, which yields the following closed-form updates:

$$\boldsymbol{\mu}_k = \frac{\frac{\gamma}{|\mathcal{S}|} \sum_{i \in \mathcal{S}} z_{i,k} \mathbf{f}_i + \frac{1}{|\mathcal{Q}|} \sum_{i \in \mathcal{Q}} z_{i,k} \mathbf{f}_i}{\frac{\gamma}{|\mathcal{S}|} \sum_{i \in \mathcal{S}} z_{i,k} + \frac{1}{|\mathcal{Q}|} \sum_{i \in \mathcal{Q}} z_{i,k}} \tag{6}$$

Similarly, when both $\mathbf{z}$ and $\boldsymbol{\mu}$ are fixed, the following closed-form updates minimize the overall objective w.r.t $\boldsymbol{\Sigma}$:

$$\text{diag}(\boldsymbol{\Sigma}) = \frac{\frac{\gamma}{|\mathcal{S}|} \sum_{i \in \mathcal{S}} \sum_k z_{i,k} (\mathbf{f}_i - \boldsymbol{\mu}_k)^2 + \frac{1}{|\mathcal{Q}|} \sum_{i \in \mathcal{Q}} \sum_k z_{i,k} (\mathbf{f}_i - \boldsymbol{\mu}_k)^2}{\gamma + 1} \tag{7}$$

The complete procedure is summarized in Appendix B. Note that, after convergence, we use the sample-to-class assignment variables $\mathbf{z}_i$ as predictions for each sample $i$ of the query set $\mathcal{Q}$ using the argmax operation for conventional classification.

## 3.4    Convergence

Our optimizer can be viewed as an instance of the general Block Majorize-Minimize paradigm for optimization [51], which optimizes a majorizing function for each block of variables. The convergence of general BMM procedures is well studied in the optimization community [51]. Indeed, under certain conditions (such as the strong convexity of the block-wise majorizing functions), we can establish convergence of our procedure using the following result (more details in Appendix A):

**Theorem 1 (Convergence of BMM [51])** *Assume that, for each block, the majorizing function is quasi-convex, and its first-order behavior is the same as the original objective locally. Furthermore, assume that the sub-problem solved for each block has a unique solution. Then, every limit point of the iterates generated by BMM is a coordinate-wise minimum of the overall objective.*

## 4    Experiments

**Datasets.**    Following the setting of previous works [73, 43], we assess TransCLIP on ImageNet [11] and ten datasets for fine-grained classification of scenes (SUN397 [64]), aircraft types (Aircraft [42]), satellite imagery (EuroSAT [18]), automobiles (Cars [32]), food items (Food [4]), pet breeds (Pets [49]), flowers (Flowers [47]), general objects (Caltech101 [14]), textures (DTD [10]) and human actions (UCF101 [54]). We additionally measure performance on four variants of ImageNet (Adversarial [20], ImageNetV2 [52], Rendition [19], Sketch [61]). Numerical results are reported in terms of the top-1 accuracy with the ViT-B/16 encoder, averaged over three random seeds.

**Benchmarks.**    We aim to show the breadth of potential applications of transduction in the context of VLMs. Notably, employing supervised fine-tuning, followed by transduction with TransCLIP on the unlabeled test samples, emerges as a powerful and efficient solution. This is particularly convenient when the labeled samples (the support set) and/or computational power are not accessible at inference (i.e., test) time[7]. To this end, we first study the applicability of our zero-shot formulation TransCLIP-ZS (Eq. (2)) across three settings: (i) on top of inductive *zero-shot* learning and popular *few-shot* learning methods; (ii) on top of 16-shot ImageNet pretraining for *cross-dataset* transferability, and

---

[7]This application is hardly discussed in the transductive literature. We make all zero-shot- and few-shot text and image embeddings publicly available, to ease future works without resorting to heavy computations.

Table 1: TransCLIP atop inductive vision-language zero-shot and popular few-shot methods.

| | Method | ImageNet | SUN397 | Aircraft | EuroSAT | StanfordCars | Food101 | Pets | Flower102 | Caltech101 | DTD | UCF101 | Average |
|---|---|---|---|---|---|---|---|---|---|---|---|---|---|
| **0-shot** | **CLIP-ViT-B/16** | 66.6 | 62.5 | 24.7 | 48.3 | 65.6 | 85.9 | 89.1 | 70.7 | 93.2 | 43.5 | 67.5 | 65.3 |
| | + TransCLIP-ZS | $70.3_{+3.7}$ | $68.9_{+6.3}$ | $26.9_{+2.2}$ | $65.1_{+16.8}$ | $69.4_{+3.8}$ | $87.1_{+1.2}$ | $92.6_{+3.5}$ | $76.7_{+5.9}$ | $92.7_{-0.5}$ | $49.5_{+6.0}$ | $74.4_{+6.9}$ | $70.3_{+5.1}$ |
| **1-shot** | CoOp (IJCV '22) | 65.7 | 66.9 | 20.7 | 56.4 | 67.6 | 84.3 | 90.2 | 78.2 | 92.5 | 50.1 | 71.2 | 67.6 |
| | + TransCLIP-ZS | $69.3_{+3.6}$ | $71.5_{+4.6}$ | $23.8_{+3.1}$ | $65.3_{+8.9}$ | $71.9_{+4.3}$ | $86.3_{+2.0}$ | $91.9_{+1.8}$ | $89.8_{+11.5}$ | $93.8_{+1.3}$ | $55.4_{+5.4}$ | $77.7_{+6.5}$ | $72.4_{+4.8}$ |
| | TIP-Adapter-F (ECCV '22) | 69.5 | 67.2 | 28.8 | 67.8 | 67.1 | 85.8 | 90.6 | 83.7 | 94.0 | 51.6 | 73.4 | 70.9 |
| | + TransCLIP-ZS | $72.0_{+2.5}$ | $71.8_{+4.6}$ | $30.7_{+1.9}$ | $76.9_{+9.1}$ | $71.0_{+3.9}$ | $86.9_{+1.1}$ | $93.1_{+2.4}$ | $92.8_{+9.1}$ | $93.5_{-0.5}$ | $57.7_{+6.1}$ | $80.0_{+6.7}$ | $75.1_{+4.3}$ |
| | PLOT (ICLR '23) | 66.9 | 67.0 | 28.9 | 72.8 | 68.5 | 84.9 | 91.9 | 81.8 | 94.0 | 52.8 | 74.7 | 71.3 |
| | + TransCLIP-ZS | $75.8_{+8.9}$ | $70.3_{+3.3}$ | $28.1_{-0.8}$ | $78.8_{+6.0}$ | $70.0_{+1.6}$ | $85.3_{+0.4}$ | $91.1_{-0.8}$ | $93.2_{+11.4}$ | $94.0_{-0.0}$ | $56.7_{+3.9}$ | $81.4_{+6.7}$ | $75.0_{+3.7}$ |
| | TaskRes (CVPR '23) | 69.6 | 68.1 | 31.2 | 65.6 | 69.1 | 84.5 | 90.2 | 81.6 | 93.6 | 53.4 | 71.8 | 70.8 |
| | + TransCLIP-ZS | $72.0_{+2.5}$ | $72.5_{+4.4}$ | $31.4_{+0.2}$ | $73.7_{+8.1}$ | $71.6_{+2.4}$ | $86.5_{+2.0}$ | $91.6_{+1.5}$ | $90.7_{+9.1}$ | $94.0_{+0.4}$ | $59.4_{+6.0}$ | $76.4_{+4.6}$ | $74.5_{+3.7}$ |
| | ProGrad (ICCV '23) | 67.0 | 67.0 | 28.7 | 57.0 | 68.2 | 84.9 | 91.4 | 80.8 | 93.5 | 52.8 | 73.3 | 69.5 |
| | + TransCLIP-ZS | $70.1_{+3.1}$ | $71.6_{+4.6}$ | $30.5_{+1.8}$ | $70.9_{+13.9}$ | $72.3_{+4.1}$ | $86.5_{+1.6}$ | $92.7_{+1.4}$ | $91.5_{+10.7}$ | $94.1_{+0.7}$ | $57.9_{+5.1}$ | $79.3_{+6.1}$ | $74.3_{+4.8}$ |
| **4-shot** | CoOp (IJCV '22) | 68.8 | 69.7 | 30.8 | 69.6 | 74.4 | 84.5 | 92.5 | 74.4 | 92.2 | 59.4 | 77.5 | 74.0 |
| | + TransCLIP-ZS | $71.4_{+2.6}$ | $73.3_{+3.5}$ | $33.1_{+2.3}$ | $77.2_{+7.5}$ | $77.7_{+3.2}$ | $86.5_{+1.9}$ | $93.6_{+1.1}$ | $95.3_{+3.1}$ | $95.1_{+0.6}$ | $63.0_{+3.6}$ | $81.8_{+4.3}$ | $77.1_{+3.1}$ |
| | TIP-Adapter-F (ECCV '22) | 70.7 | 70.8 | 35.7 | 76.8 | 74.1 | 86.5 | 91.9 | 92.1 | 94.0 | 59.8 | 78.1 | 75.6 |
| | + TransCLIP-ZS | $72.7_{+1.9}$ | $74.4_{+3.5}$ | $36.1_{+0.5}$ | $79.7_{+2.9}$ | $75.9_{+1.8}$ | $87.4_{+0.9}$ | $93.2_{+1.3}$ | $95.5_{+3.3}$ | $95.1_{+0.4}$ | $64.0_{+4.2}$ | $83.3_{+5.2}$ | $77.9_{+2.3}$ |
| | PLOT (ICLR '23) | 70.0 | 71.8 | 34.8 | 84.7 | 76.6 | 83.5 | 92.8 | 93.2 | 94.9 | 61.0 | 79.7 | 76.6 |
| | + TransCLIP-ZS | $77.2_{+7.2}$ | $73.5_{+1.7}$ | $33.9_{-0.9}$ | $81.8_{+2.9}$ | $75.8_{-0.8}$ | $85.6_{+2.2}$ | $92.5_{-0.3}$ | $95.8_{+2.6}$ | $94.8_{-0.1}$ | $63.6_{+2.6}$ | $83.3_{+3.6}$ | $78.0_{+1.4}$ |
| | TaskRes (CVPR '23) | 71.0 | 72.8 | 33.3 | 73.8 | 76.1 | 86.1 | 91.9 | 85.0 | 94.9 | 59.7 | 75.5 | 74.6 |
| | + TransCLIP-ZS | $73.0_{+2.0}$ | $75.3_{+2.5}$ | $34.4_{+1.1}$ | $78.1_{+4.4}$ | $77.2_{+1.1}$ | $87.3_{+1.2}$ | $93.0_{+1.1}$ | $92.4_{+7.4}$ | $95.1_{+0.2}$ | $64.3_{+4.6}$ | $79.2_{+3.7}$ | $77.2_{+2.7}$ |
| | ProGrad (ICCV '23) | 70.2 | 71.7 | 34.0 | 69.6 | 75.0 | 85.4 | 92.0 | 91.1 | 94.4 | 59.8 | 77.9 | 74.6 |
| | + TransCLIP-ZS | $72.3_{+2.1}$ | $75.0_{+3.3}$ | $35.5_{+1.6}$ | $74.9_{+5.3}$ | $77.9_{+2.9}$ | $87.0_{+1.5}$ | $93.7_{+1.7}$ | $95.3_{+4.2}$ | $95.1_{+0.8}$ | $64.8_{+5.1}$ | $83.2_{+5.4}$ | $77.7_{+3.1}$ |
| **16-shot** | CoOp (IJCV '22) | 71.9 | 74.9 | 43.3 | 85.0 | 82.8 | 84.2 | 91.9 | 96.8 | 95.8 | 69.7 | 83.1 | 79.9 |
| | + TransCLIP-ZS | $73.3_{+1.4}$ | $76.6_{+1.8}$ | $42.9_{-0.4}$ | $86.0_{+1.0}$ | $83.0_{+0.2}$ | $86.3_{+2.1}$ | $93.2_{+1.2}$ | $97.5_{+0.8}$ | $95.9_{+0.1}$ | $71.3_{+1.7}$ | $85.4_{+2.3}$ | $81.1_{+1.1}$ |
| | TIP-Adapter-F (ECCV '22) | 73.3 | 76.0 | 44.6 | 85.9 | 82.3 | 86.8 | 92.6 | 96.2 | 95.7 | 70.8 | 83.9 | 80.7 |
| | + TransCLIP-ZS | $74.2_{+0.9}$ | $76.8_{+0.8}$ | $44.9_{+0.3}$ | $85.2_{-0.7}$ | $82.7_{+0.4}$ | $87.4_{+0.6}$ | $93.5_{+0.9}$ | $96.9_{+0.7}$ | $95.7_{+0.1}$ | $69.2_{-1.5}$ | $85.6_{+1.7}$ | $81.1_{+0.4}$ |
| | PLOT (ICLR '23) | 72.5 | 76.0 | 46.8 | 92.1 | 84.6 | 85.6 | 92.5 | 97.1 | 96.0 | 71.1 | 84.8 | 81.7 |
| | + TransCLIP-ZS | $77.8_{+5.3}$ | $75.0_{-1.0}$ | $41.8_{-4.9}$ | $84.6_{-7.5}$ | $79.6_{-4.9}$ | $85.9_{+0.2}$ | $92.2_{-0.4}$ | $97.3_{+0.1}$ | $95.0_{-1.0}$ | $68.7_{-2.4}$ | $85.7_{+0.9}$ | $80.3_{-1.4}$ |
| | TaskRes (CVPR '23) | 73.0 | 76.0 | 44.8 | 80.7 | 82.8 | 85.8 | 92.5 | 97.3 | 95.9 | 70.9 | 83.4 | 80.5 |
| | + TransCLIP-ZS | $74.1_{+1.0}$ | $76.9_{+0.8}$ | $43.6_{-1.2}$ | $80.5_{-0.3}$ | $82.8_{-0.7}$ | $87.5_{+0.6}$ | $92.9_{+0.4}$ | $97.6_{+0.3}$ | $96.0_{+0.1}$ | $70.2_{-0.7}$ | $86.2_{+2.8}$ | $80.8_{+0.3}$ |
| | ProGrad (ICCV '23) | 72.1 | 75.1 | 42.8 | 83.6 | 82.9 | 85.8 | 92.9 | 96.6 | 95.9 | 68.9 | 82.6 | 79.9 |
| | + TransCLIP-ZS | $73.5_{+1.4}$ | $76.8_{+1.7}$ | $42.8_{-0.0}$ | $83.7_{+0.2}$ | $83.1_{+0.2}$ | $87.2_{+1.3}$ | $93.7_{+0.8}$ | $97.4_{+0.8}$ | $96.0_{+0.1}$ | $71.4_{+2.5}$ | $86.1_{+3.4}$ | $81.1_{+1.1}$ |

Table 2: Cross-Dataset transferability evaluation. Few-shot learning methods are trained on 16-shot ImageNet and evaluate on the ten other fine-grained datasets. Average excludes ImageNet.

| | | Source | Target | | | | | | | | | | |
|---|---|---|---|---|---|---|---|---|---|---|---|---|---|
| | Method | ImageNet | SUN397 | Aircraft | EuroSAT | StanfordCars | Food101 | Pets | Flower102 | Caltech101 | DTD | UCF101 | Average |
| **Cross-Dataset** | CoOp (IJCV '22) | 71.9 | 62.0 | 15.7 | 44.6 | 62.1 | 84.3 | 88.3 | 67.1 | 92.7 | 39.5 | 64.1 | 62.0 |
| | + TransCLIP-ZS | $73.3_{+1.4}$ | $67.4_{+5.4}$ | $17.1_{+1.4}$ | $54.5_{+9.9}$ | $66.8_{+4.8}$ | $86.3_{+2.0}$ | $89.4_{+1.1}$ | $74.2_{+7.2}$ | $93.4_{+0.7}$ | $42.1_{+2.6}$ | $69.9_{+5.7}$ | $66.1_{+4.1}$ |
| | CoCoOp (CVPR '22) | 71.1 | 67.0 | 22.7 | 44.6 | 64.9 | 86.2 | 90.7 | 71.6 | 93.9 | 45.2 | 68.8 | 65.6 |
| | + TransCLIP-ZS | $76.8_{+5.7}$ | $69.6_{+2.7}$ | $22.6_{-0.1}$ | $59.2_{+14.6}$ | $67.0_{+2.1}$ | $85.4_{-0.8}$ | $89.8_{-0.9}$ | $79.0_{+7.4}$ | $94.3_{+0.3}$ | $50.6_{+5.4}$ | $74.5_{+5.7}$ | $69.2_{+3.6}$ |
| | MaPLE (CVPR '23) | 70.5 | 67.3 | 24.4 | 45.8 | 65.7 | 86.4 | 90.4 | 72.0 | 93.7 | 46.3 | 68.7 | 66.1 |
| | + TransCLIP-ZS | $76.6_{+6.1}$ | $69.8_{+2.5}$ | $24.5_{+0.2}$ | $59.5_{+13.7}$ | $66.8_{+1.2}$ | $85.4_{-1.0}$ | $89.7_{-0.7}$ | $78.0_{+6.0}$ | $94.3_{+0.6}$ | $49.4_{+3.1}$ | $74.4_{+5.6}$ | $69.2_{+3.1}$ |
| | ProGrad (ICCV '23) | 72.1 | 63.9 | 21.6 | 38.9 | 64.0 | 85.9 | 90.2 | 67.8 | 92.9 | 43.2 | 65.9 | 63.4 |
| | + TransCLIP-ZS | $73.5_{+1.4}$ | $68.6_{+4.7}$ | $22.7_{+1.1}$ | $55.2_{+16.4}$ | $67.9_{+3.8}$ | $87.0_{+1.2}$ | $91.3_{+1.1}$ | $73.9_{+6.1}$ | $94.0_{+1.1}$ | $46.6_{+3.4}$ | $73.5_{+7.6}$ | $68.1_{+4.6}$ |
| | PromptSRC (ICCV '23) | 71.4 | 67.3 | 24.1 | 45.0 | 65.6 | 86.5 | 90.1 | 70.5 | 93.8 | 46.2 | 68.9 | 65.8 |
| | + TransCLIP-ZS | $76.9_{+5.5}$ | $69.9_{+2.6}$ | $24.9_{+0.8}$ | $59.4_{+14.4}$ | $67.6_{+2.0}$ | $85.3_{-1.2}$ | $89.4_{-0.7}$ | $76.7_{+6.2}$ | $94.2_{+0.4}$ | $51.1_{+5.0}$ | $76.0_{+7.0}$ | $69.4_{+3.7}$ |

(iii) on top of 16-shot ImageNet pretraining for *domain generalization* on the four ImageNet variants. Secondly, we compare our few-shot extension TransCLIP-FS (Eq. (4)) to transductive few-shot learning methods. As for TransCLIP-ZS, we operate in a black-box setting (i.e., using only the output embeddings, without training the model parameters).

**Implementation details.**  The main component of our transductive formulation is the text-guided KL divergence penalty. We fix $\lambda = 1$ for all our zero-shot experiments (see ablation study in Table 6), and $\lambda = 0.5$ in all the few-shot experiments to reduce the impact of the text-driven regularization. Another component of our optimization problem is the Laplacian regularization, which enforces consistent predictions for close instances. We truncate the affinity matrix to the 3 nearest-neighbors, making it sparse. $\mu$ is initialized with the top-8 most confident samples of each class for the zero-shot setting. For the few-shot setting, we use the class-wise average over the shot embeddings.

## 4.1 Main results

**Transduction improvements.**  Table 1 and 2 demonstrate the advantages of our transductive approach in zero-shot, few-shot, and cross-dataset transferability. TransCLIP enhances the zero-shot top-1 accuracy by over 5% and popular few-shot methods by 4% (1-shot) on average, without the need for additional labels. Table 3 further highlights that TransCLIP can be applied on top of prompt tuning and adapter fine-tuning solutions, enhancing performance for both in-domain and domain

Table 3: Domain Generalization evaluation with improved manual prompting strategy (custom templates are given in Table 24b), 16-shot prompt-tuning and 16-shot adapter.

| | Method | Source | Target | | | | Average | Average OOD |
|---|---|---|---|---|---|---|---|---|
| | | ImageNet | Adversarial | ImageNetV2 | Rendition | Sketch | | |
| 0-shot | **CLIP-ViT-B/16** *w/* `a photo of a` | 66.6 | 47.9 | 60.6 | 73.8 | 46.0 | 59.0 | 57.1 |
| | + TransCLIP-ZS | $70.3_{+3.7}$ | $49.5_{+1.7}$ | $62.3_{+1.7}$ | $75.0_{+1.3}$ | $49.7_{+3.7}$ | $61.4_{+2.4}$ | $59.2_{+2.1}$ |
| | **CLIP-ViT-B/16** *w/ custom templates* | 68.8 | 50.6 | 62.3 | 77.8 | 48.4 | 61.6 | 59.8 |
| | + TransCLIP-ZS | $71.5_{+2.7}$ | $52.1_{+1.4}$ | $63.4_{+1.1}$ | $78.1_{+0.2}$ | $51.1_{+2.7}$ | $63.2_{+1.6}$ | $61.1_{+1.3}$ |
| Domain G. | **CLIP-ViT-B/16** *w/ prompt tuning (CoOp)* | 71.9 | 49.4 | 64.1 | 75.1 | 47.2 | 61.5 | 59.0 |
| | + TransCLIP-ZS | $73.3_{+1.4}$ | $50.8_{+1.4}$ | $64.6_{+0.5}$ | $75.8_{+0.7}$ | $50.3_{+3.1}$ | $63.0_{+1.5}$ | $60.4_{+1.4}$ |
| | **CLIP-ViT-B/16** *w/ adapter (TaskRes)* | 73.0 | 50.3 | 65.6 | 77.8 | 49.2 | 63.2 | 60.7 |
| | + TransCLIP-ZS | $74.1_{+1.1}$ | $51.9_{+1.6}$ | $65.4_{-0.2}$ | $78.4_{+0.6}$ | $51.6_{+2.4}$ | $64.3_{+1.1}$ | $61.8_{+1.1}$ |

Table 4: Transductive few-shot learning evaluation. *w/o text* denotes $\lambda = 0$ in Eq. (3).

| Shots | Method | ImageNet | SUN397 | Aircraft | EuroSAT | StanfordCars | Food101 | Pets | Flowers102 | Caltech101 | DTD | UCF101 | Average |
|---|---|---|---|---|---|---|---|---|---|---|---|---|---|
| **0** | **CLIP-ViT-B/16** | 66.6 | 62.5 | 24.7 | 48.3 | 65.6 | 85.9 | 89.1 | 70.7 | 93.2 | 43.5 | 67.5 | 65.3 |
| **1** | TF [13] | 29.7 | 38.1 | 19.2 | 46.0 | 32.5 | 43.5 | 38.2 | 67.8 | 75.5 | 31.6 | 48.8 | 42.8 |
| | BD-CSPN [37] | 35.4 | 45.7 | 22.0 | 45.7 | 42.0 | 54.2 | 52.9 | 82.9 | 83.5 | 34.7 | 58.0 | 50.6 |
| | LaplacianShot [76] | 34.9 | 44.5 | 22.1 | 52.1 | 41.1 | 53.0 | 52.2 | 83.1 | 83.4 | 35.8 | 57.3 | 50.9 |
| | PT-MAP [24] | 40.1 | 52.6 | 23.8 | 59.7 | 48.4 | 64.4 | 61.8 | 69.4 | 54.1 | 41.8 | 63.5 | 52.7 |
| | TIM [5] | 37.5 | 48.3 | 22.8 | 48.2 | 44.8 | 65.7 | 53.9 | **86.4** | 75.1 | 35.8 | 62.7 | 52.8 |
| | TransCLIP-FS *w/o text* | 30.2 | 43.4 | 23.7 | 56.6 | 41.0 | 50.9 | 54.3 | 83.5 | 77.7 | 36.9 | 54.5 | 50.2 |
| | TransCLIP-FS | 69.8 | 70.6 | 29.9 | 72.5 | 70.9 | 87.9 | 93.8 | 84.8 | 93.1 | 53.3 | 78.4 | 73.2 |
| **4** | TF [13] | 51.1 | 61.0 | 30.3 | 64.9 | 56.8 | 71.0 | 65.9 | 90.9 | 91.5 | 53.7 | 67.9 | 64.1 |
| | BD-CSPN [37] | 53.8 | 62.5 | 30.5 | 64.8 | 58.5 | 75.3 | 72.0 | 92.5 | 92.0 | 52.1 | 70.9 | 65.9 |
| | LaplacianShot [76] | 53.5 | 62.5 | 29.6 | 74.3 | 58.5 | 75.7 | 73.4 | 92.8 | 92.0 | 52.7 | 71.7 | 67.0 |
| | PT-MAP [24] | 57.6 | 68.1 | 31.2 | 74.9 | 63.1 | 81.1 | 79.5 | 76.2 | 60.2 | 58.4 | 73.9 | 65.8 |
| | TIM [5] | 57.4 | 67.0 | 32.8 | 79.3 | 65.8 | 83.5 | 82.3 | 93.4 | 88.5 | 58.1 | 76.5 | 71.3 |
| | TransCLIP-FS *w/o text* | 53.9 | 63.8 | **34.2** | **79.4** | 63.5 | 76.7 | 76.7 | 93.3 | 92.8 | 57.0 | 74.8 | 69.6 |
| | TransCLIP-FS | 70.3 | 71.9 | 34.0 | 79.4 | 74.0 | 86.4 | 91.6 | 93.6 | 94.0 | 61.1 | 79.1 | 75.9 |
| **16** | TF [13] | 61.8 | 70.1 | 38.3 | 74.3 | 71.2 | 80.7 | 79.5 | **95.4** | 93.6 | 62.9 | 76.0 | 73.1 |
| | BD-CSPN [37] | 61.7 | 69.4 | 37.7 | 73.4 | 70.7 | 80.2 | 81.2 | 94.8 | 93.3 | 61.3 | 76.0 | 72.7 |
| | LaplacianShot [76] | 60.9 | 68.3 | 36.1 | 78.1 | 69.2 | 81.2 | 81.7 | 94.8 | 93.1 | 58.6 | 76.3 | 72.6 |
| | PT-MAP [24] | 64.0 | 72.0 | 37.4 | 75.6 | 72.0 | 82.7 | 86.1 | 78.5 | 63.7 | 63.7 | 76.3 | 70.2 |
| | TIM [5] | 67.8 | 73.6 | 40.6 | **83.6** | 79.5 | 84.9 | 88.7 | **95.4** | 92.4 | **67.5** | **82.1** | 77.8 |
| | TransCLIP-FS *w/o text* | 65.9 | 72.6 | **41.9** | 81.1 | 77.0 | 83.2 | 86.1 | 95.2 | **94.6** | 65.3 | 80.0 | 76.6 |
| | TransCLIP-FS | **71.8** | **74.7** | 38.6 | 83.0 | **79.8** | **86.9** | **92.4** | 94.4 | 94.0 | 65.1 | **82.1** | **78.4** |

generalization tasks. However, we observe in Table 1 that transductive gains sometimes decrease with the number of shots, presumably because data structure information can be partially captured in the shots. These results underline the value of considering the structure of the unlabeled test samples during prediction, especially on top of zero- and low-shot models or when facing domain shifts, an aspect not leveraged by the current zero- and few-shot VLM literature. More detailed results for five different backbone architectures and comparisons with unsupervised non-transductive methods are provided in Appendix C.1 for the zero-shot setting, in Appendix C.2 for TransCLIP on top of popular few-shot methods, in Appendix C.3 for cross-dataset transferability and in Appendix C.4 for domain generalization. *With its hyper-parameters unchanged*, TransCLIP exhibits strong generalization from convolutional networks to transformer-based models, as also depicted in Figure 1.

**Transductive few-shot learning.** We compare TransCLIP-FS, TransCLIP-FS without text regularization (i.e., $\lambda = 0$) and state-of-the-art transductive few-shot methods. It is important to note that these few-shot methods were primarily developed for vision-centric tasks. Hence, they rely on visual information, omitting the textual elements. This allows us to study the impact of our text-based regularization term. Table 4 shows that incorporating language in the transductive paradigm boosts the performance over vision-only methods. Especially for the 1- to 4-shot settings, our language-driven KL penalty enhances the performance by a large margin on many tasks (e.g., ImageNet, SUN397, StanfordCars, DTD). As the number of shots increases, the text-driven penalty becomes less useful, especially for the datasets capitalizing on the visual shots rather than the text-encoder knowledge (e.g., EuroSat and Flowers). This points to promising future directions involving more flexible text regularization (e.g., an adaptable $\lambda$ taking into account the number of shots and the quality of the text embeddings). Detailed results for five different encoder architectures are provided in Appendix C.5, consistently showing similar conclusions.

Table 5: Performance and runtime comparison between TransCLIP and prompt learning solutions on average over ImageNet and the 10 fine-grained classification datasets. UPL* is a transductive adaptation of the original unsupervised procedure in [25], more details in Appendices C.1 and C.5.

(a) Zero-shot setting.

|  | Performance | Runtime |
|---|---|---|
| UPL* | 69.8 | >150 min |
| TransCLIP-ZS | **70.3** | **14.4 sec** |

(b) Few-shot setting (4-shot).

|  | Performance | Runtime |
|---|---|---|
| CoOp+UPL* | 74.4 | >12h |
| TransCLIP-FS | **75.9** | **35.3 sec** |

Table 6: Analysis on the components and sensitivity to hyper-parameters of TransCLIP-ZS.

(a) Components of the procedure.

| Update $\mu$ | Update $\Sigma$ | Lapl. $w$ | ImageNet | SUN397 | Aircraft | EuroSAT |
|---|---|---|---|---|---|---|
| ✗ | ✓ | ✓ | 69.7 | 67.5 | 25.5 | 63.9 |
| ✓ | ✗ | ✓ | 68.7 | 66.0 | 25.1 | 51.6 |
| ✓ | ✓ | ✗ | 69.9 | 68.8 | **27.0** | 64.5 |
| ✗ | ✗ | ✓ | 68.6 | 65.9 | 25.2 | 61.8 |
| ✓ | ✓ | ✓ | **70.3** | **68.9** | 26.9 | **65.1** |

(b) Text regularization hyper-parameter $\lambda$.

| $\lambda$ | ImageNet | SUN397 | Aircraft | EuroSAT |
|---|---|---|---|---|
| 0.1 | 56.3 | 58.6 | 26.0 | 65.5 |
| 0.5 | 69.8 | **69.3** | 26.6 | **65.6** |
| 1 | **70.3** | 68.9 | **26.9** | 65.1 |
| 2 | 69.5 | 67.6 | 26.2 | 64.1 |
| 5 | 68.2 | 65.2 | 25.2 | 51.2 |

(c) Number of nearest-neighbors.

| # neighbors | ImageNet | SUN397 | Aircraft | EuroSAT |
|---|---|---|---|---|
| 3 | 70.3 | 68.9 | 26.9 | 65.1 |
| 5 | 70.3 | 68.9 | 26.8 | 65.1 |
| 10 | 70.2 | 68.8 | 26.9 | 65.2 |

(d) Impact of an isotropic $\Sigma$.

|  | ImageNet | SUN397 | Aircraft | EuroSAT |
|---|---|---|---|---|
| $\Sigma$ (ours) | 70.3 | 68.9 | 26.9 | 65.1 |
| $\Sigma$ isotropic | 69.4 | 68.0 | 26.4 | 64.1 |
| $\Delta$ | -0.9 | -0.9 | -0.5 | -1.0 |

**Comparison with prompt learning.** Following current VLMs literature, adapting the input prompt instead of GMM parameters could be seen as a more straightforward solution. For a fair comparison, we adapt Unsupervised Prompt Learning (UPL) [25] for the transductive setting and reevaluate its main hyper-parameter (see Appendix C.1). Table 5 shows clearly that TransCLIP outperforms UPL while being two to three orders of magnitude faster. Additional details on runtime are provided in Table 8 of the Appendix.

## 4.2 Ablation studies

**Components of TransCLIP.** We study the impact of the principal components involved in the TransCLIP procedure over four diverse datasets. Table 6a shows that updating $\mu$ and $\Sigma$ allows to significantly boost TransCLIP's performance. This indicates the importance of having a dynamic parametric model instead of a fixed one. Table 6b demonstrates the critical role of text-driven penalty for TransCLIP in the zero-shot setting. Additional results on the sensitivity of $\lambda$ in the few-shot setting are depicted in Figure 2 of the Appendix. Alongside the prior findings from Table 4, it is evident that incorporating text information is key to the success of TransCLIP and its wide applicability across the zero- and few-shot learning scenarios. The number of nearest-neighbors considered in the Laplacian term (Eq. (2)) does not make a significant difference in TransCLIP's performance as suggested by Table 6c. However, removing the Laplacian regularization (Table 6a) leads to inferior results on some datasets such as ImageNet and EuroSAT. We choose to consider 3 nearest-neighbors to make the affinity matrix $W$ sparse and reduce memory consumption. We also investigate the diagonal covariance matrix design by restricting it to be isotropic (i.e., $\Sigma = \sigma^2 I_d$ with $I_d$ the identity matrix). Table 6d shows that a non-isotropic $\Sigma$ performs better without significantly increasing the amount of trainable parameters.

**Scaling to larger VLMs.** We report TransCLIP-ZS performance on EVA-CLIP 8 billion parameter version [55] (approximately 42 times larger than the CLIP-ViT-B/16). It is worth mentioning that TransCLIP is easily applicable to multi-billion parameter models since it does not necessitate gradient computation or model parameter training (i.e., it only requires the memory needed for single-sample inference because the whole dataset processing can be performed one sample at a time). Table 7 shows that transduction can also bring significant improvements to larger models (details in Appendix C.1).

Table 7: Performance of TransCLIP-ZS for increasingly large VLMs. Relative $\Delta$ is the improvement normalized by the zero-shot error: $(\text{ACC}_{\text{TRANSCLIP}} - \text{ACC}_{\text{ZERO-SHOT}}) / (100 - \text{ACC}_{\text{ZERO-SHOT}})$.

| | | ImageNet | | | Average (11 datasets) | | |
|---|---|---|---|---|---|---|---|
| | #Params | Zero-shot | w/ TransCLIP-ZS | relative $\Delta$ | Zero-shot | w/ TransCLIP-ZS | relative $\Delta$ |
| CLIP-ViT-B/16 | 177M | 66.6 | $70.3_{+3.7}$ | **+11** % | 65.3 | $70.3_{+5.0}$ | **+14** % |
| CLIP-ViT-L/14 | 427M | 72.9 | $77.2_{+4.3}$ | **+16** % | 72.5 | $77.4_{+4.9}$ | **+18** % |
| EVA-CLIP-8B | 7.5B | 82.5 | $84.6_{+2.1}$ | **+12** % | 81.5 | $85.8_{+4.3}$ | **+23** % |

# 5  Conclusion

In this work, we studied the transductive paradigm in the context of Vision-Language Models and proposed the TransCLIP method. Our algorithm is highly efficient, as it operates solely in the output embedding space (i.e., black-box setting), making it suitable for a wide range of models, including very large ones. This also enables TransCLIP to be compatible with models that are accessible only through APIs. We first showed how TransCLIP can bring transduction to the inductive zero-shot setting, achieving consistent gains without additional supervision. Then, we proposed a new setting that applies transduction on top of popular few-shot methods, offering a convenient strategy to combine computationally intensive supervised fine-tuning with efficient test-time transduction. Finally, we highlighted the limitations of current transductive few-shot methods and proposed a simple extension of TransCLIP to incorporate labeled samples. In all our experiments, TransCLIP's text-guided KL divergence term appears as a key factor in its success. Future work may focus on further enhancing this regularization term, for example, by making it more resilient (e.g., with adaptive class-wise weighting) when text prompts are less reliable.

# 6  Acknowledgments

M. Zanella and B. Gérin are funded by the Walloon region under grant No. 2010235 (ARIAC by DIGITALWALLONIA4.AI). The present research benefited from computational resources made available on Lucia, infrastructure funded by the Walloon Region under grant No. 1910247.

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

# A  More details on convergence

As mentioned in the main paper, our derived block-wise optimization procedure in Eqs. (5), (6) and (7) can be viewed as an instance of the the general Block Majorize-Minimize paradigm for non-convex optimization, also referred to as the Block Successive Minimization (BSUM) method [51]. We update each block of variables, with the other blocks fixed, by minimizing a tight upper bound (majorizing function), thereby guaranteeing the overall objective does not increase at each step. In the steps with respect to $\boldsymbol{\mu}$ and $\boldsymbol{\Sigma}$, we optimize directly the objective in closed-form, which could be also viewed as a particular case of optimizing a tight upper bound. The convergence of the general BSUM procedure is well studied in the optimization community [51]. Indeed, under the following assumptions for each block of variables, one can establish convergence results for the application of BSUM to non-convex problems [51]:

- A1: The majorizing function is a tight upper bound, i.e., equal to the objective at the current solution.

- A2: The first-order behavior of the majorizing function is the same as the original objective locally.

Indeed, when assumptions A1 and A2 are verified for each block, we have the result in Theorem 1 [51].

As for our case of alternating Eqs. (5), (6) and (7), it is straightforward to verify that Assumptions A1 and A2 are satisfied for each block of variables. Furthermore, the majorizing functions are convex and thus quasi-convex. Also, the sub-problem solved for each block has a unique solution. In particular, for the $\mathbf{z}$-updates, the majorizing function is the sum of a linear and a strongly convex function (the negative entropy). Therefore, it is strongly convex. As for the $\boldsymbol{\mu}$- and $\boldsymbol{\Sigma}$-updates, the solutions are obtained in closed form (hence unique).

---

**Algorithm 1** TransCLIP

---

**Require:** A set of image embeddings $(\mathbf{f}_i)_{1 \leq i \leq N}$, a set of textual class embeddings $(\mathbf{t}_k)_{1 \leq k \leq K}$, $\tau$ the temperature of the CLIP model.

1: $w_{i,j} \leftarrow \mathbf{f}_i^\top \mathbf{f}_j \quad \forall i, j$ ▷ Affinity measure, truncated with top-3 values

2: $\hat{\mathbf{y}}_i \leftarrow \varphi(\tau \mathbf{f}_i^\top \mathbf{t})) \quad \forall i$ ▷ Initial predictions, $\varphi$ the softmax function

3: $\boldsymbol{\mu}_k \leftarrow \text{mean}\{\mathbf{f}_i \text{ s.t } y = k, i \in \mathcal{S}\}^8 \quad \forall k$ ▷ Class centroids initialization

4: $\text{diag}(\boldsymbol{\Sigma}) \leftarrow \mathbf{1}\frac{1}{d}$ ▷ Covariance matrix initialization, $d$ is the emb. dim.

5: $\mathbf{z}_i \leftarrow \hat{\mathbf{y}}_i \quad \forall i$ ▷ Initial assignments

6: **while** *(1)*, *(2)* and *(3)* not converged **do** ▷ Block-wise updates loop

7:     **while** *(1)* not converged **do** ▷ z-update loop

8:       $z_{i,k} \leftarrow \dfrac{\hat{y}_{i,k}^\lambda \exp(\log(p_{i,k}) + \sum_{j \in \mathcal{D}} w_{ij} z_{j,k})}{\sum_{k'} \hat{y}_{i,k'}^\lambda \exp(\log(p_{i,k'}) + \sum_{j \in \mathcal{D}} w_{ij} z_{j,k'})} \quad \forall i \, \forall k$ ▷ *(1)* z-step

9:     **end while**

10:     $\boldsymbol{\mu}_k \leftarrow \dfrac{\frac{\gamma}{|\mathcal{S}|} \sum_{i \in \mathcal{S}} z_{i,k} \mathbf{f}_i + \frac{1}{|\mathcal{Q}|} \sum_{i \in \mathcal{Q}} z_{i,k} \mathbf{f}_i}{\frac{\gamma}{|\mathcal{S}|} \sum_{i \in \mathcal{S}} z_{i,k} + \frac{1}{|\mathcal{Q}|} \sum_{i \in \mathcal{Q}} z_{i,k}} \quad \forall k$ ▷ *(2)* $\boldsymbol{\mu}$-step

11:     $\text{diag}(\boldsymbol{\Sigma}) \leftarrow \dfrac{\frac{\gamma}{|\mathcal{S}|} \sum_{i \in \mathcal{S}} \sum_k z_{i,k} (\mathbf{f}_i - \boldsymbol{\mu}_k)^2 + \frac{1}{|\mathcal{Q}|} \sum_{i \in \mathcal{Q}} \sum_k z_{i,k} (\mathbf{f}_i - \boldsymbol{\mu}_k)^2}{\gamma + 1}$ ▷ *(3)* $\boldsymbol{\Sigma}$-step

12: **end while**

13: **return** $\text{argmax}_k(\mathbf{z})$ ▷ Prediction with assignment variables

---

# B  Further details on TransCLIP implementation

This section aims to provide an additional pseudo-algorithm to supplement Section 3 as well as more details on TransCLIP hyper-parameters presented in Section 4. Our code is available at `https://github.com/MaxZanella/transduction-for-vlms` and a pseudo-code in Algorithm 1 summarizes the main steps of the TransCLIP algorithm.

---

[8] For the zero-shot setting, we use the embedding of top-8 most confident initial predictions for each class as explained in Section 4.

**Hardware.** All our experiments were conducted on a single A100-40 GB. In terms of memory, TransCLIP consumes 16.9 GB when inferring on ImageNet, and can therefore process large datasets on a smaller 24 GB GPU.

**Hyper-parameters.** In practice, TransCLIP performs 10 iterations of $\mathbf{z}, \boldsymbol{\mu}, \boldsymbol{\Sigma}$ block-wise updates. For each $\mathbf{z}$-update, we perform 5 iterations, as we found it sufficient for convergence. In the zero-shot setting, we set $\lambda = 1$ and $\gamma = 0$ (as there are no support samples). In the few-shot setting, we set $\lambda = 0.5$ and search for the value of $\gamma$ in $\{0.002, 0.01, 0.02, 0.2\}$. The number of validation shots is set at $\min(4, \#\text{shots})$, and we build a 1-nearest neighbor classifier with the query samples and their final class assignment to predict the class of each validation sample.

**Prompt templates.** We employ the prompt templates detailed in Table 24a for all our experiments in zero-shot setting unless otherwise explicitly specified. Only when specified, we utilize the custom template ensembling for ImageNet as in [70] (see Table 24b).

## C   Additional results.

We provide detailed results for all the studied vision backbones of CLIP over the 11 datasets to support the transferability of TransCLIP across both convolutional networks and transformer-based models. We additionally report other methods that do not fit into the transductive setting.

### C.1   Zero-shot

In Table 9. We report performances of 5 CLIP encoders as well as the 8 billion parameter EVA-CLIP [55]. We compare TransCLIP-ZS to unsupervised methods namely TPT [43], MTA [68], SwapPrompt [41], and UPL [25]. Note that TPT and MTA are two test-time augmentation methods working on a single image at a time, thus they differ from our transductive setting, still we report their performance for informational purposes.

**UPL**$^*$**.** As mentioned in Section 4, we slightly modify UPL to apply it to the test set in a transductive manner (transductive UPL is denoted UPL$^*$). Indeed, UPL relies on the generation of $N = 16$ hard pseudo-labels per class from a training set, after what a cross-entropy loss function on soft tokens is minimized. Instead, UPL$^*$ generates the pseudo-labels directly from the test set. For fairness, we reevaluated the number of pseudo-labels to select and still found that 16 per class yields the best results on average, as seen in Table 23.

### C.2   TransCLIP-ZS on top of few-shot methods

In Tables 10, 11, 12, 13 and 14. We report the performance of TransCLIP-ZS on top of CoOp [73], Tip-Adapter-F [70], PLOT [8], TaskRes [66] and ProGrad [74] for five encoders. The results are consistent with the main findings of Section 4 and indicate their generalization for several encoder architectures.

### C.3   Cross-Dataset transferability

In Table 15. We report the performance of TransCLIP-ZS on top of CoOp [73], CoCoOp [72], ProGrad [74], PromptSRC [30] and MaPLE[29]. We additionally report PromptAlign [1], which is working on a single image at a time and thus differs from our transductive setting. Note that PromptSRC and MaPLE introduce learnable vision tokens, and are therefore not compatible with convolutional-based encoders. The results are similar to those of Section 4.

### C.4   Domain Generalization

In Tables 16 and 17. We extend the results from Table 3 to five encoders. These results support those of Section 4 and show that TransCLIP can improve both zero- and few-shot model generalization for various encoders.

## C.5 Transductive few-shot learning

In Tables 18, 19, 20, 21 and 22, we implemented transductive methods from the traditional few-shot literature that align the most with our work in terms of computational efficiency and wide applicability: TIM [5], LaplacianShot [76], BD-CSPN [37], TF [13], and PT-MAP [24]. Additionally, due to the lack of transductive methods in Vision-Language and to ensure more comprehensive comparisons, we introduce a hybrid method named CoOp+UPL. This method combines prompt learning with both labeled shots and selected pseudo-labels following the methodology of UPL [25]. More details on each method and their validation procedure are outlined below. Methods with tunable hyper-parameters are fine-tuned using the validation split provided with each dataset. In line with other work [48], validation is performed for each dataset and for every shot number, setting the number of validation shots at $\min(4, \#\text{shots})$. Hyper-parameters are then optimized through a grid search to maximize accuracy on the validation set. Note that we only search for $\gamma$ across 4 values for TransCLIP. More details on the grid search for each method is given below. Detailed results for the five architectures studied in this paper are available in Table 18, 19, 20, 21, 22. Now we describe the implementation details for each reported transductive few-shot methods.

**Transductive Fine-Tuning.** We follow the original implementation of Transductive Fine-Tuning [13]. The authors kept the hyper-parameters fixed for all datasets since the goal was to propose a simple baseline, with a temperature set to 1 and the number of training steps to 25. However, they pointed out possible improvements if the hyper-parameters were tuned for each dataset. Therefore, we search for the optimal temperature value by validation in $\{0.25, 0.5, 1, 2, 4\}$ and the number of iterations in $\{10, 15, 20, 25, 30, 35, 40\}$.

**BD-CSPN.** We follow the original implementation of BD-CSPN [37]. Regarding the hyper-parameters, this method generates $Z$ pseudo-labels per class from the query set to augment the support set and to build the $K$ prototype vectors. They also introduce a temperature scaling parameter $\varepsilon$ for the computation of the prototype vectors. The authors set $Z$ to 8 and the temperature scaling $\varepsilon$ to 10. We search for the value of $Z$ in $\{0, 1, 2, 3, 4, 5, 6, 7, 8, 9, 10\}$ and $\varepsilon$ in $\{2.5, 5, 10, 20, 40\}$ by validation.

**LaplacianShot.** We follow the original implementation of LaplacianShot [76]. They balanced the Laplacian regularization term with a factor $\lambda$ and used $k$-nearest neighbors consistency. We follow the proposed ranges to find the hyper-parameter values by validation, with $\lambda$ in $\{0.1, 0.3, 0.5, 0.7, 0.8, 1, 1.2, 1.5\}$ and the number of neighbors to consider $k$ in $\{3, 5, 10\}$.

**PT-MAP.** We follow the original implementation of PT-MAP [24]. In their work, the authors show a small performance sensitivity to the learning rate $\alpha$ used to update the class prototypes through iterative adaptation. Following their discussion, we search $\alpha$ in $\{0.2, 0.4\}$.

**TIM.** We follow the original implementation of TIM [5]. The authors proposed two solvers to find the solution to the minimization problem: gradient-descent TIM (TIM-GD) and alternating-direction method (TIM-ADM). We decide to focus on the second approach since there are fewer hyper-parameters to tune. They set the weighting factors of the cross-entropy, the marginal entropy, and the conditional entropy terms to 0.1, 1 and 0.1, respectively. They also introduced a temperature parameter $\tau$ in their classifier and set it to 15. We search for the values of the cross-entropy and the conditional entropy factors in $\{0.05, 0.1, 0.4, 0.7, 1\}$ and the temperature in $\{5, 10, 15, 30, 60\}$ by validation.

**CoOp+UPL.** We implement a natural extension of CoOp to include the pseudo-labels proposed by UPL. As in UPL, $N = 16$ hard pseudo-labels per class are generated according to the prediction's confidence. Pseudo-labels from the query set $\mathcal{P} \subseteq \mathcal{Q}$ and labeled shots from $\mathcal{S}$ are unified into a single learning set $\mathcal{S} \cup \mathcal{P}$. To separate the contribution of the pseudo-labels from the labeled shots,

we split the cross-entropy loss function into two terms:

$$\mathcal{L}_{\mathcal{S}\cup\mathcal{P}}(\overline{\mathbf{V}}|\{\mathbf{x}_i\}_{i=1}^{|\mathcal{S}\cup\mathcal{P}|}) = \beta \frac{1}{|\mathcal{S}|} \sum_{j\in\mathcal{S}} \mathcal{L}_{\text{CoOp}}(\overline{\mathbf{V}}|\mathbf{x}_j) \tag{8}$$

$$+ (1-\beta) \frac{1}{|\mathcal{P}|} \sum_{j\in\mathcal{P}} \mathcal{L}_{\text{UPL}}(\overline{\mathbf{V}}|\mathbf{x}_j), \quad \beta \in [0,1]$$

Where $\overline{\mathbf{V}}$ denotes the vector of learnable context token embeddings. Despite increased computational needs, we search for the value of $\beta$ in $\{0.1, 0.3, 0.5, 0.7, 0.9\}$ by validation for the sake of fairness. The number of epochs, the learning rate and its schedule, the optimizer and the context tokens initialization follow exactly the CoOp implementation.

## D  Limitations

As discussed in Section 4, the gain of TransCLIP-ZS on top of few-shot methods tends to decrease when the number of shots is high (e.g., 16 shots) and future works may investigate this aspect.

Secondly, as TransCLIP's performance relies greatly on its text-regularization term, TransCLIP is subject to some biases. One notable bias pertains to the quality of text embeddings within each class. Recent literature has highlighted that these embeddings exhibit a preference for more frequently occurring concepts [57]. However, this issue may be mitigated through our proposed few-shot extension (e.g., introducing labels for more challenging classes).

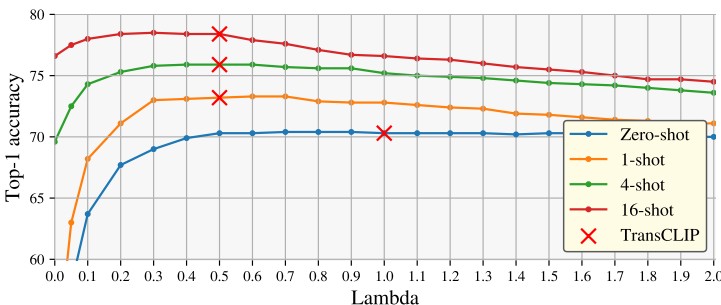

Figure 2: Sensitivity analysis of $\lambda$. Lower values reduce penalty towards zero-shot prediction and are more appropriate for higher number of shots. Top-1 accuracy averaged over 11 datasets is reported.

Table 8: Runtime and performance comparison between TransCLIP-ZS and zero-shot prompt learning. UPL$^*$ is a transductive adaptation of the original unsupervised procedure in [25]. "Prediction" refers to similarity measurement for CLIP and UPL$^*$, and to the iterative procedure for TransCLIP-ZS.

| Dataset | #samples | | Training | Images + Texts encoding | Prediction | Total | Top-1 accuracy |
|---|---|---|---|---|---|---|---|
| ImageNet | 50,000 | CLIP | / | 58.7 sec | $\sim$0 sec | 58.7 sec | 66.6 |
| | | UPL$^*$ | 151 min | 58.7 sec | $\sim$0 sec | 152 min | 69.6 |
| | | TransCLIP-ZS | / | 58.7 sec | 14.4 sec | 73.1 sec | 70.3 |
| SUN397 | 19,850 | CLIP | / | 49.2 sec | $\sim$0 sec | 49.2 sec | 62.5 |
| | | UPL$^*$ | 39 min | 49.2 sec | $\sim$0 sec | 40 min | 67.4 |
| | | TransCLIP-ZS | / | 49.2 sec | 2.6 sec | 51.8 sec | 68.9 |
| StanfordCars | 8,041 | CLIP | / | 20.5 sec | $\sim$0 sec | 20.5 sec | 65.6 |
| | | UPL$^*$ | 20 min | 20.5 sec | $\sim$0 sec | 20 min | 71.1 |
| | | TransCLIP-ZS | / | 20.5 sec | 0.7 sec | 21.2 sec | 69.4 |
| Flowers | 2,463 | CLIP | / | 4.8 sec | $\sim$0 sec | 4.8 sec | 70.7 |
| | | UPL$^*$ | 9 min | 4.8 sec | $\sim$0 sec | 9 min | 73.5 |
| | | TransCLIP-ZS | / | 4.8 sec | 0.2 sec | 5.0 sec | 76.7 |

Table 9: Adaptation of CLIP on 11 classification datasets with zero-shot methods.

| Method | ImageNet | SUN397 | Aircraft | EuroSAT | StanfordCars | Food101 | Pets | Flower102 | Caltech101 | DTD | UCF101 | Average |
|---|---|---|---|---|---|---|---|---|---|---|---|---|
| **CLIP-ResNet-50** | 58.0 | 58.8 | 17.0 | 36.2 | 55.7 | 77.4 | 85.8 | 66.0 | 85.7 | 42.9 | 61.9 | 58.7 |
| + TransCLIP-ZS | $60.8_{+2.8}$ | $64.2_{+5.4}$ | $16.6_{-0.4}$ | $59.6_{+23.4}$ | $57.9_{+2.2}$ | $78.0_{+0.6}$ | $89.3_{+3.6}$ | $72.2_{+6.2}$ | $88.6_{+3.0}$ | $47.8_{+5.0}$ | $68.8_{+6.9}$ | $64.0_{+5.3}$ |
| TPT *w/ a photo of a* | 60.7 | 61.5 | 17.6 | 28.3 | 58.5 | 74.9 | 84.5 | 62.7 | 87.0 | 40.8 | 60.8 | 57.9 |
| UPL* | 61.6 | 63.3 | 16.7 | 52.1 | 63.1 | 78.0 | 89.1 | 69.3 | 85.7 | 47.0 | 65.8 | 62.9 |
| SwapPrompt | 61.8 | 63.9 | 18.0 | 46.6 | 59.6 | 75.1 | 89.1 | 70.22 | 89.9 | 47.3 | 65.7 | 62.5 |
| **CLIP-ResNet-101** | 60.6 | 59.0 | 17.9 | 32.7 | 63.2 | 80.7 | 87.0 | 64.4 | 89.9 | 37.2 | 61.1 | 59.4 |
| + TransCLIP-ZS | $64.8_{+4.2}$ | $65.1_{+6.0}$ | $19.2_{+1.3}$ | $59.3_{+26.6}$ | $68.6_{+5.4}$ | $81.9_{+1.2}$ | $89.8_{+2.7}$ | $72.6_{+8.2}$ | $93.0_{+3.1}$ | $42.9_{+5.7}$ | $68.9_{+7.8}$ | $66.0_{+6.6}$ |
| UPL* | 63.7 | 63.5 | 18.1 | 61.3 | 69.5 | 80.9 | 90.0 | 67.3 | 88.3 | 42.8 | 67.3 | 64.8 |
| **CLIP-ViT-B/32** | 61.9 | 62.1 | 19.1 | 45.2 | 60.2 | 80.4 | 87.4 | 66.5 | 91.5 | 42.7 | 63.6 | 61.9 |
| + TransCLIP-ZS | $64.9_{+3.0}$ | $67.6_{+5.5}$ | $20.3_{+1.3}$ | $59.0_{+13.8}$ | $63.3_{+3.2}$ | $81.5_{+1.1}$ | $89.0_{+1.7}$ | $74.4_{+7.9}$ | $91.8_{+0.3}$ | $50.4_{+7.7}$ | $68.7_{+5.1}$ | $66.5_{+4.6}$ |
| UPL* | 64.6 | 66.4 | 19.1 | 59.3 | 64.8 | 81.0 | 89.8 | 69.7 | 89.8 | 48.3 | 67.8 | 65.5 |
| **CLIP-ViT-B/16** | 66.6 | 62.5 | 24.7 | 48.3 | 65.6 | 85.9 | 89.1 | 70.7 | 93.2 | 43.5 | 67.5 | 65.3 |
| + TransCLIP-ZS | $70.3_{+3.7}$ | $68.9_{+6.3}$ | $26.9_{+2.2}$ | $65.1_{+16.8}$ | $69.4_{+3.8}$ | $87.1_{+1.2}$ | $92.6_{+3.5}$ | $76.7_{+5.9}$ | $92.7_{-0.5}$ | $49.5_{+6.0}$ | $74.4_{+6.9}$ | $70.3_{+5.1}$ |
| TPT *w/ a photo of a* | 69.0 | 65.5 | 24.8 | 42.4 | 66.9 | 84.7 | 87.8 | 69.0 | 94.2 | 47.8 | 68.0 | 65.5 |
| MTA *w/ a photo of a* | 69.3 | 65.0 | 25.3 | 38.7 | 68.1 | 85.0 | 88.2 | 68.3 | 94.1 | 45.6 | 68.1 | 65.1 |
| UPL* | 69.6 | 67.4 | 24.7 | 69.5 | 71.1 | 85.8 | 92.4 | 73.5 | 91.9 | 47.7 | 73.7 | 69.8 |
| **CLIP-ViT-L/14** | 72.9 | 67.7 | 32.6 | 60.3 | 76.9 | 90.9 | 93.5 | 79.5 | 95.2 | 53.5 | 74.9 | 72.5 |
| + TransCLIP-ZS | $77.2_{+4.3}$ | $73.5_{+5.9}$ | $35.3_{+2.7}$ | $75.9_{+15.6}$ | $79.0_{+2.1}$ | $91.9_{+1.0}$ | $94.7_{+1.2}$ | $85.3_{+5.8}$ | $97.4_{+2.3}$ | $60.0_{+6.5}$ | $81.7_{+6.7}$ | $77.4_{+4.9}$ |
| UPL* | 76.6 | 72.2 | 35.1 | 61.7 | 82.6 | 90.9 | 95.2 | 83.7 | 94.9 | 57.2 | 80.1 | 75.5 |
| **EVA-CLIP-8B** | 82.5 | 76.4 | 57.9 | 62.5 | 94.8 | 93.5 | 96.3 | 86.8 | 98.0 | 63.6 | 84.4 | 81.5 |
| + TransCLIP-ZS | $84.6_{+2.1}$ | $80.1_{+3.7}$ | $59.4_{+1.5}$ | $81.9_{+19.4}$ | $95.0_{+0.2}$ | $93.9_{+0.4}$ | $96.3_{+0.0}$ | $91.8_{+5.0}$ | $98.3_{+0.3}$ | $68.6_{+5.0}$ | $93.6_{+9.2}$ | $85.8_{+4.3}$ |

Table 10: TransCLIP atop inductive vision-language zero-shot and popular few-shot methods for ResNet-50 vision encoder.

| | Method | ImageNet | SUN397 | Aircraft | EuroSAT | StanfordCars | Food101 | Pets | Flower102 | Caltech101 | DTD | UCF101 | Average |
|---|---|---|---|---|---|---|---|---|---|---|---|---|---|
| 0-shot | **CLIP-ResNet-50** | 58.0 | 58.8 | 17.0 | 36.2 | 55.7 | 77.4 | 85.8 | 66.0 | 85.7 | 42.9 | 61.9 | 58.7 |
| | + TransCLIP-ZS | $60.8_{+2.8}$ | $64.2_{+5.4}$ | $16.6_{-0.4}$ | $59.6_{+23.4}$ | $57.9_{+2.2}$ | $78.0_{+0.6}$ | $89.3_{+3.6}$ | $72.2_{+6.2}$ | $88.6_{+3.0}$ | $47.8_{+5.0}$ | $68.8_{+6.9}$ | $64.0_{+5.3}$ |
| 1-shot | CoOp | 57.4 | 60.0 | 8.5 | 49.4 | 55.8 | 74.2 | 85.9 | 69.0 | 87.3 | 45.1 | 62.9 | 59.6 |
| | + TransCLIP-ZS | $60.2_{+2.8}$ | $65.3_{+5.3}$ | $9.3_{+0.8}$ | $57.1_{+7.7}$ | $58.8_{+3.0}$ | $77.0_{+2.7}$ | $86.9_{+1.1}$ | $81.6_{+12.6}$ | $88.7_{+1.4}$ | $53.2_{+8.2}$ | $69.1_{+6.1}$ | $64.3_{+4.7}$ |
| | TIP-Adapter-F | 61.1 | 62.1 | 18.6 | 50.2 | 59.2 | 77.1 | 86.3 | 78.1 | 88.3 | 47.6 | 64.7 | 63.0 |
| | + TransCLIP-ZS | $62.3_{+1.1}$ | $66.5_{+4.3}$ | $19.2_{+0.6}$ | $66.4_{+16.2}$ | $60.3_{+1.1}$ | $77.8_{+0.7}$ | $89.2_{+2.9}$ | $89.1_{+11.0}$ | $88.9_{+0.6}$ | $52.8_{+5.2}$ | $71.0_{+6.3}$ | $67.6_{+4.6}$ |
| | TaskRes | 61.4 | 62.0 | 20.9 | 59.8 | 59.4 | 74.8 | 84.4 | 75.4 | 88.5 | 49.6 | 64.5 | 63.7 |
| | + TransCLIP-ZS | $62.4_{+1.0}$ | $66.4_{+4.4}$ | $20.4_{-0.5}$ | $69.4_{+9.6}$ | $60.2_{+0.9}$ | $77.1_{+2.3}$ | $87.1_{+2.7}$ | $84.4_{+9.0}$ | $88.3_{-0.3}$ | $56.8_{+7.2}$ | $69.3_{+4.8}$ | $67.4_{+3.7}$ |
| | ProGrad | 57.8 | 60.9 | 18.9 | 55.0 | 55.0 | 76.3 | 88.0 | 72.2 | 88.1 | 46.4 | 64.1 | 62.4 |
| | + TransCLIP-ZS | $60.5_{+2.7}$ | $66.0_{+5.1}$ | $18.4_{-0.6}$ | $69.4_{+14.4}$ | $60.5_{+1.9}$ | $77.7_{+1.3}$ | $87.9_{-0.0}$ | $83.8_{+11.6}$ | $88.7_{+0.6}$ | $51.8_{+5.4}$ | $71.8_{+7.7}$ | $67.0_{+4.6}$ |
| 4-shot | CoOp | 59.8 | 63.5 | 20.5 | 71.3 | 62.9 | 73.8 | 87.0 | 85.7 | 89.3 | 54.0 | 67.6 | 66.8 |
| | + TransCLIP-ZS | $61.7_{+1.9}$ | $68.1_{+4.6}$ | $21.8_{+1.3}$ | $74.4_{+3.1}$ | $64.0_{+1.1}$ | $76.9_{+3.1}$ | $88.9_{+1.9}$ | $91.4_{+5.8}$ | $90.5_{+1.3}$ | $59.6_{+5.7}$ | $73.8_{+6.3}$ | $70.1_{+3.3}$ |
| | TIP-Adapter-F | 62.6 | 65.6 | 25.4 | 70.5 | 63.4 | 77.9 | 86.7 | 87.5 | 91.1 | 55.4 | 70.9 | 68.8 |
| | + TransCLIP-ZS | $63.0_{+0.4}$ | $68.5_{+2.9}$ | $24.6_{-0.7}$ | $70.9_{+0.4}$ | $63.2_{-0.2}$ | $78.1_{+0.2}$ | $89.0_{+2.4}$ | $92.4_{+4.9}$ | $90.3_{-0.8}$ | $59.7_{+4.3}$ | $76.7_{+5.8}$ | $70.6_{+1.8}$ |
| | TaskRes | 62.8 | 66.7 | 23.1 | 70.3 | 66.3 | 76.8 | 86.7 | 79.3 | 90.6 | 57.4 | 67.9 | 68.0 |
| | + TransCLIP-ZS | $63.3_{+0.5}$ | $69.2_{+2.5}$ | $21.7_{-1.5}$ | $72.2_{+1.9}$ | $64.5_{-1.8}$ | $77.9_{+1.2}$ | $88.9_{+2.2}$ | $85.1_{+5.8}$ | $90.4_{-0.1}$ | $60.9_{+3.5}$ | $74.4_{+6.5}$ | $69.9_{+1.9}$ |
| | ProGrad | 62.5 | 69.3 | 23.1 | 74.1 | 65.1 | 77.7 | 89.6 | 91.7 | 90.8 | 59.8 | 76.2 | 70.9 |
| | + TransCLIP-ZS | $62.5_{+1.2}$ | $69.3_{+3.3}$ | $23.1_{-0.4}$ | $74.1_{+3.6}$ | $65.1_{-0.3}$ | $77.7_{+1.8}$ | $89.6_{+1.2}$ | $91.7_{+7.4}$ | $90.8_{+1.0}$ | $59.8_{+5.3}$ | $76.2_{+7.4}$ | $70.9_{+2.9}$ |
| 16-shot | CoOp | 63.0 | 69.4 | 31.4 | 82.2 | 73.6 | 74.5 | 86.6 | 94.6 | 91.8 | 63.3 | 74.4 | 73.2 |
| | + TransCLIP-ZS | $63.5_{+0.5}$ | $71.1_{+1.7}$ | $29.9_{-1.5}$ | $81.3_{-0.8}$ | $70.5_{-3.1}$ | $77.5_{+2.9}$ | $88.6_{+2.0}$ | $94.9_{+0.3}$ | $91.4_{-0.5}$ | $65.7_{+2.4}$ | $79.4_{+5.0}$ | $74.0_{+0.8}$ |
| | TIP-Adapter-F | 65.2 | 71.2 | 33.9 | 83.3 | 74.3 | 78.9 | 89.0 | 92.7 | 92.5 | 66.0 | 76.5 | 74.9 |
| | + TransCLIP-ZS | $64.6_{-0.5}$ | $71.0_{-0.2}$ | $33.3_{-0.7}$ | $80.5_{-2.8}$ | $72.9_{-1.5}$ | $78.4_{+0.5}$ | $89.4_{+0.5}$ | $94.1_{+1.4}$ | $91.0_{-1.6}$ | $65.5_{-0.6}$ | $79.9_{+3.4}$ | $74.6_{-0.3}$ |
| | TaskRes | 64.4 | 70.8 | 29.1 | 75.5 | 69.8 | 78.6 | 89.3 | 94.7 | 90.5 | 64.7 | 79.4 | 73.3 |
| | + TransCLIP-ZS | $64.4_{-0.2}$ | $70.8_{+0.6}$ | $29.1_{-3.9}$ | $75.5_{-4.2}$ | $69.8_{-5.0}$ | $78.6_{+0.1}$ | $89.3_{+0.8}$ | $94.7_{+0.3}$ | $90.5_{-2.5}$ | $64.7_{-2.6}$ | $79.4_{+3.6}$ | $73.3_{-1.2}$ |
| | ProGrad | 63.4 | 69.9 | 31.8 | 81.9 | 73.9 | 77.0 | 88.2 | 94.2 | 92.3 | 63.6 | 75.4 | 73.8 |
| | + TransCLIP-ZS | $63.5_{+0.2}$ | $71.2_{+1.3}$ | $29.0_{-2.8}$ | $79.8_{-2.2}$ | $70.2_{-3.7}$ | $78.4_{+1.4}$ | $89.2_{+1.0}$ | $94.5_{+0.3}$ | $91.0_{-1.3}$ | $65.7_{+2.0}$ | $79.9_{+4.5}$ | $73.9_{+0.1}$ |

Table 11: TransCLIP atop inductive vision-language zero-shot and popular few-shot methods for ResNet-101 vision encoder.

| Shots | Method | ImageNet | SUN397 | Aircraft | EuroSAT | StanfordCars | Food101 | Pets | Flower102 | Caltech101 | DTD | UCF101 | Average |
|---|---|---|---|---|---|---|---|---|---|---|---|---|---|
| 0-shot | **CLIP-ResNet-101** | 60.6 | 59.0 | 17.9 | 32.7 | 63.2 | 80.7 | 87.0 | 64.4 | 89.9 | 37.2 | 61.1 | 59.4 |
| | + TransCLIP-ZS | 64.8$_{+4.2}$ | 65.1$_{+6.0}$ | 19.2$_{+1.3}$ | 59.3$_{+26.6}$ | 68.6$_{+5.4}$ | 81.9$_{+1.2}$ | 89.8$_{+2.7}$ | 72.6$_{+8.2}$ | 93.0$_{+3.1}$ | 42.9$_{+5.7}$ | 68.9$_{+7.8}$ | 66.0$_{+6.6}$ |
| 1-shot | CoOp | 60.8 | 61.3 | 14.8 | 51.0 | 64.5 | 76.9 | 86.5 | 69.5 | 89.8 | 44.3 | 65.7 | 62.3 |
| | + TransCLIP-ZS | 64.3$_{+3.5}$ | 66.3$_{+4.9}$ | 16.3$_{+1.6}$ | 58.2$_{+7.2}$ | 70.2$_{+5.7}$ | 79.8$_{+2.9}$ | 89.1$_{+2.6}$ | 80.6$_{+11.1}$ | 92.5$_{+2.8}$ | 49.9$_{+5.5}$ | 72.7$_{+6.9}$ | 67.3$_{+5.0}$ |
| | TIP-Adapter-F | 63.6 | 61.4 | 19.2 | 46.3 | 64.8 | 80.2 | 87.2 | 77.5 | 91.7 | 46.3 | 65.9 | 64.0 |
| | + TransCLIP-ZS | 66.4$_{+2.8}$ | 67.1$_{+5.7}$ | 21.0$_{+1.8}$ | 66.1$_{+19.8}$ | 70.6$_{+5.7}$ | 81.8$_{+1.6}$ | 90.3$_{+3.1}$ | 88.4$_{+10.9}$ | 92.8$_{+1.1}$ | 51.7$_{+5.4}$ | 73.1$_{+7.3}$ | 69.9$_{+5.9}$ |
| | TaskRes | 63.6 | 62.6 | 22.5 | 52.9 | 66.4 | 78.2 | 86.3 | 74.8 | 91.2 | 49.2 | 67.3 | 65.0 |
| | + TransCLIP-ZS | 66.6$_{+3.0}$ | 67.9$_{+5.3}$ | 23.3$_{+0.8}$ | 64.0$_{+11.1}$ | 70.3$_{+3.9}$ | 80.7$_{+2.4}$ | 89.7$_{+3.3}$ | 85.4$_{+10.6}$ | 92.1$_{+0.8}$ | 53.5$_{+4.3}$ | 74.2$_{+6.9}$ | 69.8$_{+4.8}$ |
| 4-shot | CoOp | 63.0 | 65.9 | 26.8 | 67.4 | 70.3 | 77.8 | 87.4 | 85.5 | 92.3 | 55.5 | 72.3 | 69.5 |
| | + TransCLIP-ZS | 66.0$_{+3.0}$ | 69.7$_{+3.8}$ | 27.7$_{+1.0}$ | 71.4$_{+4.1}$ | 73.8$_{+3.5}$ | 80.5$_{+2.7}$ | 90.1$_{+2.7}$ | 91.4$_{+5.8}$ | 93.8$_{+1.5}$ | 59.7$_{+4.1}$ | 77.3$_{+4.9}$ | 72.9$_{+3.4}$ |
| | TIP-Adapter-F | 65.0 | 65.3 | 27.4 | 68.6 | 70.9 | 81.2 | 88.4 | 90.1 | 93.0 | 58.3 | 74.1 | 71.1 |
| | + TransCLIP-ZS | 67.4$_{+2.4}$ | 69.8$_{+4.4}$ | 28.6$_{+1.1}$ | 70.8$_{+2.2}$ | 74.0$_{+3.1}$ | 82.2$_{+1.0}$ | 90.6$_{+2.2}$ | 93.5$_{+3.3}$ | 93.6$_{+0.6}$ | 62.0$_{+3.7}$ | 79.2$_{+5.1}$ | 73.8$_{+2.7}$ |
| | TaskRes | 65.3 | 68.0 | 24.3 | 61.9 | 72.4 | 80.4 | 88.0 | 78.6 | 92.9 | 56.5 | 71.0 | 69.0 |
| | + TransCLIP-ZS | 67.7$_{+2.4}$ | 71.3$_{+3.3}$ | 25.4$_{+1.1}$ | 68.5$_{+6.5}$ | 74.9$_{+2.5}$ | 81.8$_{+1.4}$ | 90.9$_{+2.8}$ | 86.9$_{+8.3}$ | 93.8$_{+0.9}$ | 61.6$_{+5.2}$ | 78.5$_{+7.5}$ | 72.8$_{+3.8}$ |
| 16-shot | CoOp | 66.5 | 71.0 | 34.8 | 83.4 | 79.1 | 78.9 | 89.0 | 95.1 | 93.5 | 65.1 | 78.1 | 75.9 |
| | + TransCLIP-ZS | 68.5$_{+2.0}$ | 73.0$_{+2.0}$ | 34.9$_{+0.1}$ | 83.0$_{-0.4}$ | 79.8$_{+0.8}$ | 81.1$_{+2.3}$ | 90.9$_{+1.9}$ | 95.8$_{+0.7}$ | 93.6$_{+0.1}$ | 68.2$_{+3.1}$ | 81.2$_{+3.1}$ | 77.3$_{+1.4}$ |
| | TIP-Adapter-F | 68.3 | 72.8 | 36.2 | 82.0 | 80.5 | 81.9 | 89.9 | 94.4 | 94.2 | 67.6 | 79.4 | 77.0 |
| | + TransCLIP-ZS | 69.2$_{+0.9}$ | 73.5$_{+0.7}$ | 36.6$_{+0.4}$ | 80.0$_{-2.0}$ | 81.3$_{+0.8}$ | 82.4$_{+0.5}$ | 91.7$_{+1.8}$ | 95.3$_{+0.9}$ | 94.2$_{+0.3}$ | 68.0$_{+0.3}$ | 81.9$_{+2.6}$ | 77.7$_{+0.7}$ |
| | TaskRes | 67.6 | 72.1 | 35.5 | 74.9 | 80.6 | 81.9 | 89.5 | 94.9 | 94.6 | 68.1 | 79.5 | 76.3 |
| | + TransCLIP-ZS | 69.3$_{+1.7}$ | 73.3$_{+1.2}$ | 34.8$_{-0.8}$ | 73.9$_{-1.0}$ | 80.8$_{+0.2}$ | 82.6$_{+0.7}$ | 90.9$_{+1.4}$ | 95.5$_{+0.6}$ | 94.5$_{-0.1}$ | 68.3$_{+0.2}$ | 82.8$_{+3.3}$ | 77.0$_{+0.7}$ |

Table 12: TransCLIP atop inductive vision-language zero-shot and popular few-shot methods for ViT-B/32 vision encoder.

| Shots | Method | ImageNet | SUN397 | Aircraft | EuroSAT | StanfordCars | Food101 | Pets | Flower102 | Caltech101 | DTD | UCF101 | Average |
|---|---|---|---|---|---|---|---|---|---|---|---|---|---|
| 0-shot | **CLIP-ViT-B/32** | 61.9 | 62.1 | 19.1 | 45.2 | 60.2 | 80.4 | 87.4 | 66.5 | 91.5 | 42.7 | 63.6 | 61.9 |
| | + TransCLIP-ZS | 64.9$_{+3.0}$ | 67.6$_{+5.5}$ | 20.3$_{+1.3}$ | 59.0$_{+13.8}$ | 63.3$_{+3.2}$ | 81.5$_{+1.1}$ | 89.0$_{+1.7}$ | 74.4$_{+7.9}$ | 91.8$_{+0.3}$ | 50.4$_{+7.7}$ | 68.7$_{+5.1}$ | 66.5$_{+4.6}$ |
| 1-shot | CoOp | 60.8 | 63.3 | 15.6 | 51.9 | 59.5 | 75.7 | 87.7 | 71.5 | 91.8 | 47.1 | 66.0 | 62.8 |
| | + TransCLIP-ZS | 63.9$_{+3.1}$ | 68.3$_{+5.0}$ | 17.7$_{+2.0}$ | 64.9$_{+13.0}$ | 63.4$_{+3.9}$ | 78.8$_{+3.0}$ | 89.2$_{+1.5}$ | 84.3$_{+12.8}$ | 92.2$_{+0.4}$ | 53.1$_{+5.9}$ | 71.9$_{+5.9}$ | 68.0$_{+5.1}$ |
| | TIP-Adapter-F | 64.3 | 65.4 | 22.2 | 59.7 | 61.1 | 80.4 | 89.2 | 81.1 | 92.4 | 50.9 | 66.5 | 66.5 |
| | + TransCLIP-ZS | 66.5$_{+2.1}$ | 69.9$_{+4.5}$ | 23.3$_{+1.1}$ | 71.8$_{+12.1}$ | 64.8$_{+3.7}$ | 81.4$_{+1.0}$ | 89.5$_{+1.9}$ | 89.6$_{+8.5}$ | 92.3$_{-0.2}$ | 55.9$_{+5.0}$ | 72.5$_{+5.9}$ | 70.7$_{+4.1}$ |
| | TaskRes | 64.6 | 65.3 | 23.8 | 60.8 | 62.4 | 79.0 | 84.6 | 77.7 | 91.2 | 52.7 | 67.5 | 66.3 |
| | + TransCLIP-ZS | 66.7$_{+2.1}$ | 69.8$_{+4.5}$ | 23.9$_{+0.1}$ | 73.4$_{+12.6}$ | 64.2$_{+1.8}$ | 80.7$_{+1.7}$ | 88.2$_{+3.5}$ | 86.6$_{+8.9}$ | 91.8$_{+0.6}$ | 57.0$_{+4.3}$ | 72.8$_{+5.3}$ | 70.5$_{+4.1}$ |
| | ProGrad | 62.0 | 64.8 | 21.1 | 53.5 | 60.5 | 78.2 | 87.9 | 74.4 | 91.5 | 51.1 | 66.6 | 64.7 |
| | + TransCLIP-ZS | 64.9$_{+2.9}$ | 69.2$_{+4.4}$ | 22.2$_{+1.1}$ | 63.3$_{+9.8}$ | 63.6$_{+3.1}$ | 80.2$_{+2.0}$ | 89.6$_{+1.7}$ | 86.4$_{+12.0}$ | 92.1$_{+0.6}$ | 55.8$_{+4.7}$ | 72.0$_{+5.4}$ | 69.0$_{+4.3}$ |
| 4-shot | CoOp | 63.2 | 67.1 | 24.1 | 67.8 | 66.4 | 75.5 | 88.8 | 92.9 | 92.9 | 55.1 | 74.9 | 69.4 |
| | + TransCLIP-ZS | 65.7$_{+2.6}$ | 70.7$_{+3.7}$ | 25.3$_{+1.3}$ | 77.2$_{+9.4}$ | 69.4$_{+3.0}$ | 78.8$_{+3.3}$ | 90.5$_{+1.7}$ | 92.0$_{+4.4}$ | 93.4$_{+0.5}$ | 59.3$_{+4.2}$ | 79.3$_{+4.4}$ | 72.9$_{+3.5}$ |
| | TIP-Adapter-F | 65.8 | 68.3 | 28.8 | 71.5 | 67.6 | 80.9 | 88.6 | 88.9 | 94.6 | 58.0 | 75.1 | 71.6 |
| | + TransCLIP-ZS | 67.5$_{+1.7}$ | 72.0$_{+3.7}$ | 28.5$_{-0.3}$ | 76.8$_{+5.3}$ | 68.5$_{+0.9}$ | 81.7$_{+0.8}$ | 90.2$_{+1.6}$ | 93.5$_{+3.5}$ | 93.8$_{-0.8}$ | 62.1$_{+4.1}$ | 78.5$_{+3.5}$ | 73.8$_{+2.2}$ |
| | TaskRes | 66.1 | 70.1 | 25.3 | 68.8 | 69.5 | 80.4 | 87.3 | 81.8 | 93.9 | 57.9 | 71.7 | 70.2 |
| | + TransCLIP-ZS | 67.8$_{+1.7}$ | 72.7$_{+2.6}$ | 25.6$_{+0.4}$ | 77.1$_{+8.3}$ | 70.3$_{+0.8}$ | 81.6$_{+1.1}$ | 90.0$_{+2.7}$ | 88.3$_{+6.5}$ | 94.2$_{+0.3}$ | 61.9$_{+4.0}$ | 76.1$_{+4.4}$ | 73.2$_{+3.0}$ |
| | ProGrad | 65.2 | 69.6 | 24.8 | 63.0 | 66.5 | 79.2 | 89.4 | 87.7 | 93.4 | 56.1 | 73.7 | 69.9 |
| | + TransCLIP-ZS | 67.1$_{+1.9}$ | 72.7$_{+3.0}$ | 25.6$_{+0.8}$ | 74.0$_{+11.0}$ | 69.5$_{+2.9}$ | 80.9$_{+1.7}$ | 91.1$_{+1.7}$ | 92.7$_{+5.0}$ | 92.8$_{-0.5}$ | 61.2$_{+5.1}$ | 78.0$_{+4.3}$ | 73.2$_{+3.4}$ |
| 16-shot | CoOp | 66.8 | 72.3 | 32.8 | 82.4 | 76.1 | 78.6 | 88.8 | 95.5 | 94.9 | 64.9 | 78.5 | 75.6 |
| | + TransCLIP-ZS | 68.4$_{+1.6}$ | 74.2$_{+1.9}$ | 32.7$_{-0.1}$ | 84.0$_{+1.6}$ | 77.1$_{+1.0}$ | 80.7$_{+2.2}$ | 90.2$_{+1.4}$ | 95.5$_{-0.0}$ | 95.4$_{+0.5}$ | 67.3$_{+2.4}$ | 81.2$_{+2.7}$ | 77.0$_{+1.4}$ |
| | TIP-Adapter-F | 68.4 | 74.1 | 34.8 | 83.4 | 77.0 | 81.7 | 90.4 | 94.3 | 95.1 | 68.0 | 80.5 | 77.1 |
| | + TransCLIP-ZS | 69.0$_{+0.5}$ | 74.8$_{+0.7}$ | 35.0$_{+0.2}$ | 84.1$_{+0.7}$ | 77.3$_{+0.3}$ | 82.0$_{+0.3}$ | 91.0$_{+0.6}$ | 95.3$_{+1.0}$ | 95.1$_{-0.1}$ | 67.4$_{-0.6}$ | 82.5$_{+2.0}$ | 77.6$_{+0.5}$ |
| | TaskRes | 68.2 | 73.5 | 37.0 | 76.9 | 77.2 | 81.4 | 89.4 | 95.5 | 95.6 | 68.1 | 80.3 | 76.7 |
| | + TransCLIP-ZS | 69.2$_{+1.1}$ | 74.6$_{+1.0}$ | 35.3$_{-1.7}$ | 80.3$_{+3.4}$ | 77.2$_{-0.8}$ | 82.0$_{+0.6}$ | 90.7$_{+1.3}$ | 95.1$_{-0.4}$ | 94.8$_{-0.9}$ | 67.8$_{-0.4}$ | 82.3$_{+2.0}$ | 77.2$_{-0.5}$ |
| | ProGrad | 66.9 | 73.2 | 33.2 | 80.6 | 76.2 | 80.2 | 89.4 | 95.1 | 95.0 | 65.3 | 80.0 | 75.9 |
| | + TransCLIP-ZS | 68.4$_{+1.5}$ | 74.8$_{+1.5}$ | 33.2$_{-0.0}$ | 82.8$_{+2.2}$ | 77.1$_{+0.9}$ | 81.6$_{+1.4}$ | 90.4$_{+1.0}$ | 95.3$_{+0.3}$ | 94.3$_{-0.8}$ | 67.8$_{+2.5}$ | 82.7$_{+2.8}$ | 77.1$_{+1.2}$ |

Table 13: TransCLIP atop inductive vision-language zero-shot and popular few-shot methods for ViT-B/16 vision encoder.

| | Method | ImageNet | SUN397 | Aircraft | EuroSAT | StanfordCars | Food101 | Pets | Flower102 | Caltech101 | DTD | UCF101 | Average |
|---|---|---|---|---|---|---|---|---|---|---|---|---|---|
| 0-shot | **CLIP-ViT-B/16** | 66.6 | 62.5 | 24.7 | 48.3 | 65.6 | 85.9 | 89.1 | 70.7 | 93.2 | 43.5 | 67.5 | 65.3 |
| | + TransCLIP-ZS | 70.3$_{+3.7}$ | 68.9$_{+6.3}$ | 26.9$_{+2.2}$ | 65.1$_{+16.8}$ | 69.4$_{+3.8}$ | 87.1$_{+1.2}$ | 92.6$_{+3.5}$ | 76.7$_{+5.9}$ | 92.7$_{-0.5}$ | 49.5$_{+6.0}$ | 74.4$_{+6.9}$ | 70.3$_{+5.1}$ |
| 1-shot | CoOp | 65.7 | 66.9 | 20.7 | 56.4 | 67.6 | 84.3 | 90.2 | 78.2 | 92.5 | 50.1 | 71.2 | 67.6 |
| | + TransCLIP-ZS | 69.3$_{+3.6}$ | 71.5$_{+4.6}$ | 23.8$_{+3.1}$ | 65.3$_{+8.9}$ | 71.9$_{+4.3}$ | 86.3$_{+2.0}$ | 91.9$_{+1.8}$ | 89.8$_{+11.5}$ | 93.8$_{+1.3}$ | 55.4$_{+5.4}$ | 77.7$_{+6.5}$ | 72.4$_{+4.8}$ |
| | TIP-Adapter-F | 69.5 | 67.2 | 28.8 | 67.8 | 67.1 | 85.8 | 90.6 | 83.7 | 94.0 | 51.6 | 73.4 | 70.9 |
| | + TransCLIP-ZS | 72.0$_{+2.5}$ | 71.8$_{+4.6}$ | 30.7$_{+1.9}$ | 76.9$_{+9.1}$ | 71.0$_{+3.9}$ | 86.9$_{+1.1}$ | 93.1$_{+2.4}$ | 92.8$_{+9.1}$ | 93.5$_{-0.5}$ | 57.7$_{+6.1}$ | 80.0$_{+6.7}$ | 75.1$_{+4.3}$ |
| | PLOT | 66.9 | 67.0 | 28.9 | 72.8 | 68.5 | 84.9 | 91.9 | 81.8 | 94.0 | 52.8 | 74.7 | 71.3 |
| | + TransCLIP-ZS | 75.8$_{+8.9}$ | 70.3$_{+3.3}$ | 28.1$_{-0.8}$ | 78.8$_{+6.0}$ | 70.0$_{+1.6}$ | 85.3$_{+0.4}$ | 91.1$_{-0.8}$ | 93.2$_{+11.4}$ | 94.0$_{-0.0}$ | 56.7$_{+3.9}$ | 81.4$_{+6.7}$ | 75.0$_{+3.7}$ |
| | TaskRes | 69.6 | 68.1 | 31.2 | 65.6 | 69.1 | 84.5 | 90.2 | 81.6 | 93.6 | 53.4 | 71.8 | 70.8 |
| | + TransCLIP-ZS | 72.0$_{+2.5}$ | 72.5$_{+4.4}$ | 31.4$_{+0.2}$ | 73.7$_{+8.1}$ | 71.6$_{+2.4}$ | 86.5$_{+2.0}$ | 91.6$_{+1.5}$ | 90.7$_{+9.1}$ | 94.0$_{+0.4}$ | 59.4$_{+6.0}$ | 76.4$_{+4.6}$ | 74.5$_{+3.7}$ |
| | ProGrad | 67.0 | 67.0 | 28.7 | 57.0 | 68.2 | 84.9 | 91.4 | 80.8 | 93.5 | 52.8 | 73.3 | 69.5 |
| | + TransCLIP-ZS | 70.1$_{+3.1}$ | 71.6$_{+4.6}$ | 30.5$_{+1.8}$ | 70.9$_{+13.9}$ | 72.3$_{+4.1}$ | 86.5$_{+1.6}$ | 92.7$_{+1.4}$ | 91.5$_{+10.7}$ | 94.1$_{+0.7}$ | 57.9$_{+5.1}$ | 79.3$_{+6.1}$ | 74.3$_{+4.8}$ |
| 4-shot | CoOp | 68.8 | 69.7 | 30.8 | 69.6 | 74.4 | 84.5 | 92.5 | 92.2 | 94.5 | 59.4 | 77.5 | 74.0 |
| | + TransCLIP-ZS | 71.4$_{+2.6}$ | 73.3$_{+3.6}$ | 33.1$_{+2.3}$ | 77.2$_{+7.5}$ | 77.7$_{+3.2}$ | 86.5$_{+1.9}$ | 93.6$_{+1.1}$ | 95.3$_{+3.1}$ | 95.1$_{+0.6}$ | 63.0$_{+3.6}$ | 81.8$_{+4.3}$ | 77.1$_{+3.1}$ |
| | TIP-Adapter-F | 70.7 | 70.8 | 35.7 | 76.8 | 74.1 | 86.5 | 91.9 | 92.1 | 94.8 | 59.8 | 78.1 | 75.6 |
| | + TransCLIP-ZS | 72.7$_{+1.9}$ | 74.4$_{+3.5}$ | 36.1$_{+0.4}$ | 79.7$_{+2.9}$ | 75.9$_{+1.8}$ | 87.4$_{+0.9}$ | 93.2$_{+1.3}$ | 95.5$_{+3.3}$ | 95.1$_{+0.3}$ | 64.0$_{+4.2}$ | 83.3$_{+5.2}$ | 77.9$_{+2.3}$ |
| | PLOT | 70.0 | 71.8 | 34.8 | 84.7 | 76.6 | 83.5 | 92.8 | 93.2 | 94.9 | 61.0 | 79.7 | 76.6 |
| | + TransCLIP-ZS | 77.2$_{+7.2}$ | 73.5$_{+1.7}$ | 33.9$_{-0.9}$ | 81.8$_{-2.9}$ | 75.8$_{-0.8}$ | 85.6$_{+2.2}$ | 92.5$_{-0.3}$ | 95.8$_{+2.6}$ | 94.8$_{-0.1}$ | 63.6$_{+2.6}$ | 83.3$_{+3.6}$ | 78.0$_{+1.4}$ |
| | TaskRes | 71.0 | 72.8 | 33.3 | 73.8 | 76.6 | 86.1 | 91.9 | 85.0 | 94.9 | 59.7 | 75.5 | 74.6 |
| | + TransCLIP-ZS | 73.0$_{+2.0}$ | 75.3$_{+2.5}$ | 34.4$_{+1.1}$ | 78.1$_{+4.4}$ | 77.2$_{+1.1}$ | 87.3$_{+1.2}$ | 93.0$_{+1.1}$ | 92.4$_{+7.4}$ | 95.1$_{+0.2}$ | 64.3$_{+4.6}$ | 79.2$_{+3.7}$ | 77.2$_{+2.7}$ |
| | ProGrad | 70.2 | 71.7 | 34.0 | 69.5 | 75.0 | 85.4 | 92.0 | 91.1 | 94.4 | 59.8 | 77.9 | 74.6 |
| | + TransCLIP-ZS | 72.3$_{+2.1}$ | 75.0$_{+3.3}$ | 35.5$_{+1.6}$ | 74.9$_{+5.3}$ | 77.9$_{+2.9}$ | 87.0$_{+1.5}$ | 93.7$_{+1.7}$ | 95.3$_{+4.2}$ | 95.1$_{+0.8}$ | 64.8$_{+5.1}$ | 83.2$_{+5.4}$ | 77.7$_{+3.1}$ |
| 16-shot | CoOp | 71.9 | 74.9 | 43.3 | 85.0 | 82.8 | 84.2 | 91.9 | 96.8 | 95.8 | 69.7 | 83.1 | 79.9 |
| | + TransCLIP-ZS | 73.3$_{+1.4}$ | 76.6$_{+1.8}$ | 42.9$_{-0.4}$ | 86.0$_{+1.0}$ | 83.0$_{+0.2}$ | 86.3$_{+2.1}$ | 93.2$_{+1.2}$ | 97.5$_{+0.8}$ | 95.9$_{+0.1}$ | 71.3$_{+1.7}$ | 85.4$_{+2.3}$ | 81.1$_{+1.1}$ |
| | TIP-Adapter-F | 73.3 | 76.0 | 44.6 | 85.9 | 82.3 | 86.8 | 92.6 | 96.2 | 95.7 | 70.8 | 83.9 | 80.7 |
| | + TransCLIP-ZS | 74.2$_{+0.9}$ | 76.8$_{+0.8}$ | 44.9$_{+0.3}$ | 85.2$_{-0.7}$ | 82.7$_{+0.4}$ | 87.4$_{+0.6}$ | 93.5$_{+0.9}$ | 96.9$_{+0.7}$ | 95.7$_{-0.1}$ | 69.2$_{-1.5}$ | 85.6$_{+1.7}$ | 81.1$_{-0.4}$ |
| | PLOT | 72.5 | 76.0 | 46.8 | 92.1 | 84.6 | 85.6 | 92.5 | 97.1 | 96.0 | 71.1 | 84.8 | 81.7 |
| | + TransCLIP-ZS | 77.8$_{+5.3}$ | 75.0$_{-1.0}$ | 41.8$_{-4.9}$ | 84.6$_{-7.5}$ | 79.6$_{-4.9}$ | 85.9$_{+0.2}$ | 92.2$_{-0.4}$ | 97.3$_{+0.2}$ | 95.0$_{-1.0}$ | 68.7$_{-2.4}$ | 83.7$_{-1.0}$ | 80.3$_{-1.4}$ |
| | TaskRes | 73.0 | 76.0 | 44.8 | 80.7 | 83.5 | 86.9 | 92.5 | 97.3 | 95.9 | 70.9 | 83.4 | 80.5 |
| | + TransCLIP-ZS | 74.1$_{+1.0}$ | 76.9$_{+0.8}$ | 43.6$_{-1.2}$ | 80.5$_{-0.3}$ | 82.8$_{-0.7}$ | 87.5$_{+0.6}$ | 92.9$_{+0.4}$ | 97.6$_{+0.3}$ | 96.0$_{+0.1}$ | 70.2$_{-0.7}$ | 86.2$_{+2.8}$ | 80.8$_{+0.3}$ |
| | ProGrad | 72.1 | 75.1 | 42.8 | 83.6 | 82.9 | 85.8 | 92.9 | 96.6 | 95.9 | 68.9 | 82.6 | 79.9 |
| | + TransCLIP-ZS | 73.5$_{+1.4}$ | 76.8$_{+1.7}$ | 42.8$_{-0.0}$ | 83.7$_{+0.2}$ | 83.1$_{+0.2}$ | 87.2$_{+1.3}$ | 93.7$_{+0.8}$ | 97.4$_{+0.8}$ | 96.0$_{+0.1}$ | 71.4$_{+2.5}$ | 86.1$_{+3.4}$ | 81.1$_{+1.1}$ |

Table 14: TransCLIP atop inductive vision-language zero-shot and popular few-shot methods for ViT-L/14 vision encoder.

| | Method | ImageNet | SUN397 | Aircraft | EuroSAT | StanfordCars | Food101 | Pets | Flower102 | Caltech101 | DTD | UCF101 | Average |
|---|---|---|---|---|---|---|---|---|---|---|---|---|---|
| 0-shot | **CLIP-ViT-L/14** | 72.9 | 67.7 | 32.6 | 60.3 | 76.9 | 90.9 | 93.5 | 79.5 | 95.2 | 53.5 | 74.9 | 72.5 |
| | + TransCLIP-ZS | 77.2$_{+4.3}$ | 73.5$_{+5.9}$ | 35.3$_{+2.7}$ | 75.9$_{+15.6}$ | 79.0$_{+2.1}$ | 91.9$_{+1.0}$ | 94.7$_{+1.2}$ | 85.3$_{+5.8}$ | 97.4$_{+2.3}$ | 60.0$_{+6.5}$ | 81.7$_{+6.7}$ | 77.4$_{+4.9}$ |
| 1-shot | CoOp | 71.5 | 68.9 | 36.9 | 68.4 | 78.8 | 89.0 | 94.0 | 87.2 | 95.0 | 58.6 | 78.7 | 75.2 |
| | + TransCLIP-ZS | 75.9$_{+4.5}$ | 74.3$_{+5.4}$ | 38.0$_{+1.0}$ | 80.4$_{+11.9}$ | 81.5$_{+2.8}$ | 91.0$_{+2.1}$ | 95.3$_{+1.4}$ | 95.0$_{+7.8}$ | 96.3$_{+1.3}$ | 64.1$_{+5.5}$ | 83.5$_{+4.8}$ | 79.6$_{+4.4}$ |
| | TIP-Adapter-F | 76.4 | 71.0 | 38.5 | 67.8 | 79.2 | 91.0 | 93.2 | 90.9 | 95.3 | 59.3 | 77.9 | 76.4 |
| | + TransCLIP-ZS | 78.8$_{+2.4}$ | 75.5$_{+4.5}$ | 40.9$_{+2.4}$ | 75.5$_{+7.7}$ | 80.5$_{+1.3}$ | 91.9$_{+0.9}$ | 94.1$_{+0.9}$ | 97.4$_{+6.5}$ | 96.9$_{+1.6}$ | 64.9$_{+5.6}$ | 83.8$_{+5.9}$ | 80.0$_{+3.6}$ |
| | TaskRes | 76.2 | 71.4 | 39.6 | 71.8 | 79.9 | 89.8 | 93.5 | 87.4 | 95.0 | 60.1 | 77.7 | 76.6 |
| | + TransCLIP-ZS | 78.8$_{+2.5}$ | 75.9$_{+4.5}$ | 41.2$_{+1.6}$ | 82.0$_{+10.2}$ | 81.1$_{+1.2}$ | 91.5$_{+1.6}$ | 94.9$_{+1.4}$ | 94.7$_{+7.2}$ | 96.2$_{+1.2}$ | 65.7$_{+5.6}$ | 83.8$_{+6.1}$ | 80.5$_{+3.9}$ |
| | ProGrad | 73.6 | 71.1 | 38.4 | 71.4 | 80.0 | 90.5 | 94.4 | 89.0 | 95.7 | 58.8 | 80.2 | 76.6 |
| | + TransCLIP-ZS | 76.9$_{+3.3}$ | 75.8$_{+4.7}$ | 41.1$_{+2.8}$ | 78.7$_{+7.3}$ | 81.1$_{+1.1}$ | 91.7$_{+1.2}$ | 95.6$_{+1.2}$ | 97.5$_{+8.5}$ | 96.4$_{+0.7}$ | 65.9$_{+7.0}$ | 84.0$_{+3.8}$ | 80.4$_{+3.8}$ |
| 4-shot | CoOp | 74.9 | 73.1 | 43.6 | 76.2 | 83.3 | 88.8 | 94.6 | 95.9 | 96.7 | 64.1 | 83.3 | 79.5 |
| | + TransCLIP-ZS | 77.9$_{+3.0}$ | 76.9$_{+3.8}$ | 44.0$_{+0.5}$ | 81.6$_{+5.5}$ | 84.0$_{+0.7}$ | 91.2$_{+2.4}$ | 95.8$_{+1.2}$ | 97.3$_{+1.4}$ | 97.4$_{+0.7}$ | 67.7$_{+3.6}$ | 85.9$_{+3.0}$ | 81.8$_{+2.3}$ |
| | TIP-Adapter-F | 77.0 | 74.1 | 47.4 | 81.4 | 82.3 | 91.2 | 94.0 | 95.5 | 96.5 | 64.4 | 83.9 | 80.7 |
| | + TransCLIP-ZS | 79.0$_{+2.0}$ | 77.2$_{+3.1}$ | 47.6$_{+0.2}$ | 83.0$_{+1.6}$ | 82.9$_{+0.6}$ | 91.9$_{+0.7}$ | 94.8$_{+0.8}$ | 98.5$_{+3.0}$ | 97.5$_{+1.1}$ | 69.0$_{+4.6}$ | 80.0 | 82.6$_{+1.9}$ |
| | TaskRes | 77.1 | 74.9 | 42.5 | 77.3 | 83.6 | 90.6 | 94.4 | 90.1 | 96.6 | 65.1 | 80.0 | 79.3 |
| | + TransCLIP-ZS | 79.4$_{+2.2}$ | 78.5$_{+3.6}$ | 44.9$_{+2.4}$ | 81.4$_{+4.1}$ | 83.2$_{-0.3}$ | 91.8$_{+1.1}$ | 95.7$_{+1.3}$ | 96.5$_{+6.4}$ | 97.7$_{+1.1}$ | 68.0$_{+2.9}$ | 86.1$_{+6.1}$ | 82.1$_{+2.8}$ |
| | ProGrad | 76.5 | 74.9 | 44.5 | 77.3 | 83.9 | 90.6 | 94.8 | 95.6 | 96.7 | 66.1 | 83.9 | 80.6 |
| | + TransCLIP-ZS | 78.8$_{+2.3}$ | 78.2$_{+3.2}$ | 46.8$_{+2.3}$ | 82.6$_{+3.3}$ | 84.0$_{+0.1}$ | 91.8$_{+1.2}$ | 95.8$_{+1.1}$ | 97.9$_{+2.3}$ | 97.4$_{+0.6}$ | 70.3$_{+4.2}$ | 87.7$_{+3.8}$ | 82.8$_{+2.2}$ |
| 16-shot | CoOp | 78.2 | 77.5 | 55.4 | 87.2 | 89.1 | 89.8 | 94.6 | 99.1 | 97.2 | 74.1 | 87.2 | 84.5 |
| | + TransCLIP-ZS | 79.5$_{+1.3}$ | 79.8$_{+2.3}$ | 54.6$_{-0.7}$ | 90.5$_{+3.4}$ | 88.0$_{-1.1}$ | 91.5$_{+1.7}$ | 95.4$_{+0.8}$ | 99.4$_{+0.4}$ | 98.1$_{+0.9}$ | 75.3$_{+1.2}$ | 88.1$_{+0.9}$ | 85.6$_{+1.1}$ |
| | TIP-Adapter-F | 79.3 | 79.6 | 55.8 | 86.1 | 88.1 | 91.6 | 94.6 | 98.3 | 97.5 | 74.0 | 87.4 | 84.7 |
| | + TransCLIP-ZS | 80.1$_{+0.9}$ | 80.0$_{+0.4}$ | 56.0$_{+0.2}$ | 88.8$_{+2.7}$ | 87.4$_{-0.7}$ | 91.9$_{+0.4}$ | 95.7$_{+1.1}$ | 99.1$_{+0.9}$ | 97.9$_{+0.4}$ | 73.9$_{-0.1}$ | 88.8$_{+1.4}$ | 85.4$_{+0.7}$ |
| | TaskRes | 78.1 | 76.7 | 55.0 | 83.7 | 87.6 | 91.5 | 94.6 | 97.7 | 97.2 | 74.2 | 86.2 | 83.9 |
| | + TransCLIP-ZS | 79.8$_{+1.7}$ | 79.4$_{+2.7}$ | 52.9$_{-2.2}$ | 85.3$_{+1.6}$ | 85.4$_{-2.2}$ | 92.0$_{+0.5}$ | 95.3$_{+0.7}$ | 99.4$_{+1.7}$ | 97.8$_{+0.6}$ | 72.6$_{-1.5}$ | 88.9$_{+2.6}$ | 84.4$_{-0.6}$ |
| | ProGrad | 78.4 | 78.3 | 55.6 | 88.5 | 88.7 | 90.8 | 94.8 | 98.8 | 97.3 | 73.7 | 87.9 | 84.8 |
| | + TransCLIP-ZS | 79.6$_{+1.2}$ | 80.1$_{+1.8}$ | 54.2$_{-1.4}$ | 90.7$_{+2.2}$ | 87.3$_{-1.4}$ | 91.9$_{+1.1}$ | 95.8$_{+1.0}$ | 99.4$_{+0.6}$ | 97.8$_{+0.5}$ | 75.1$_{+1.3}$ | 90.0$_{+2.1}$ | 85.6$_{+0.8}$ |

Table 15: Cross-Dataset transferability evaluation for five encoders. Few-shot learning methods are trained on 16-shot ImageNet and evaluate on the ten other fine-grained datasets. Average excludes ImageNet.

| | Method | Source ImageNet | SUN397 | Aircraft | EuroSAT | StanfordCars | Food101 | Pets | Flower102 | Caltech101 | DTD | UCF101 | Average |
|---|---|---|---|---|---|---|---|---|---|---|---|---|---|
| **ResNet-50** | CoOp | 63.0 | 56.5 | 13.8 | 22.7 | 53.1 | 73.6 | 84.2 | 56.7 | 85.7 | 34.5 | 56.9 | 53.8 |
| | + TransCLIP-ZS | $63.5_{+0.5}$ | $62.4_{+5.9}$ | $14.2_{-0.4}$ | $38.6_{+16.0}$ | $56.1_{+3.0}$ | $76.2_{+2.6}$ | $84.7_{+0.5}$ | $66.2_{+9.5}$ | $87.4_{+1.7}$ | $38.3_{+3.7}$ | $62.5_{+5.6}$ | $58.7_{+4.9}$ |
| | CoCoOp | 63.2 | 61.5 | 16.5 | 27.1 | 55.9 | 78.1 | 88.2 | 65.5 | 88.6 | 39.6 | 61.1 | 58.2 |
| | + TransCLIP-ZS | $66.5_{+3.2}$ | $63.2_{+1.7}$ | $16.5_{-0.1}$ | $36.0_{+8.9}$ | $57.2_{+1.3}$ | $74.7_{-3.5}$ | $86.1_{-2.1}$ | $70.8_{+5.3}$ | $88.5_{-0.1}$ | $43.3_{+3.7}$ | $65.0_{+3.9}$ | $60.1_{+1.9}$ |
| | ProGrad | 63.4 | 58.4 | 13.5 | 24.2 | 52.6 | 75.9 | 85.9 | 61.8 | 85.9 | 36.1 | 57.6 | 55.2 |
| | + TransCLIP-ZS | $63.5_{+0.2}$ | $63.3_{+4.9}$ | $14.1_{+0.6}$ | $37.2_{+13.0}$ | $55.7_{+3.0}$ | $77.2_{+1.3}$ | $87.5_{+1.6}$ | $70.0_{+8.2}$ | $88.5_{+2.6}$ | $42.1_{+6.0}$ | $62.5_{+4.9}$ | $59.8_{+4.6}$ |
| **ResNet-101** | CoOp | 66.5 | 58.4 | 14.2 | 25.3 | 59.5 | 79.1 | 86.0 | 60.4 | 88.3 | 34.2 | 56.4 | 56.2 |
| | + TransCLIP-ZS | $68.5_{+2.0}$ | $63.7_{+5.3}$ | $15.2_{+1.0}$ | $30.9_{+5.6}$ | $65.2_{+5.7}$ | $81.0_{+1.9}$ | $86.9_{+0.9}$ | $69.7_{+9.4}$ | $90.3_{+1.9}$ | $37.9_{+3.7}$ | $63.7_{+7.3}$ | $60.4_{+4.3}$ |
| | CoCoOp | 65.2 | 62.9 | 17.8 | 25.8 | 62.8 | 81.4 | 87.2 | 64.0 | 91.3 | 39.8 | 61.1 | 59.4 |
| | + TransCLIP-ZS | $73.4_{+8.1}$ | $65.6_{+2.7}$ | $17.8_{-0.1}$ | $45.2_{+19.3}$ | $67.3_{+4.4}$ | $79.9_{-1.5}$ | $87.1_{-0.1}$ | $71.6_{+7.6}$ | $90.9_{-0.4}$ | $40.0_{+0.3}$ | $67.4_{+6.3}$ | $63.3_{+3.9}$ |
| **ViT-B/32** | CoOp | 66.8 | 60.6 | 14.2 | 31.8 | 56.9 | 78.8 | 85.6 | 58.9 | 90.3 | 35.9 | 61.8 | 57.5 |
| | + TransCLIP-ZS | $68.4_{+1.6}$ | $65.7_{+5.0}$ | $14.9_{+0.7}$ | $49.5_{+17.7}$ | $60.4_{+3.5}$ | $80.4_{+1.5}$ | $86.5_{+0.9}$ | $68.0_{+9.0}$ | $92.9_{+2.6}$ | $40.4_{+4.5}$ | $67.6_{+5.8}$ | $62.6_{+5.1}$ |
| | CoCoOp | 66.0 | 64.6 | 17.8 | 40.5 | 59.6 | 80.8 | 88.2 | 65.4 | 92.1 | 42.7 | 64.9 | 61.7 |
| | + TransCLIP-ZS | $71.9_{+5.9}$ | $67.4_{+2.8}$ | $17.8_{+0.0}$ | $54.4_{+13.9}$ | $61.0_{+1.5}$ | $79.0_{-1.9}$ | $85.7_{-2.5}$ | $73.9_{+8.5}$ | $92.4_{+0.3}$ | $47.8_{+5.1}$ | $71.0_{+6.0}$ | $65.0_{+3.4}$ |
| | ProGrad | 66.9 | 61.9 | 13.5 | 33.4 | 56.3 | 79.6 | 86.3 | 60.8 | 91.4 | 38.0 | 62.5 | 58.4 |
| | + TransCLIP-ZS | $68.4_{+1.5}$ | $66.5_{+4.5}$ | $14.2_{+0.7}$ | $51.7_{+18.4}$ | $59.8_{+3.5}$ | $80.8_{+1.3}$ | $86.9_{+0.6}$ | $70.9_{+10.1}$ | $92.5_{+1.0}$ | $42.5_{+4.5}$ | $67.8_{+5.3}$ | $63.4_{+5.0}$ |
| | MaPLE | 65.7 | 65.0 | 18.1 | 41.0 | 60.6 | 80.8 | 88.4 | 65.5 | 91.6 | 42.3 | 63.6 | 61.7 |
| | + TransCLIP-ZS | $71.4_{+5.8}$ | $67.7_{+2.7}$ | $18.5_{+0.4}$ | $54.6_{+13.6}$ | $60.4_{-0.2}$ | $78.7_{-2.2}$ | $85.4_{-3.0}$ | $72.5_{+7.0}$ | $92.2_{+0.6}$ | $46.8_{+4.5}$ | $68.9_{+5.3}$ | $64.6_{+2.9}$ |
| | MaPLE w/ PromptAlign | / | 66.1 | 18.8 | 39.7 | 63.5 | 82.1 | 88.4 | 66.1 | 92.1 | 42.5 | 65.6 | 62.5 |
| **ViT-B/16** | CoOp | 71.9 | 64.9 | 15.7 | 44.6 | 62.1 | 84.3 | 88.3 | 67.1 | 92.7 | 39.5 | 64.1 | 62.0 |
| | + TransCLIP-ZS | $73.3_{+1.4}$ | $67.4_{+5.4}$ | $17.1_{+1.4}$ | $54.5_{+9.9}$ | $66.8_{+4.8}$ | $86.3_{+2.0}$ | $89.4_{+1.1}$ | $74.2_{+7.2}$ | $93.4_{+0.7}$ | $42.1_{+2.6}$ | $69.9_{+5.7}$ | $66.1_{+4.1}$ |
| | CoCoOp | 71.1 | 67.0 | 22.7 | 44.6 | 64.9 | 86.2 | 90.7 | 71.6 | 93.9 | 45.2 | 68.8 | 65.6 |
| | + TransCLIP-ZS | $76.8_{+5.7}$ | $69.6_{+2.7}$ | $22.6_{-0.1}$ | $59.2_{+14.6}$ | $67.0_{+2.1}$ | $85.4_{-0.8}$ | $89.8_{-0.9}$ | $79.0_{+7.4}$ | $94.3_{+0.3}$ | $50.6_{+5.4}$ | $74.5_{+5.7}$ | $69.2_{+3.6}$ |
| | ProGrad | 72.1 | 63.9 | 21.6 | 38.9 | 64.0 | 85.9 | 90.2 | 67.8 | 92.9 | 43.2 | 65.9 | 63.4 |
| | + TransCLIP-ZS | $73.5_{+1.4}$ | $68.6_{+4.7}$ | $22.7_{+1.1}$ | $55.2_{+16.4}$ | $67.9_{+3.8}$ | $87.0_{+1.2}$ | $91.3_{+1.1}$ | $73.9_{+6.1}$ | $94.0_{+1.1}$ | $46.6_{+3.4}$ | $73.5_{+7.6}$ | $68.1_{+4.6}$ |
| | PromptSRC | 71.4 | 67.3 | 24.1 | 45.0 | 65.6 | 86.5 | 90.1 | 70.5 | 93.8 | 46.2 | 68.9 | 65.8 |
| | + TransCLIP-ZS | $76.9_{+5.5}$ | $69.9_{+2.6}$ | $24.9_{+0.8}$ | $59.4_{+14.4}$ | $67.6_{+2.0}$ | $85.3_{-1.2}$ | $89.4_{-0.7}$ | $76.7_{+6.2}$ | $94.2_{+0.4}$ | $51.1_{+5.0}$ | $76.0_{+7.0}$ | $69.4_{+3.7}$ |
| | MaPLE | 70.5 | 67.3 | 24.4 | 45.8 | 65.7 | 86.4 | 90.4 | 72.0 | 93.7 | 46.3 | 68.7 | 66.1 |
| | + TransCLIP-ZS | $76.6_{+6.1}$ | $69.8_{+2.5}$ | $24.5_{+0.2}$ | $59.5_{+13.7}$ | $66.8_{+1.2}$ | $85.4_{-1.0}$ | $89.7_{-0.7}$ | $78.0_{+6.0}$ | $94.3_{+0.6}$ | $49.4_{+3.1}$ | $74.4_{+5.6}$ | $69.2_{+3.1}$ |
| | Maple w/ PromptAlign | / | 67.5 | 24.8 | 47.9 | 68.5 | 86.7 | 90.8 | 72.4 | 94.0 | 47.2 | 69.5 | 66.9 |
| **ViT-L/14** | CoOp | 78.2 | 64.9 | 21.6 | 51.4 | 75.5 | 89.3 | 91.0 | 68.9 | 93.6 | 43.6 | 68.8 | 66.9 |
| | + TransCLIP-ZS | $79.5_{+1.4}$ | $70.6_{+5.7}$ | $24.3_{+2.8}$ | $72.7_{+21.3}$ | $79.0_{+3.4}$ | $91.1_{+1.8}$ | $93.6_{+2.6}$ | $78.1_{+9.2}$ | $96.2_{+2.5}$ | $48.2_{+4.6}$ | $75.3_{+6.5}$ | $72.9_{+6.0}$ |
| | CoCoOp | 77.8 | 70.8 | 31.0 | 47.4 | 77.9 | 91.4 | 94.1 | 76.2 | 97.1 | 50.7 | 74.1 | 71.1 |
| | + TransCLIP-ZS | $81.9_{+4.1}$ | $73.8_{+3.0}$ | $33.2_{+2.1}$ | $76.3_{+28.9}$ | $78.7_{+0.8}$ | $90.6_{-0.8}$ | $94.4_{+0.3}$ | $81.4_{+5.1}$ | $97.1_{+0.1}$ | $55.5_{+4.7}$ | $79.2_{+5.1}$ | $76.0_{+4.9}$ |
| | ProGrad | 78.4 | 66.9 | 24.8 | 45.4 | 75.9 | 90.4 | 93.1 | 73.4 | 95.3 | 45.8 | 71.8 | 68.3 |
| | + TransCLIP-ZS | $79.6_{+1.2}$ | $72.4_{+5.5}$ | $26.8_{+2.0}$ | $67.2_{+21.7}$ | $78.7_{+2.8}$ | $91.6_{+1.2}$ | $95.6_{+2.5}$ | $79.4_{+6.0}$ | $96.0_{+1.3}$ | $51.9_{+6.0}$ | $78.4_{+6.6}$ | $73.8_{+5.6}$ |
| | MaPLE | 77.2 | 71.6 | 30.2 | 55.7 | 77.3 | 91.3 | 93.1 | 76.7 | 96.2 | 53.8 | 74.9 | 72.1 |
| | + TransCLIP-ZS | $81.6_{+4.4}$ | $74.1_{+2.5}$ | $32.8_{+2.6}$ | $75.2_{+19.6}$ | $78.3_{+1.0}$ | $90.5_{-0.8}$ | $94.2_{+1.1}$ | $83.0_{+6.3}$ | $97.4_{+1.2}$ | $56.2_{+2.4}$ | $81.0_{+6.1}$ | $76.3_{+4.2}$ |

Table 16: Domain Generalization evaluation for five encoders with manual prompting strategies.

| | Method | Source ImageNet | Adversarial | ImageNetV2 | Rendition | Sketch | Average | Average OOD |
|---|---|---|---|---|---|---|---|---|
| **ResNet-50** | w/ a photo of a | 58.0 | 22.0 | 51.2 | 56.1 | 33.3 | 44.1 | 40.7 |
| | + TransCLIP-ZS | $60.8_{+2.8}$ | $21.5_{-0.4}$ | $51.4_{+0.1}$ | $52.8_{-3.3}$ | $35.1_{+1.8}$ | $44.3_{+0.2}$ | $40.2_{-0.5}$ |
| | w/ custom templates | 60.3 | 23.8 | 53.4 | 60.5 | 35.5 | 46.7 | 43.3 |
| | + TransCLIP-ZS | $61.7_{+1.4}$ | $23.4_{-0.5}$ | $52.6_{-0.8}$ | $56.4_{-4.2}$ | $36.6_{+1.1}$ | $46.1_{-0.6}$ | $42.2_{-1.1}$ |
| **ResNet-101** | w/ a photo of a | 60.6 | 28.2 | 54.3 | 64.2 | 38.0 | 49.1 | 46.2 |
| | + TransCLIP-ZS | $64.8_{+4.2}$ | $29.2_{+1.0}$ | $56.2_{+1.9}$ | $65.1_{+1.0}$ | $42.2_{+4.3}$ | $51.5_{+2.5}$ | $48.2_{+2.0}$ |
| | w/ custom templates | 62.5 | 29.8 | 56.1 | 67.7 | 40.6 | 51.4 | 48.6 |
| | + TransCLIP-ZS | $65.6_{+3.0}$ | $30.6_{+0.8}$ | $57.0_{+0.9}$ | $68.2_{+0.5}$ | $44.0_{+3.4}$ | $53.1_{+1.7}$ | $49.9_{+1.4}$ |
| **ViT-B/32** | w/ a photo of a | 61.9 | 29.9 | 54.7 | 66.8 | 40.8 | 50.8 | 48.1 |
| | + TransCLIP-ZS | $64.9_{+3.0}$ | $30.5_{+0.6}$ | $55.7_{+1.1}$ | $67.0_{+0.2}$ | $43.6_{+2.8}$ | $52.4_{+1.5}$ | $49.2_{+1.2}$ |
| | w/ custom templates | 63.8 | 32.1 | 56.3 | 69.5 | 42.1 | 52.8 | 50.0 |
| | + TransCLIP-ZS | $66.2_{+2.5}$ | $32.4_{+0.3}$ | $56.6_{+0.2}$ | $69.2_{-0.3}$ | $44.3_{+2.1}$ | $53.7_{+1.0}$ | $50.6_{+0.6}$ |
| **ViT-B/16** | w/ a photo of a | 66.6 | 47.9 | 60.6 | 73.8 | 46.0 | 59.0 | 57.1 |
| | + TransCLIP-ZS | $70.3_{+3.7}$ | $49.5_{+1.7}$ | $62.3_{+1.7}$ | $75.0_{+1.3}$ | $49.7_{+3.7}$ | $61.4_{+2.4}$ | $59.1_{+2.1}$ |
| | w/ custom templates | 68.8 | 50.6 | 62.3 | 77.8 | 48.4 | 61.6 | 59.8 |
| | + TransCLIP-ZS | $71.5_{+2.7}$ | $52.1_{+1.4}$ | $63.4_{+1.1}$ | $78.1_{+0.2}$ | $51.1_{+2.7}$ | $63.2_{+1.6}$ | $61.2_{+1.4}$ |
| **ViT-L/14** | w/ a photo of a | 72.9 | 68.4 | 67.2 | 85.3 | 57.4 | 70.2 | 69.6 |
| | + TransCLIP-ZS | $77.2_{+4.3}$ | $71.4_{+3.0}$ | $69.1_{+1.8}$ | $87.1_{+1.8}$ | $60.0_{+2.6}$ | $72.9_{+2.7}$ | $71.9_{+2.3}$ |
| | w/ custom templates | 75.9 | 70.9 | 70.2 | 87.8 | 59.7 | 72.9 | 72.2 |
| | + TransCLIP-ZS | $78.6_{+2.7}$ | $73.6_{+2.7}$ | $70.8_{+0.5}$ | $89.0_{+1.1}$ | $61.9_{+2.2}$ | $74.8_{+1.8}$ | $73.8_{+1.6}$ |

Table 17: Domain Generalization evaluation for five encoders. Few-shot learning methods are trained on 16-shot ImageNet and evaluated on the 4 other variants.

| | Method | Source ImageNet | Target Adversarial | Target ImageNetV2 | Target Rendition | Target Sketch | Average | Average OOD |
|---|---|---|---|---|---|---|---|---|
| **ResNet-50** | CoOp | 63.0 | 22.0 | 55.0 | 55.0 | 32.8 | 45.5 | 41.2 |
| | + TransCLIP-ZS | $63.5_{+0.5}$ | $21.0_{-1.0}$ | $53.6_{-1.4}$ | $52.3_{-2.7}$ | $34.8_{+2.0}$ | $45.0_{-0.5}$ | $40.4_{-0.8}$ |
| | TaskRes | 64.6 | 22.9 | 56.4 | 60.8 | 35.9 | 48.1 | 44.0 |
| | + TransCLIP-ZS | $64.4_{-0.2}$ | $21.7_{-1.2}$ | $54.8_{-1.6}$ | $56.2_{-4.6}$ | $36.9_{+1.0}$ | $46.8_{-1.3}$ | $42.4_{-1.6}$ |
| **ResNet-101** | CoOp | 66.5 | 29.5 | 58.3 | 63.6 | 39.0 | 51.4 | 47.6 |
| | + TransCLIP-ZS | $68.5_{+2.0}$ | $29.9_{+0.5}$ | $58.6_{+0.2}$ | $64.8_{+1.2}$ | $42.3_{+3.3}$ | $52.8_{+1.4}$ | $48.9_{+1.3}$ |
| | TaskRes | 67.6 | 30.0 | 59.6 | 68.4 | 41.8 | 53.5 | 49.9 |
| | + TransCLIP-ZS | $69.3_{+1.7}$ | $30.2_{+0.2}$ | $59.3_{-0.4}$ | $68.8_{+0.4}$ | $44.6_{+2.9}$ | $54.4_{+1.0}$ | $50.7_{+0.8}$ |
| **ViT-B/32** | CoOp | 66.8 | 31.2 | 58.5 | 65.2 | 40.1 | 52.3 | 48.7 |
| | + TransCLIP-ZS | $68.4_{+1.6}$ | $31.3_{+0.1}$ | $58.3_{-0.2}$ | $65.5_{+0.3}$ | $42.7_{+2.6}$ | $53.2_{+0.9}$ | $49.4_{+0.7}$ |
| | TaskRes | 68.2 | 31.3 | 59.3 | 69.5 | 42.5 | 54.2 | 50.6 |
| | + TransCLIP-ZS | $69.2_{+1.1}$ | $31.3_{+0.1}$ | $59.1_{-0.2}$ | $69.3_{-0.3}$ | $44.9_{+2.4}$ | $54.8_{+0.6}$ | $51.2_{+0.5}$ |
| **ViT-B/16** | CoOp | 71.9 | 49.4 | 64.1 | 75.1 | 47.1 | 61.5 | 58.9 |
| | + TransCLIP-ZS | $73.3_{+1.4}$ | $50.8_{+1.3}$ | $64.6_{+0.4}$ | $75.7_{+0.7}$ | $50.3_{+3.2}$ | $62.9_{+1.4}$ | $60.4_{+1.4}$ |
| | TaskRes | 73.0 | 50.3 | 65.6 | 77.8 | 49.2 | 63.2 | 60.7 |
| | + TransCLIP-ZS | $74.1_{+1.0}$ | $51.9_{+1.6}$ | $65.4_{-0.2}$ | $78.4_{+0.6}$ | $51.6_{+2.4}$ | $64.3_{+1.1}$ | $61.8_{+1.1}$ |
| **ViT-L/14** | CoOp | 78.2 | 69.4 | 70.8 | 85.4 | 57.5 | 72.3 | 70.8 |
| | + TransCLIP-ZS | $79.5_{+1.3}$ | $71.9_{+2.6}$ | $71.1_{+0.3}$ | $86.9_{+1.5}$ | $60.0_{+2.5}$ | $73.9_{+1.6}$ | $72.5_{+1.7}$ |
| | TaskRes | 78.1 | 71.3 | 71.6 | 87.9 | 60.1 | 73.8 | 72.7 |
| | + TransCLIP-ZS | $79.8_{+1.7}$ | $74.2_{+3.0}$ | $71.8_{+0.2}$ | $88.9_{+1.1}$ | $62.0_{+1.9}$ | $75.4_{+1.6}$ | $74.2_{+1.5}$ |

Table 18: Detailed results of transductive methods in the few-shot setting for the 11 datasets with ResNet-50 as visual backbone.

| Shots | Method | ImageNet | SUN397 | Aircraft | EuroSAT | StanfordCars | Food101 | Pets | Flowers102 | Caltech101 | DTD | UCF101 | Average |
|---|---|---|---|---|---|---|---|---|---|---|---|---|---|
| 1 | TF | 20.6 | 31.2 | 13.1 | 39.0 | 21.8 | 28.3 | 27.2 | 53.6 | 66.1 | 27.7 | 38.1 | 33.3 |
| | BD-CSPN | 24.7 | 36.9 | 13.9 | 40.3 | 27.2 | 34.1 | 34.1 | 66.7 | 74.3 | 32.8 | 43.4 | 38.9 |
| | LaplacianShot | 23.8 | 35.5 | 14.0 | 42.3 | 27.0 | 34.7 | 37.3 | 66.6 | 72.4 | 32.8 | 43.2 | 39.1 |
| | PT-MAP | 29.4 | 42.9 | 15.7 | 48.0 | 33.8 | 44.8 | 56.5 | 61.4 | 46.9 | 38.6 | 52.2 | 42.7 |
| | TIM | 26.1 | 40.0 | 13.4 | 42.5 | 27.3 | 41.4 | 35.0 | 69.1 | 62.3 | 31.7 | 46.9 | 39.6 |
| | CoOp + UPL | 59.6 | 63.4 | 17.5 | 54.7 | 56.4 | 75.3 | 82.8 | 73.5 | 87.4 | 48.3 | 66.1 | 62.3 |
| | TransCLIP-FS | 55.7 | 63.5 | 20.6 | 70.3 | 56.2 | 77.2 | 86.9 | 83.7 | 87.4 | 51.3 | 70.7 | 65.8 |
| 2 | TF | 29.6 | 43.1 | 16.6 | 57.2 | 32.3 | 41.4 | 40.1 | 68.4 | 77.5 | 41.4 | 51.3 | 45.4 |
| | BD-CSPN | 33.2 | 48.1 | 17.8 | 58.6 | 36.2 | 47.4 | 50.0 | 77.0 | 80.7 | 43.2 | 54.1 | 49.7 |
| | LaplacianShot | 33.1 | 47.8 | 17.7 | 60.0 | 36.1 | 48.7 | 50.4 | 77.5 | 81.0 | 43.3 | 55.2 | 50.1 |
| | PT-MAP | 39.3 | 54.6 | 19.3 | 61.4 | 43.5 | 60.1 | 67.0 | 68.9 | 51.5 | 50.4 | 61.9 | 52.5 |
| | TIM | 35.5 | 52.2 | 18.2 | 60.2 | 38.1 | 57.2 | 51.7 | 79.7 | 76.1 | 44.2 | 59.6 | 52.1 |
| | CoOp + UPL | 59.8 | 64.0 | 19.3 | 62.9 | 59.2 | 74.8 | 81.2 | 80.5 | 88.1 | 49.5 | 68.0 | 64.3 |
| | TransCLIP-FS | 59.3 | 66.2 | 20.3 | 71.5 | 58.7 | 77.2 | 86.0 | 87.1 | 87.8 | 55.2 | 72.8 | 67.5 |
| 4 | TF | 38.5 | 53.1 | 20.4 | 64.9 | 42.8 | 52.5 | 49.3 | 80.7 | 83.6 | 48.4 | 59.3 | 54.0 |
| | BD-CSPN | 40.7 | 54.9 | 20.2 | 65.4 | 43.4 | 56.6 | 54.3 | 83.7 | 84.0 | 48.1 | 59.8 | 55.6 |
| | LaplacianShot | 40.5 | 54.9 | 19.7 | 68.0 | 43.3 | 58.0 | 55.5 | 84.2 | 83.9 | 47.9 | 60.1 | 56.0 |
| | PT-MAP | 46.8 | 61.4 | 22.8 | 69.5 | 50.7 | 66.6 | 70.0 | 71.0 | 54.6 | 56.3 | 68.0 | 58.0 |
| | TIM | 43.3 | 59.1 | 22.9 | 71.0 | 49.6 | 64.0 | 58.8 | 87.6 | 79.1 | 53.2 | 65.8 | 59.5 |
| | CoOp + UPL | 60.3 | 65.7 | 23.3 | 71.0 | 63.0 | 75.8 | 83.6 | 87.3 | 88.0 | 55.2 | 69.1 | 67.5 |
| | TransCLIP-FS | 59.3 | 66.5 | 25.0 | 73.8 | 61.4 | 76.6 | 81.6 | 88.4 | 88.2 | 57.6 | 73.3 | 68.4 |
| 8 | TF | 45.1 | 59.7 | 24.1 | 66.8 | 51.2 | 61.1 | 61.7 | 86.4 | 86.3 | 55.9 | 65.1 | 60.3 |
| | BD-CSPN | 45.6 | 59.6 | 22.9 | 66.2 | 50.4 | 62.4 | 65.7 | 87.5 | 85.5 | 54.6 | 65.1 | 60.5 |
| | LaplacianShot | 45.2 | 59.1 | 22.4 | 69.1 | 49.6 | 63.4 | 65.7 | 87.6 | 85.8 | 53.9 | 65.9 | 60.7 |
| | PT-MAP | 50.6 | 64.2 | 23.4 | 66.7 | 55.9 | 69.6 | 76.9 | 72.9 | 54.8 | 60.4 | 70.6 | 60.5 |
| | TIM | 49.9 | 63.4 | 25.0 | 69.5 | 59.7 | 70.0 | 71.8 | 89.9 | 82.9 | 59.1 | 70.8 | 64.7 |
| | CoOp + UPL | 60.9 | 67.0 | 26.0 | 71.7 | 66.5 | 75.5 | 82.7 | 91.2 | 88.3 | 59.0 | 71.4 | 69.1 |
| | TransCLIP-FS | 59.9 | 68.3 | 28.0 | 74.5 | 67.6 | 76.9 | 86.6 | 90.4 | 88.7 | 62.1 | 76.1 | 70.8 |
| 16 | TF | 50.0 | 63.2 | 26.6 | 71.8 | 57.7 | 66.1 | 66.4 | 90.3 | 87.3 | 58.8 | 67.7 | 64.2 |
| | BD-CSPN | 49.7 | 62.4 | 25.5 | 71.3 | 56.6 | 66.0 | 66.2 | 89.6 | 86.7 | 57.8 | 67.2 | 63.5 |
| | LaplacianShot | 48.9 | 61.5 | 24.6 | 71.5 | 54.8 | 66.7 | 67.5 | 89.5 | 86.4 | 56.2 | 67.5 | 63.2 |
| | PT-MAP | 54.1 | 66.1 | 25.6 | 68.1 | 61.1 | 70.6 | 79.0 | 75.2 | 57.0 | 62.4 | 71.0 | 62.7 |
| | TIM | 55.5 | 66.8 | 30.8 | 81.6 | 68.0 | 72.4 | 75.0 | 88.9 | 85.7 | 63.1 | 74.4 | 69.3 |
| | CoOp + UPL | 60.9 | 69.4 | 31.6 | 78.0 | 71.4 | 76.2 | 83.5 | 93.6 | 89.1 | 62.8 | 73.5 | 71.8 |
| | TransCLIP-FS | 62.6 | 70.4 | 30.3 | 77.6 | 71.5 | 77.1 | 87.3 | 92.5 | 88.7 | 64.4 | 77.7 | 72.7 |

Table 19: Detailed results of transductive methods in the few-shot setting for the 11 datasets with ResNet-101 as visual backbone.

| Shots | Method | ImageNet | SUN397 | Aircraft | EuroSAT | StanfordCars | Food101 | Pets | Flowers102 | Caltech101 | DTD | UCF101 | Average |
|---|---|---|---|---|---|---|---|---|---|---|---|---|---|
| 1 | TF | 24.9 | 33.3 | 16.0 | 38.5 | 29.4 | 34.3 | 37.1 | 57.0 | 71.6 | 29.7 | 43.5 | 37.8 |
| | BD-CSPN | 29.9 | 40.2 | 16.8 | 39.5 | 35.1 | 42.6 | 51.0 | 70.0 | 79.6 | 32.1 | 51.8 | 44.4 |
| | LaplacianShot | 30.0 | 40.0 | 17.1 | 40.6 | 37.2 | 43.8 | 51.8 | 71.7 | 79.4 | 34.9 | 52.1 | 45.3 |
| | PT-MAP | 34.3 | 46.2 | 18.1 | 49.3 | 44.0 | 53.0 | 69.5 | 65.0 | 51.6 | 39.1 | 58.9 | 48.1 |
| | TIM | 31.5 | 44.2 | 16.6 | 42.9 | 39.0 | 54.9 | 51.8 | 77.6 | 66.5 | 36.1 | 56.2 | 47.0 |
| | CoOp + UPL | 62.7 | 64.5 | 20.8 | 63.6 | 61.7 | 77.8 | 83.8 | 72.8 | 89.6 | 47.0 | 69.1 | 64.9 |
| | TransCLIP-FS | 64.3 | 66.6 | 19.6 | 67.2 | 70.0 | 82.9 | 91.5 | 80.4 | 91.2 | 47.0 | 70.1 | 68.3 |
| 2 | TF | 34.8 | 46.6 | 19.6 | 53.7 | 41.2 | 49.1 | 51.1 | 73.8 | 83.1 | 42.3 | 56.3 | 50.1 |
| | BD-CSPN | 39.9 | 51.7 | 20.3 | 54.2 | 46.7 | 57.7 | 60.4 | 80.8 | 85.5 | 45.5 | 59.4 | 54.7 |
| | LaplacianShot | 39.9 | 51.8 | 20.9 | 59.3 | 46.9 | 59.0 | 63.2 | 81.9 | 85.9 | 45.5 | 59.8 | 55.8 |
| | PT-MAP | 44.3 | 57.4 | 21.8 | 62.0 | 52.9 | 65.7 | 76.6 | 71.0 | 56.2 | 52.5 | 65.8 | 56.9 |
| | TIM | 42.4 | 55.6 | 19.9 | 63.5 | 50.2 | 69.2 | 67.3 | 85.5 | 81.5 | 49.0 | 62.6 | 58.8 |
| | CoOp + UPL | 63.0 | 65.4 | 23.6 | 66.4 | 66.6 | 77.8 | 85.2 | 81.2 | 89.4 | 51.4 | 70.9 | 67.4 |
| | TransCLIP-FS | 64.6 | 67.2 | 22.7 | 68.3 | 70.7 | 80.8 | 89.1 | 85.2 | 91.5 | 49.8 | 72.8 | 69.3 |
| 4 | TF | 44.9 | 56.9 | 23.7 | 62.8 | 53.4 | 61.6 | 61.1 | 83.7 | 87.5 | 51.5 | 65.4 | 59.3 |
| | BD-CSPN | 47.8 | 58.8 | 23.7 | 62.1 | 54.4 | 66.0 | 70.1 | 86.1 | 87.7 | 51.2 | 65.4 | 61.2 |
| | LaplacianShot | 47.7 | 58.9 | 23.4 | 71.9 | 54.3 | 67.3 | 70.9 | 86.8 | 87.7 | 51.1 | 65.8 | 62.3 |
| | PT-MAP | 51.7 | 63.8 | 25.5 | 68.0 | 60.3 | 71.6 | 79.9 | 74.6 | 56.4 | 57.4 | 71.0 | 61.8 |
| | TIM | 51.2 | 63.2 | 25.1 | 73.6 | 61.4 | 75.8 | 76.8 | 87.0 | 87.8 | 55.3 | 71.7 | 66.3 |
| | CoOp + UPL | 63.9 | 67.4 | 25.4 | 70.8 | 69.3 | 79.5 | 85.5 | 87.4 | 90.3 | 55.6 | 73.2 | 69.2 |
| | TransCLIP-FS | 65.1 | 68.7 | 26.2 | 73.7 | 71.6 | 81.3 | 90.1 | 88.6 | 91.7 | 56.4 | 73.2 | 71.5 |
| 8 | TF | 51.5 | 62.9 | 27.1 | 63.3 | 61.5 | 69.0 | 72.3 | 89.1 | 89.7 | 58.2 | 70.2 | 65.0 |
| | BD-CSPN | 52.7 | 63.1 | 27.3 | 62.7 | 61.0 | 70.9 | 76.8 | 89.5 | 89.4 | 57.0 | 70.3 | 65.5 |
| | LaplacianShot | 52.3 | 62.8 | 26.8 | 68.4 | 60.7 | 71.7 | 77.3 | 89.6 | 89.2 | 56.0 | 70.3 | 65.9 |
| | PT-MAP | 55.5 | 66.5 | 28.1 | 67.0 | 64.6 | 73.7 | 84.6 | 76.6 | 59.4 | 61.1 | 72.2 | 64.5 |
| | TIM | 56.6 | 67.3 | 28.1 | 74.3 | 70.0 | 77.0 | 85.3 | 91.5 | 88.6 | 60.5 | 71.7 | 70.1 |
| | CoOp + UPL | 64.6 | 69.0 | 28.3 | 77.9 | 73.5 | 79.5 | 85.8 | 92.1 | 90.7 | 61.2 | 75.8 | 72.6 |
| | TransCLIP-FS | 65.0 | 69.6 | 27.9 | 71.2 | 74.4 | 81.5 | 90.3 | 89.0 | 91.7 | 61.7 | 76.1 | 72.6 |
| 16 | TF | 56.3 | 66.8 | 30.7 | 68.0 | 68.0 | 73.6 | 76.3 | 92.0 | 90.9 | 61.9 | 72.6 | 68.8 |
| | BD-CSPN | 56.4 | 66.1 | 30.8 | 66.0 | 67.2 | 73.4 | 76.4 | 91.8 | 90.8 | 60.5 | 72.4 | 68.3 |
| | LaplacianShot | 56.0 | 65.5 | 29.4 | 71.2 | 65.8 | 74.4 | 78.6 | 91.7 | 90.2 | 58.8 | 72.3 | 68.5 |
| | PT-MAP | 58.6 | 68.3 | 30.9 | 69.5 | 69.2 | 75.3 | 85.3 | 78.2 | 61.5 | 62.9 | 73.4 | 66.6 |
| | TIM | 61.4 | 70.6 | 34.6 | 79.2 | 75.8 | 78.8 | 84.4 | 91.8 | 88.9 | 67.2 | 76.4 | 73.6 |
| | CoOp + UPL | 64.6 | 71.1 | 34.9 | 82.1 | 77.6 | 79.5 | 85.7 | 94.0 | 92.1 | 65.2 | 77.1 | 74.9 |
| | TransCLIP-FS | 66.4 | 71.1 | 28.4 | 73.8 | 77.1 | 81.6 | 90.6 | 90.8 | 92.3 | 61.5 | 76.8 | 73.7 |

Table 20: Detailed results of transductive methods in the few-shot setting for the 11 datasets with ViT-B/32 as visual backbone.

| Shots | Method | ImageNet | SUN397 | Aircraft | EuroSAT | StanfordCars | Food101 | Pets | Flowers102 | Caltech101 | DTD | UCF101 | Average |
|---|---|---|---|---|---|---|---|---|---|---|---|---|---|
| 1 | TF | 25.1 | 36.1 | 14.6 | 44.4 | 26.7 | 34.4 | 33.3 | 60.0 | 74.4 | 29.0 | 46.4 | 38.6 |
| | BD-CSPN | 30.1 | 42.9 | 16.2 | 45.7 | 33.8 | 41.2 | 43.9 | 73.1 | 80.2 | 30.8 | 52.6 | 44.6 |
| | LaplacianShot | 29.2 | 41.7 | 16.1 | 48.6 | 33.2 | 43.1 | 43.8 | 73.3 | 80.6 | 32.7 | 52.9 | 45.0 |
| | PT-MAP | 33.1 | 48.8 | 17.0 | 54.8 | 38.6 | 49.8 | 50.9 | 62.4 | 52.5 | 37.9 | 57.0 | 45.7 |
| | TIM | 31.5 | 47.6 | 16.6 | 55.2 | 36.4 | 51.4 | 48.4 | 76.8 | 71.5 | 35.6 | 57.6 | 48.1 |
| | CoOp + UPL | 63.0 | 66.2 | 21.0 | 64.0 | 58.1 | 78.8 | 84.0 | 74.4 | 89.7 | 52.0 | 68.3 | 65.4 |
| | TransCLIP-FS | 64.3 | 68.9 | 22.7 | 63.5 | 63.7 | 82.2 | 90.1 | 83.2 | 92.2 | 52.3 | 69.5 | 68.4 |
| 2 | TF | 34.7 | 49.5 | 19.3 | 56.5 | 37.4 | 48.7 | 47.4 | 75.1 | 83.9 | 44.5 | 57.7 | 50.4 |
| | BD-CSPN | 39.2 | 53.1 | 20.7 | 57.2 | 42.1 | 55.5 | 55.2 | 82.4 | 86.8 | 45.6 | 61.6 | 54.5 |
| | LaplacianShot | 39.1 | 53.9 | 20.4 | 58.3 | 42.4 | 57.7 | 57.3 | 82.5 | 86.7 | 45.9 | 62.6 | 55.2 |
| | PT-MAP | 42.6 | 60.1 | 22.3 | 63.7 | 46.0 | 63.9 | 64.0 | 69.5 | 55.6 | 50.4 | 66.8 | 55.0 |
| | TIM | 41.1 | 59.0 | 21.1 | 68.9 | 44.1 | 66.2 | 60.1 | 86.5 | 81.5 | 48.6 | 68.1 | 58.7 |
| | CoOp + UPL | 63.4 | 66.6 | 22.8 | 71.9 | 60.8 | 78.5 | 85.0 | 81.0 | 90.1 | 53.5 | 70.2 | 67.6 |
| | TransCLIP-FS | 64.8 | 69.5 | 22.9 | 76.9 | 63.8 | 81.2 | 89.9 | 85.4 | 92.1 | 52.9 | 71.0 | 70.0 |
| 4 | TF | 44.5 | 59.4 | 23.2 | 62.1 | 48.6 | 60.8 | 57.9 | 85.2 | 89.1 | 52.6 | 65.2 | 59.0 |
| | BD-CSPN | 47.0 | 61.1 | 23.4 | 64.2 | 49.1 | 65.3 | 64.8 | 87.2 | 89.4 | 52.0 | 67.0 | 61.0 |
| | LaplacianShot | 46.8 | 61.1 | 23.6 | 68.4 | 49.2 | 65.6 | 66.6 | 87.6 | 89.3 | 51.4 | 67.5 | 61.6 |
| | PT-MAP | 50.1 | 65.5 | 24.1 | 68.9 | 52.3 | 70.3 | 69.0 | 73.3 | 57.3 | 56.1 | 70.1 | 59.7 |
| | TIM | 50.4 | 65.0 | 24.7 | 70.0 | 56.1 | 73.0 | 74.4 | 90.5 | 88.7 | 55.9 | 71.8 | 65.5 |
| | CoOp + UPL | 63.9 | 68.8 | 26.6 | 72.6 | 63.7 | 78.2 | 85.2 | 88.8 | 90.1 | 55.4 | 73.1 | 69.7 |
| | TransCLIP-FS | 64.7 | 70.1 | 26.4 | 78.0 | 66.5 | 80.3 | 87.2 | 88.7 | 92.2 | 58.0 | 74.3 | 71.5 |
| 8 | TF | 50.9 | 64.7 | 27.1 | 67.6 | 57.1 | 68.5 | 68.0 | 89.4 | 90.5 | 58.2 | 70.7 | 64.8 |
| | BD-CSPN | 51.2 | 64.8 | 27.4 | 66.5 | 56.9 | 69.6 | 71.7 | 90.0 | 89.6 | 56.3 | 71.0 | 65.0 |
| | LaplacianShot | 51.0 | 64.3 | 26.4 | 70.0 | 55.9 | 70.4 | 73.7 | 90.2 | 90.1 | 55.4 | 70.8 | 65.3 |
| | PT-MAP | 53.7 | 68.3 | 27.4 | 70.9 | 58.5 | 72.8 | 75.4 | 75.2 | 59.7 | 59.4 | 71.5 | 63.0 |
| | TIM | 56.2 | 69.0 | 28.4 | 75.8 | 65.1 | 76.1 | 79.6 | 92.3 | 87.4 | 63.3 | 75.4 | 69.9 |
| | CoOp + UPL | 64.8 | 69.7 | 30.0 | 79.6 | 68.9 | 79.3 | 85.5 | 91.6 | 91.8 | 62.1 | 73.9 | 72.5 |
| | TransCLIP-FS | 65.5 | 71.3 | 28.0 | 78.2 | 70.8 | 81.0 | 89.4 | 90.0 | 92.3 | 61.1 | 77.0 | 73.2 |
| 16 | TF | 55.6 | 68.0 | 29.7 | 69.7 | 62.9 | 72.6 | 73.7 | 92.0 | 91.6 | 61.6 | 73.1 | 68.2 |
| | BD-CSPN | 55.3 | 67.5 | 29.8 | 69.5 | 62.3 | 72.9 | 74.2 | 91.9 | 91.7 | 59.6 | 73.3 | 68.0 |
| | LaplacianShot | 54.8 | 66.7 | 28.4 | 71.2 | 60.9 | 73.2 | 75.3 | 91.3 | 91.3 | 58.3 | 72.9 | 67.7 |
| | PT-MAP | 56.9 | 69.9 | 29.2 | 71.3 | 63.1 | 74.1 | 78.7 | 77.1 | 60.7 | 61.9 | 72.9 | 65.1 |
| | TIM | 60.5 | 71.8 | 33.0 | 79.4 | 72.2 | 78.1 | 85.0 | 92.8 | 88.4 | 64.6 | 78.1 | 73.3 |
| | CoOp + UPL | 64.8 | 71.9 | 34.1 | 84.3 | 73.6 | 79.0 | 85.8 | 94.2 | 92.4 | 64.8 | 78.3 | 74.8 |
| | TransCLIP-FS | 66.6 | 72.6 | 30.1 | 78.9 | 73.2 | 81.1 | 89.5 | 90.9 | 94.4 | 62.7 | 77.2 | 74.3 |

Table 21: Detailed results of transductive methods in the few-shot setting for the 11 datasets with ViT-B/16 as visual backbone.

| Shots | Method | ImageNet | SUN397 | Aircraft | EuroSAT | StanfordCars | Food101 | Pets | Flowers102 | Caltech101 | DTD | UCF101 | Average |
|---|---|---|---|---|---|---|---|---|---|---|---|---|---|
| 1 | TF | 29.7 | 38.1 | 19.2 | 46.0 | 32.5 | 43.5 | 38.2 | 67.8 | 75.5 | 31.6 | 48.8 | 42.8 |
| | BD-CSPN | 35.4 | 45.7 | 22.0 | 45.7 | 42.0 | 54.2 | 52.9 | 82.9 | 83.5 | 34.7 | 58.0 | 50.6 |
| | LaplacianShot | 34.9 | 44.5 | 22.1 | 52.1 | 41.1 | 53.0 | 52.2 | 83.1 | 83.4 | 35.8 | 57.3 | 50.9 |
| | PT-MAP | 40.1 | 52.6 | 23.8 | 59.7 | 48.4 | 64.4 | 61.8 | 69.4 | 54.1 | 41.8 | 63.5 | 52.7 |
| | TIM | 37.5 | 48.3 | 22.8 | 48.2 | 44.8 | 65.7 | 53.9 | 86.4 | 75.1 | 35.8 | 62.7 | 52.8 |
| | CoOp + UPL | 68.8 | 68.5 | 27.2 | 70.0 | 68.9 | 83.6 | 90.6 | 81.7 | 92.7 | 51.3 | 73.1 | 70.6 |
| | TransCLIP-FS | 69.8 | 70.6 | 29.9 | 72.5 | 70.9 | 87.9 | 93.8 | 84.8 | 93.1 | 53.3 | 78.4 | 73.2 |
| 2 | TF | 40.5 | 51.6 | 25.3 | 63.1 | 45.1 | 58.8 | 54.8 | 83.2 | 87.0 | 47.3 | 59.4 | 56.0 |
| | BD-CSPN | 46.1 | 56.1 | 26.7 | 64.7 | 50.7 | 67.5 | 64.6 | 89.6 | 89.6 | 48.9 | 64.0 | 60.8 |
| | LaplacianShot | 45.8 | 55.9 | 27.1 | 68.2 | 51.1 | 68.2 | 66.0 | 89.7 | 89.6 | 48.9 | 65.1 | 61.4 |
| | PT-MAP | 50.7 | 63.1 | 28.6 | 71.7 | 57.5 | 77.5 | 75.7 | 73.9 | 59.1 | 53.8 | 68.7 | 61.9 |
| | TIM | 47.9 | 60.7 | 28.1 | 75.8 | 55.7 | 78.7 | 70.6 | 91.4 | 86.6 | 52.3 | 66.4 | 64.9 |
| | CoOp + UPL | 69.2 | 69.2 | 30.1 | 73.4 | 71.0 | 83.8 | 88.4 | 87.9 | 93.3 | 53.9 | 75.8 | 72.4 |
| | TransCLIP-FS | 70.3 | 70.9 | 30.0 | 77.1 | 71.7 | 87.0 | 91.7 | 90.6 | 93.5 | 55.1 | 78.5 | 74.2 |
| 4 | TF | 51.1 | 61.0 | 30.3 | 64.9 | 56.8 | 71.0 | 65.9 | 90.9 | 91.5 | 53.7 | 67.9 | 64.1 |
| | BD-CSPN | 53.8 | 62.5 | 30.5 | 64.8 | 58.5 | 75.3 | 72.0 | 92.5 | 92.0 | 52.1 | 70.9 | 65.9 |
| | LaplacianShot | 53.5 | 62.5 | 29.6 | 74.3 | 58.5 | 75.7 | 73.4 | 92.8 | 92.0 | 52.7 | 71.7 | 67.0 |
| | PT-MAP | 57.6 | 68.1 | 31.2 | 74.9 | 63.1 | 81.1 | 79.5 | 76.2 | 60.2 | 58.4 | 73.9 | 65.8 |
| | TIM | 57.4 | 67.0 | 32.8 | 79.3 | 65.8 | 83.5 | 82.3 | 93.4 | 88.5 | 58.1 | 76.5 | 71.3 |
| | CoOp + UPL | 69.7 | 71.4 | 32.6 | 74.0 | 74.6 | 83.8 | 91.3 | 92.1 | 93.2 | 58.9 | 76.9 | 74.4 |
| | TransCLIP-FS | 70.3 | 71.9 | 34.0 | 79.4 | 74.0 | 86.4 | 91.6 | 93.6 | 94.0 | 61.1 | 79.1 | 75.9 |
| 8 | TF | 57.2 | 66.8 | 34.7 | 68.5 | 65.4 | 77.4 | 74.3 | 93.8 | 92.4 | 60.3 | 73.8 | 69.5 |
| | BD-CSPN | 57.9 | 66.5 | 34.1 | 68.3 | 64.6 | 78.0 | 77.2 | 93.2 | 92.4 | 59.0 | 74.2 | 69.6 |
| | LaplacianShot | 57.6 | 65.9 | 33.4 | 73.2 | 64.7 | 79.3 | 79.3 | 93.3 | 92.3 | 56.5 | 74.6 | 70.0 |
| | PT-MAP | 61.0 | 70.6 | 34.1 | 75.0 | 68.5 | 82.0 | 84.5 | 77.2 | 62.1 | 62.4 | 75.6 | 68.5 |
| | TIM | 62.6 | 71.3 | 35.9 | 79.8 | 74.4 | 84.3 | 87.4 | 94.0 | 90.7 | 63.6 | 80.2 | 74.9 |
| | CoOp + UPL | 70.5 | 72.8 | 38.6 | 79.1 | 78.3 | 84.5 | 90.4 | 94.4 | 93.3 | 60.6 | 79.6 | 76.6 |
| | TransCLIP-FS | 70.5 | 73.2 | 36.4 | 79.7 | 76.9 | 86.7 | 91.9 | 93.9 | 94.2 | 65.7 | 81.5 | 77.3 |
| 16 | TF | 61.8 | 70.1 | 38.3 | 74.3 | 71.2 | 80.7 | 79.5 | 95.4 | 93.6 | 62.9 | 76.0 | 73.1 |
| | BD-CSPN | 61.7 | 69.4 | 37.7 | 73.4 | 70.7 | 80.2 | 81.2 | 94.8 | 93.3 | 61.3 | 76.0 | 72.7 |
| | LaplacianShot | 60.9 | 68.3 | 36.1 | 78.1 | 69.2 | 81.2 | 81.7 | 94.8 | 93.1 | 58.6 | 76.3 | 72.6 |
| | PT-MAP | 64.0 | 72.0 | 37.4 | 75.6 | 72.0 | 82.7 | 86.1 | 78.5 | 63.7 | 63.7 | 76.3 | 70.2 |
| | TIM | 67.8 | 73.6 | 40.6 | 83.6 | 79.5 | 84.9 | 88.7 | 95.4 | 92.4 | 67.5 | 82.1 | 77.8 |
| | CoOp + UPL | 71.6 | 75.1 | 43.2 | 83.0 | 82.3 | 85.0 | 90.4 | 95.8 | 94.3 | 68.7 | 80.4 | 79.1 |
| | TransCLIP-FS | 71.8 | 74.7 | 38.6 | 83.0 | 79.8 | 86.9 | 92.4 | 94.4 | 94.0 | 65.1 | 82.1 | 78.4 |

Table 22: Detailed results of transductive methods in the few-shot setting for the 11 datasets with ViT-L/14 as visual backbone.

| Shots | Method | ImageNet | SUN397 | Aircraft | EuroSAT | StanfordCars | Food101 | Pets | Flowers102 | Caltech101 | DTD | UCF101 | Average |
|---|---|---|---|---|---|---|---|---|---|---|---|---|---|
| 1 | TF | 36.6 | 41.2 | 26.3 | 49.8 | 45.2 | 53.9 | 45.8 | 81.8 | 79.7 | 35.8 | 58.3 | 50.4 |
| | BD-CSPN | 45.3 | 50.5 | 28.9 | 53.3 | 57.5 | 67.3 | 66.7 | 93.4 | 88.4 | 39.6 | 67.2 | 59.8 |
| | LaplacianShot | 43.5 | 48.4 | 30.9 | 56.6 | 56.1 | 69.3 | 65.8 | 93.3 | 87.9 | 40.1 | 66.2 | 59.8 |
| | PT-MAP | 49.8 | 58.1 | 33.1 | 65.6 | 60.6 | 80.1 | 78.1 | 75.2 | 58.5 | 45.7 | 69.7 | 61.3 |
| | TIM | 47.7 | 56.0 | 31.1 | 62.8 | 61.1 | 79.7 | 74.2 | 95.4 | 80.1 | 41.7 | 71.5 | 63.8 |
| | CoOp + UPL | 76.0 | 72.6 | 35.8 | 72.7 | 79.2 | 89.5 | 93.2 | 86.8 | 94.9 | 60.3 | 81.1 | 76.6 |
| | TransCLIP-FS | 75.9 | 74.5 | 37.9 | 77.4 | 78.8 | 92.2 | 95.4 | 95.9 | 95.6 | 61.3 | 83.3 | 78.9 |
| 2 | TF | 50.1 | 56.6 | 33.5 | 71.7 | 58.3 | 71.6 | 65.7 | 93.0 | 90.5 | 49.8 | 69.4 | 64.6 |
| | BD-CSPN | 57.0 | 61.2 | 35.6 | 72.6 | 65.1 | 79.9 | 77.2 | 95.7 | 92.8 | 52.3 | 74.7 | 69.5 |
| | LaplacianShot | 56.5 | 61.3 | 35.9 | 76.8 | 65.4 | 80.3 | 77.4 | 96.2 | 93.3 | 52.4 | 74.8 | 70.0 |
| | PT-MAP | 61.3 | 68.0 | 37.0 | 78.4 | 68.4 | 87.3 | 86.7 | 77.9 | 61.1 | 56.5 | 75.2 | 68.9 |
| | TIM | 59.7 | 67.6 | 35.4 | 82.2 | 69.3 | 87.4 | 85.5 | 95.1 | 91.4 | 53.2 | 78.6 | 73.2 |
| | CoOp + UPL | 76.1 | 73.4 | 39.9 | 72.3 | 81.4 | 90.3 | 92.5 | 94.0 | 94.7 | 62.0 | 82.2 | 78.1 |
| | TransCLIP-FS | 76.8 | 75.1 | 40.0 | 82.1 | 79.9 | 91.8 | 95.0 | 96.6 | 95.9 | 62.6 | 83.2 | 79.9 |
| 4 | TF | 61.6 | 66.5 | 40.6 | 71.4 | 69.6 | 81.9 | 79.0 | 96.4 | 94.4 | 58.5 | 77.5 | 72.5 |
| | BD-CSPN | 64.3 | 67.8 | 40.6 | 71.4 | 72.2 | 84.7 | 82.8 | 96.7 | 95.2 | 56.9 | 79.6 | 73.8 |
| | LaplacianShot | 63.8 | 67.6 | 40.0 | 78.9 | 72.0 | 85.4 | 85.7 | 97.3 | 95.2 | 56.7 | 79.6 | 74.7 |
| | PT-MAP | 68.0 | 72.7 | 41.7 | 77.4 | 73.8 | 88.9 | 89.9 | 78.3 | 62.9 | 60.1 | 79.2 | 72.1 |
| | TIM | 68.9 | 72.7 | 42.0 | 78.4 | 77.8 | 90.0 | 92.3 | 97.4 | 91.1 | 63.5 | 83.7 | 78.0 |
| | CoOp + UPL | 76.5 | 75.1 | 44.1 | 79.3 | 83.1 | 90.1 | 92.6 | 95.2 | 95.3 | 65.8 | 83.9 | 80.1 |
| | TransCLIP-FS | 76.9 | 76.2 | 45.9 | 81.5 | 81.2 | 91.4 | 94.3 | 98.2 | 96.1 | 66.8 | 84.9 | 81.2 |
| 8 | TF | 67.4 | 72.0 | 45.6 | 76.1 | 76.5 | 86.2 | 85.1 | 97.2 | 95.1 | 65.1 | 81.5 | 77.1 |
| | BD-CSPN | 68.0 | 71.5 | 44.8 | 76.1 | 76.5 | 86.8 | 86.8 | 97.3 | 94.9 | 63.8 | 81.3 | 77.1 |
| | LaplacianShot | 67.3 | 70.4 | 43.6 | 78.2 | 75.9 | 87.3 | 88.3 | 97.0 | 94.9 | 61.2 | 80.8 | 76.8 |
| | PT-MAP | 70.7 | 74.6 | 44.1 | 78.4 | 77.1 | 89.2 | 91.3 | 79.5 | 64.5 | 65.1 | 79.7 | 74.0 |
| | TIM | 73.1 | 76.4 | 46.7 | 86.8 | 83.2 | 89.5 | 92.7 | 96.9 | 94.4 | 70.2 | 81.3 | 81.0 |
| | CoOp + UPL | 76.9 | 75.8 | 49.6 | 81.7 | 85.5 | 90.1 | 93.2 | 95.9 | 95.3 | 65.6 | 84.0 | 81.2 |
| | TransCLIP-FS | 77.2 | 77.3 | 50.0 | 82.6 | 84.1 | 91.6 | 94.5 | 98.5 | 97.0 | 70.7 | 86.0 | 82.7 |
| 16 | TF | 71.1 | 74.9 | 50.1 | 78.6 | 81.5 | 88.1 | 88.6 | 98.5 | 96.1 | 67.3 | 83.0 | 79.8 |
| | BD-CSPN | 71.1 | 74.4 | 49.4 | 78.1 | 81.2 | 88.0 | 89.8 | 98.4 | 95.8 | 66.5 | 82.5 | 79.6 |
| | LaplacianShot | 69.8 | 72.7 | 47.0 | 81.7 | 80.2 | 88.0 | 90.1 | 98.0 | 95.7 | 63.3 | 82.8 | 79.0 |
| | PT-MAP | 72.9 | 75.9 | 48.1 | 79.1 | 79.9 | 89.4 | 92.0 | 80.5 | 66.0 | 65.6 | 80.5 | 75.4 |
| | TIM | 76.4 | 78.7 | 52.5 | 89.4 | 86.5 | 91.0 | 92.0 | 98.2 | 94.5 | 73.2 | 84.8 | 83.4 |
| | CoOp + UPL | 76.9 | 77.2 | 54.1 | 85.9 | 87.8 | 90.6 | 93.2 | 97.1 | 95.6 | 72.8 | 86.2 | 83.4 |
| | TransCLIP-FS | 77.8 | 78.7 | 53.0 | 84.4 | 86.3 | 91.6 | 94.8 | 98.8 | 97.3 | 71.2 | 86.5 | 83.7 |

Table 23: UPL* top-1 accuracy on ImageNet for 8, 16 and 32 top-confidence pseudo-labels drawn from the test set.

| Architecture | $N = 8$ | $N = 16$ | $N = 32$ |
|---|---|---|---|
| ResNet-50 | 60.60 | **61.60** | 59.66 |
| ViT-B/16 | 68.92 | **69.62** | 68.87 |

Table 24: Prompt templates for each dataset.

(a) Prompt templates used in the experiments unless otherwise specified.

| Dataset | Prompt template |
|---|---|
| ImageNet | "a photo of a []." |
| SUN397 | "a photo of a []." |
| Aircraft | "a photo of a [], a type of aircraft.", |
| EuroSAT | "a centered satellite photo of [].", |
| Cars | "a photo of a [].", |
| Food101 | "a photo of [], a type of food.", |
| Pets | "a photo of [], a type of pet.", |
| Flower102 | "a photo of a [], a type of flower.", |
| Caltech101 | "a photo of a [].", |
| DTD | "[] texture.", |
| UCF101 | "a photo of a person doing [].", |

(b) Custom prompt templates for ImageNet dataset [50].

```
"itap of a []."
"a bad photo of the []."
"a origami []."
"a photo of the large []."
"a [] in a video game."
"art of the []."
"a photo of the small []."
```

