# OpenReview forum: "Boosting Vision-Language Models with Transduction"
_NeurIPS.cc/2024/Conference — NeurIPS 2024 spotlight_

### Official Review · Reviewer_bXXg · 2024-07-10

**Soundness:** 2
**Presentation:** 3
**Contribution:** 3
**Rating:** 5
**Confidence:** 4

**Summary:**

Summary:

The paper under review investigates the integration of contrastive vision-language pretraining (CLIP) with transductive learning methodologies. The authors are motivated by the efficacy of transductive learning in utilizing unlabeled data to enhance the performance of conventional supervised learning frameworks. They propose to extend this approach to the domain of vision-language pretraining. The key innovation lies in modeling the distribution of language data using a Gaussian mixture model (GMM). The authors introduce three specific learning objectives: GMM clustering, Laplacian regularization, and text knowledge preservation. These objectives collectively aim to regularize the distribution of unlabeled data while maintaining the integrity of text-guided vision-language pretraining.
The complexity of optimizing three intertwined variables necessitates an advanced optimization strategy. To address this, the authors propose the Block Majority-Minimize optimization technique, which iteratively fixes two variables while updating the third. This approach ensures the overall minimization of all three variables, providing a robust convergence guarantee. Through comprehensive quantitative experiments, the authors demonstrate that their proposed method can serve as a versatile framework, enhancing the performance of various backbone methods across different scenarios.

**Strengths:**

Strengths:

- Organization and Clarity:

The paper is meticulously organized, with a clear and coherent presentation of concepts. The structured formulation facilitates a straightforward understanding of the core ideas. The writing is lucid, allowing readers to seamlessly follow the logical progression of the methodology.

- Experimental Rigor:

The experimental evaluation is thorough and extensive. The authors employ a diverse array of datasets and baseline methods, lending credibility and robustness to their findings. The significant performance improvements observed in experiments underscore the practical utility and contribution of the proposed method.

- Theoretical Foundations:

A detailed convergence analysis is provided, offering a solid theoretical underpinning for the optimization process. This analysis enhances the credibility of the proposed optimization strategy and reassures readers of its reliability.

**Weaknesses:**

Weaknesses:

- Unclear Motivation:

The motivation for leveraging transductive learning in this context is somewhat ambiguous. The assertion that transductive learning can enhance the handling of unlabeled data, while valid in general, seems less compelling here. In the CLIP framework, vision and language data are inherently paired and labeled, diminishing the applicability of transductive learning, which traditionally targets unlabeled data.

- Mismatch with Zero-Shot Learning:

There is a conceptual misalignment between transductive learning and zero-shot learning paradigms. Transductive learning focuses on knowledge propagation from observed training data to observed test data. In contrast, zero-shot learning aims to generalize to unseen test data. This fundamental difference makes the application of transductive learning to zero-shot scenarios appear somewhat forced and unconvincing.

- Lack of Intuitive Justification:

The introduction of the GMM and Laplacian terms lacks intuitive explanation. While the quantitative results are impressive, there is an absence of qualitative analysis to elucidate why these terms specifically enhance learning performance. A deeper exploration of the underlying reasons for their effectiveness would strengthen the paper's contributions.

- Computational Overhead:

The integration of GMM and Laplacian regularization is computationally intensive. The paper would benefit from experimental validation of the method's efficiency, demonstrating that the performance gains justify the additional computational costs. Providing benchmarks or comparative analyses regarding computational efficiency would address potential concerns about scalability and practicality.

Overall Assessment:

In conclusion, this paper presents a novel approach to enhancing vision-language pretraining by incorporating transductive learning principles. Despite some motivational and conceptual ambiguities, the paper's methodological rigor, comprehensive experiments, and theoretical contributions make it a valuable addition to the field. Addressing the highlighted weaknesses through additional qualitative analyses and computational efficiency evaluations would further solidify the paper's impact and applicability.

**Questions:**

Please see the weaknesses part.

**Limitations:**

The authors have discussed the limitation in the paper.

---

> ### Author Rebuttal · Authors · 2024-08-07
>
> **W1: Unclear motivation.**
>
> While VLMs such as CLIP enable zero-shot predictions, they are far from being perfect (please refer to the zero-shot performances of CLIP for 6 different model sizes in Table 8 in the Appendix). Please note that CLIP receives unlabeled samples and makes class predictions by comparing the images to several candidate text prompts. Therefore, CLIP yields pseudo-labels that could be used as a noisy supervision in transduction without any label.
>
> What we propose in our work is to improve zero-shot CLIP and inductive few-shot methods based on CLIP as pre-training, by incorporating the transduction paradigm during inference.
>
> **W2: Mismatch with Zero-Shot Learning.**
>
> It is true that many transductive, vision-only methods (e.g., [5, 13, 33, 36, 43, 73, 24 , 72], among others) propose to propagate knowledge from observed training (labeled) data to unlabeled test data. However, in this work, we show that, in the context of VLMs, transduction could also be conducted without any label, by leveraging the pseudo-labels generated from the text embeddings, as a form of noisy supervision.
>
> Indeed, we make zero-shot predictions by employing text-based noisy supervision (the KL term in Eq. 2) and the structure of the unlabeled data in the feature space (GMM clustering and Laplacian regularization terms), which effectively corresponds to the transductive paradigm. Please note that we propose an extension of TransCLIP, named TransCLIP-FS (the objective function in Eq. 4), for cases where labeled data are available, which combines noisy supervision from the zero-shot predictions and supervision from the few-shot samples, as in the above-mentioned transductive methods.
>
> **W3: Lack of Intuitive Justification.**
>
> Thank you for giving us the opportunity to clarify. The GMM term could be viewed as a maximum-likelihood estimation objective, like in the Expectation-Maximization algorithm. It could also be viewed as a probabilistic generalization of the K-means clustering objective (lines 157-162). The principal difference with the standard K-means clustering lies in the capability of TransCLIP to learn a covariance matrix (variable $\boldsymbol{\Sigma}$).
>
> As shown in Figures 2a and 2b, which we added for this rebuttal (please refer to the attached PDF document), the learned covariance matrix enables fitting the unevenly spread feature distribution across the dimensions of the embedding space. We also show that fixing the covariance matrix (which corresponds to K-means if we omit the Laplacian and KL terms) results in a performance drop (line 2 in Table 6a of the main paper).
>
> **W4: Computational Overhead.**
>
> The Block Majorize-Minimize procedure presented and detailed in Section 3.3 provides a decoupled $\mathbf{z}$-update equation (Eq. 5), as well as closed-form updates for GMM parameters $\boldsymbol{\mu}$ and $\boldsymbol{\Sigma}$ (Eqs. 6 and 7). These decoupled updates enable the algorithm to run substantially faster than popular prompt-tuning approaches (such as UPL [25]), which require costly forward-backward propagation for multiple epochs. We report run-time comparisons in Tables 5a and 5b in the main paper.
>
> Furthermore, we add Table 1 in this rebuttal (please refer to the attached PDF document), which provides a more detailed analysis, showing: (i) TransCLIP's prediction time is a fraction of the time required for feature encoding, which
> makes its total runtime (encoding + prediction) comparable to the total runtime of the inductive zero-shot CLIP baseline; (ii) The total runtime of UPL is an order-of-magnitude slower, due to the training overhead. We will add these detailed runtimes to Tables 5a and 5b in the paper.

---

> > ### Comment · Reviewer_bXXg · 2024-08-13
> >
> > Thanks for providing further comments to address my concerns. No further concerns remained from my side, so I would like to keep my current score.

---

### Official Review · Reviewer_uSPt · 2024-07-13

**Soundness:** 2
**Presentation:** 2
**Contribution:** 2
**Rating:** 5
**Confidence:** 4

**Summary:**

The paper introduces TransCLIP, a transductive learning method designed to enhance the performance of vision-language models (VLMs) for zero-shot and few-shot learning scenarios. By incorporating a novel objective function constrained by Kullback-Leibler divergence, TransCLIP not only improves prediction accuracy but also ensures computational efficiency. This method operates as a plug-and-play module on existing VLMs, leveraging unlabeled data to boost the model's predictive capabilities significantly across various datasets.

**Strengths:**

TransCLIP integrates transductive learning into VLMs effectively, which is traditionally challenging due to the complexity of multimodal data integration.

The method consistently outperforms standard inductive and transductive approaches by leveraging text-encoder knowledge, which guides the learning process and significantly boosts performance.

Through the iterative Block Majorize-Minimize optimization procedure, TransCLIP ensures efficient computation, making it scalable for large-scale applications.

The method's ability to function atop various pre-existing models without requiring modifications to the underlying architectures enhances its applicability in real-world scenarios.

**Weaknesses:**

The proposed objective function of TransCLIP is composed of multiple terms, including a Gaussian Mixture Model-clustering term, a Laplacian regularizer, and a Kullback-Leibler divergence penalty. While this multifaceted approach aims to integrate various aspects of the data, it raises concerns about potential conflicts or trade-offs between these terms. The interaction and balance among these components are crucial, as overemphasis on one could undermine the effectiveness of others, potentially leading to suboptimal learning outcomes.

Although the paper addresses computational efficiency through an iterative Block Majorize-Minimize (BMM) optimization procedure, a detailed comparison of computation costs with other methodologies is lacking. Understanding how TransCLIP's computational demands stack up against alternative approaches, particularly in terms of time complexity and resource usage, is essential for evaluating its practical applicability and efficiency.

The scalability and performance of TransCLIP when applied to very large datasets remain uncertain. While the method is claimed to be computationally efficient, the real-world effectiveness and efficiency on datasets significantly larger than those tested (like datasets beyond the scale of ImageNet) need thorough investigation. This includes an assessment of whether the benefits observed on smaller or moderate-sized datasets consistently translate to much larger scales.

The paper provides theoretical convergence guarantees for the optimization procedure. However, the robustness of these proofs and their assumptions in practical, real-world scenarios could be questioned. Specifically, the conditions under which convergence is guaranteed should be scrutinized to ensure they are not overly restrictive or detached from practical applications. This scrutiny is vital to validate the method's theoretical foundation and to ensure its reliability across various deployment contexts.

**Questions:**

See weaknesses section.

**Limitations:**

See weaknesses section.

---

> ### Author Rebuttal · Authors · 2024-08-07
>
> **W1: Balance of terms.**
>
> We understand your concern. The terms in our objective function are not necessarily in conflict; they could be seen as a clustering (driven by the GMM term), which is regularized by the Laplacian term propagating the labels between neighboring samples, and by the KL-divergence term that incorporates the text knowledge of the VLM.
>
> Regarding the utility of the two regularization terms, we show in Table 6a (line 3) of the main paper that the Laplacian term helps improving performance. In Table 4 of the main paper, we demonstrate the importance of the KL text regularization (TransCLIP vs. TransCLIP *w/o text*).
>
> Regarding the sensitivity to hyper-parameters, we emphasize that ***the text-regularization weighting factor $\lambda$ is set to the same value for all the experiments that involve TransCLIP-ZS, regardless of the dataset or the few-shot method***. We evaluate *a posteriori* the impact of varying $\lambda$ and show that there is only a minor sensitivity as long as $\lambda$ does not tend towards $0$ (please refer to Table 6b of the main paper, and to Figure 1 that we added in this rebuttal, in the attached PDF document). We also evaluate the sensitivity to the number of neighbors considered in the Laplacian term, and show again that it does not significantly impact the performances (please refer to Table 6c of the main paper).
>
> **W2: Computation costs.**
>
>  - *Time complexity*: The Block Majorize-Minimize procedure detailed in Section 3.3 presents a decoupled $\mathbf{z}$ update equation (Eq. 5) and closed-form update equations for the GMM parameters $\boldsymbol{\mu}$ and $\boldsymbol{\Sigma}$ (Eqs. 6 and 7). These enable our algorithm to achieve rapid convergence compared to traditional prompt tuning methods such as UPL [25], which require extensive forward-backward propagation over multiple epochs.
> We provide runtime comparisons in Tables 5a and 5b of the main paper, and a more detailed analysis in Table 1 of the attached document. We will incorporate these new results into Tables 5a and 5b of the main paper to clarify the speed and efficiency advantages of TransCLIP.
>
> - *Resource usage*: we discuss the hardware requirements of TransCLIP in Appendix B, noting that TransCLIP consumes 16.9 GB of memory when inferring on ImageNet, making it feasible to run on a single 24 GB GPU for large datasets.
>
> **W3: Scalability to even larger datasets.**
>
> As discussed in Section 4.2, TransCLIP is easily applicable to multi-billion parameter models and large datasets since it does not necessitate gradient computation or model parameter training (i.e., it only requires the memory needed for single-sample inference, because the whole dataset processing can be performed one sample at a time). Results in the Table below support the scalability of TransCLIP in effectively managing more than one million images in a few minutes (on ImageNet training set which contains 1,281,167 images categorized into 1,000 classes).
>
> | Model     | Zero-shot | TransCLIP-ZS | Runtime   |
> |-----------|-----------|--------------|-----------|
> | ViT-B/16  | 68.09     | 71.02        | 284 sec.  |
> | ViT-L/14  | 74.82     | 77.95        | 396 sec.  |
>
> **W4: Theoretical convergence guarantees.**
>
> As stated in Section 3.4, our optimizer can be viewed as an instance of the general Block Majorize-Minimize (BMM) paradigm, and we establish convergence of our procedure. The technical conditions that guarantee convergence stem from the mathematical properties of our objective
> and its block-wise majorizing functions used in the BMM optimizer (such as the strong convexity of the block-wise majorizing functions). Hence, these conditions do not restrict any practical application aspects.

---

### Official Review · Reviewer_AvCq · 2024-07-19

**Soundness:** 3
**Presentation:** 3
**Contribution:** 3
**Rating:** 5
**Confidence:** 4

**Summary:**

The paper proposes a transductive method to boost the performance of existing vision-language models by assuming that all unlabeled test samples are available during the training stage. Specifically, the paper models a Gaussian mixture model, where each class is represented by a Gaussian distribution. The method refines predictions for each test sample by constraining the similarity between predictions of similar points and the similarity between the final prediction and the prediction based on text features. Experiments are conducted on standard datasets.

**Strengths:**

1. The paper addresses a less explored area of transductive learning on vision-language models, filling a gap in the existing research.
2. The motivation for the method is clear, the modeling is reasonable, and an appropriate optimization method is provided.
3. The proposed method is quite general and can be applied in both zero-shot and few-shot scenarios. It is also compatible with other methods.

**Weaknesses:**

1. The paper does not discuss a significant limitation of the proposed method. Specifically, when the test dataset arrives online, one by one, the method may not be able to make predictions for individual data points immediately. Instead, it may require collecting a sufficient number of data points before training the model and making predictions. Furthermore, if a new data point arrives after the model has been trained, does the model need to be retrained to make a prediction for the new point? Is there an efficient way to handle new data points?
2. Although the authors provide an analysis of training times in Table 5, the testing speed is more critical for users. An analysis of the time complexity during testing is also needed.
3. Considering that the method requires using all unlabeled test data for training, it resembles unsupervised learning (e.g., [25]). It is recommended to discuss the differences between the proposed method and existing unsupervised learning methods in detail.

**Questions:**

See weaknesses.

**Limitations:**

Copy from Q1: The paper does not discuss a significant limitation of the proposed method. Specifically, when the test dataset arrives online, one by one, the method may not be able to make predictions for individual data points immediately. Instead, it may require collecting a sufficient number of data points before training the model and making predictions. Furthermore, if a new data point arrives after the model has been trained, does the model need to be retrained to make a prediction for the new point? Is there an efficient way to handle new data points?

---

> ### Author Rebuttal · Authors · 2024-08-06
>
> **Q1.a: Online setting.**
>
> This is indeed a general limitation of transductive-inference approaches. Still, we believe this transductive setting is useful in a breadth of application domains and real scenarios, as pointed out by Reviewers NaRY and uSPt and evidenced by an abundant and recent literature on transductive few-shot learning; actually, in the recent years, the transductive setting has been very popular in the vision-only few-shot learning literature (see, for instance, [5, 13, 33, 36, 43, 73, 24, 72], among others). Use cases where batches of test imaging data are available occur frequently in practice, e.g., in video-stream sequences, or in smart-device photos taken and stored every day. The setting is also very appealing in various application domains, such as remote sensing or histopathology where large images are divided into patches, or more generally, to analyze existing databases or large corpora of web-scraped unlabeled images (e.g., to create pseudo-labels for large unlabeled image sets taken and stored every day).
>
> As correctly mentioned by the reviewer, if data points are coming sequentially, we would need to wait to collect a batch of them before conducting TransCLIP (while still using the zero-shot ability of the VLM if on-the-fly prediction is needed). We will add this discussion to the limitation section, as well as to the conclusion, as this could be an interesting future work to broaden the application range of TransCLIP.
>
> **Q1.b: Prediction for new samples.**
>
> This is, indeed, a very interesting point and extension that we missed in our first version.
> In fact, after running TransCLIP, the GMM parameters could be stored to make a prediction on a new individual sample, resembling the inductive paradigm:
>
> $$
> \mathbf{z}_{i} = \frac{ \mathbf{p} _{i} }{ \mathbf{p} _{i}^\top \mathbf{1}_K }
> $$
>
> This can be seen as a *Maximum Likelihood* predictor for sample $i$. We denote this naive prediction rule $\textbf{\textit{MLH}}$.
>
> More interestingly, Update Eq. 5 can be directly adapted. Let's assume we don't have access to other data points (e.g., very low resource application or for privacy concerns), we remove the Laplacian term and get the following prediction rule:
>
> $$
> \mathbf{z}_{i} = \frac{\hat{\mathbf{y}} _{i} ^ {\lambda} \odot \exp ( \log ( \mathbf{p} _{i}))}{(\hat{\mathbf{y}} _{i} ^ {\lambda} \odot \exp (\log (\mathbf{p} _{i}))) ^ \top \mathbf{1}_K} = \frac{\hat{\mathbf{y}} _{i} ^ {\lambda} \odot \mathbf{p} _{i}}{(\hat{\mathbf{y}} _{i} ^ {\lambda} \odot \mathbf{p} _{i}) ^ \top \mathbf{1}_K}
> $$
>
> This can be seen as a *Maximum A Posteriori* predictor for sample $i$ since it is the class probability likelihood $\mathbf{p}_i$ weighted by the initial pseudo-label $\hat{\mathbf{y}} _{i}^{\lambda}$ (prior knowledge). We denote this prediction rule $\textbf{\textit{MAP}}$.
>
> Following this very interesting comment by the reviewer, we conduct new experiments (average over 100 random seeds): We split the test set, keeping randomly 10% of the data points as a held-out set, and infer TransCLIP on the remaining 90% of the samples. We then apply the prediction rule on each sample of the held-out set (independent predictions, as in the inductive setting). The Table below summarizes the performance and shows that our $\textbf{\textit{MAP}}$ predictor rule for new unseen data points still brings a significant gain, validating the feasibility of this approach.
>
> | Model                         | ImageNet | SUN397 | Aircraft | EuroSAT | Average |
> |-------------------------------|----------|--------|----------|---------|---------|
> | CLIP (*on held-out*)       | 66.6     | 62.5   | 24.7     | 48.3    | 65.3    |
> | $\textbf{\textit{MLH}}$ (*on held-out*)              | 67.5     | 67.2   | 24.6     | 64.1    | 68.0    |
> | $\textbf{\textit{MAP}}$ (*on held-out*)             | **69.9**     | **68.6**   | **26.6**     | **64.7**    |**70.1**    |
> |                               |          |        |          |         |         |
> | TransCLIP-ZS (*transductive*)   | 70.3     | 68.9   | 26.9     | 65.1    | 70.3
>
> **Q2: Precision on runtime.**
>
> We provide more details on the runtime of each step in the attached document (see Table 1). On ImageNet, UPL$^*$ takes 151 minutes for prompt tuning, followed by 59 seconds for images+texts encoding during the inference step, resulting in a total of 152 minutes. In contrast, TransCLIP has no training stage and requires 14 seconds to run, which is only a fraction of the 59 seconds needed for images+texts encoding. We will extend Table 5 of the main paper to include these clarifications.
>
> **Q3: Comparison to unsupervised learning.**
>
> We refer to Table 8 in the Appendix, which contains a direct comparison of TransCLIP with UPL [25]. Please note that UPL was initially designed to find pseudo-labels in a unlabeled training set, followed by a training stage before inferring on the test set, making UPL an inductive method. Therefore, we extended UPL to UPL$^*$, drawing pseudo-labels directly from the test set, hence making it a transductive method (please refer to the details in lines 569-572). In conjunction with our response to W6 by Reviewer 1 (NaRY), we will add a description of UPL and UPL$^*$ in the Appendix for better clarity.
>
> We also compare TransCLIP to other unsupervised methods (in Table 8 of the Appendix) such as TPT [42] and MTA [65] (two test-time augmentation techniques) and SwapPrompt [40]. We draw attention to the differences between their settings and ours (lines 567-569), which will be extended by discussion "W1 and Q2" with Reviewer aJGT.

---

> > ### Comment · Reviewer_AvCq · 2024-08-12
> >
> > Thanks for your response. Most of my concerns have been addressed. Considering the limitations in the online setting, I slightly increased my score to 5.

---

### Official Review · Reviewer_aJGT · 2024-07-22

**Soundness:** 3
**Presentation:** 3
**Contribution:** 3
**Rating:** 6
**Confidence:** 4

**Summary:**

The paper proposes a method named TransCLIP that performs transductive inference to boost classification performance of Zero-Shot &  Pre-trained Few-Shot CLIP models. The proposed methodology is an extension of [1], but for VLMs. TransCLIP proposes to learn the class prototypes, in contrast to fixed-prototypes considered in [1] and adds a language guidance term to penalize predictions that stray from the CLIP pseudo-label. Following [1], the paper uses a Majorize-Minimize framework to solve for the prototypes and class predictions in closed form. Experiments are shown on 11 standard datasets, showing that TransCLIP improves on top of existing few-shot fine-tuning methods.

[1] Imtiaz Ziko, Jose Dolz, Eric Granger, and Ismail Ben Ayed. Laplacian regularized few-shot
learning. In International conference on machine learning, pages 11660–11670. PMLR, 2020.
[2] Ma, Xiaosong et al. “SwapPrompt: Test-Time Prompt Adaptation for Vision-Language Models.” Neural Information Processing Systems (2023).
[3] Shu Manli et al. “Test-time prompt tuning for zero-shot generalization in vision-language models.” Neural Information Processing Systems (2022).

**Strengths:**

\+ The paper proposes to use transduction to improve CLIP’s downstream performance. This is a promising direction which has seen recent efforts [2,3]. The considered formulation is a scalable alternative to existing test-time methods. The solution is well motivated, and the paper is easy to follow.
\+ TransCLIP builds on top of, and modifies [1] for VLMs. TransCLIP makes two major design choices  i) The class prototypes are learnable ii) A language-guidance term is added to regularize updates. Both design choices lead to improved empirical performance on CLIP.
\+ The experiments are comprehensive . The proposed language guidance term and other design choices used in TransCLIP lead to improved accuracy over zero-shot CLIP and few-shot finetuned CLIP.

**Weaknesses:**

\- Discussion on Test-Time methods. Recently many test time methods (a type of transductive learning) have been proposed to improve VLM performance. A discussion of the tradeoffs between the proposed approach and existing test-time adaptation (TTA) methods needs to be discussed. For instance, TransCLIP requires the entire test batch, while TTA methods require fewer test samples [2,3].
\- Issues with missed references. Important references are missing in Section 3.3. The entire section follows similar arguments from [1], but re-derived by adding the language-guidance term.
\- The GMM argument is unclear to me. However, the GMM clustering term in eq 2 is minimized when label assignments are given by the closest prototype. The loss is reminiscent of $\mathcal{N}$ in [1], but the paper proposes to use a Mahalanobis distance instead of euclidean distance. I am unsure of the insight provided by posing it as GMM clustering instead of drawing parallels with [1].

**Questions:**

1. As discussed above, the GMM angle is unclear, but maybe it boils down to learning class prototypes in the absence of few-shot data.
2. Existing test-time methods show strong empirical evidence of the power of transduction for VLMs. A discussion of the scope of these methods is necessary.
3. Since the primary novelty of the paper is to suggest that language-guidance can greatly improve transduction, the appropriate ablation is necessary. Setting $\lambda=0$ in the proposed KLDiv parametrization only removes the cross-entropy term, and is not the desired ablation.

**Limitations:**

The authors have addressed some limitations of the work. There is no potential negative societal impact from this work

---

> ### Author Rebuttal · Authors · 2024-08-06
>
> **W1 and Q2: Discussion on Test-Time methods**.
>
> We agree that the mentioned test-time methods also employ the transduction paradigm. However, their settings are in fact very different from the ones studied in our work (zero- and few-shot adaptation of CLIP, improving inductive few-shot methods). We still report their performance in Table 8 of the Appendix but we agree that a more complete discussion is needed. We will add the following discussions:
>
> - *Design*: SwapPrompt [40] has been designed to make batch predictions on-the-fly, and has continual-learning related mechanisms such as an exponential moving average prompt across batches. The setting of TPT [42] also differs from ours; it has been designed to work on a single sample with many data augmentations to train one prompt per image.
>
> - *Resource*: Both methods require access to model weights for training (i.e., do not operate in a black-box setting). We also note that prompt tuning does not scale well with the model size and is even impractical on very large models such as EVA-CLIP-8B.
>
> - *Computation overhead*: TPT takes nearly 5 hours for ImageNet. As SwapPrompt relies on prompt learning for several epochs and requires data augmentation, its running time is above the one of UPL$^*$ (please see Table 1 in the attached document). In comparison, TransCLIP takes only 73 seconds for the whole pipeline (images and texts encoding + transduction; please refer to Table 1 in the attached document).
>
> - *Batch size*: It is true that TransCLIP first requires a test batch to work on. However, we would like to highlight our answer Q1.b for Reviewer AvCq, which discusses and assesses how online predictions for new (single) samples can be processed after running TransCLIP.
>
> **W2: Issues with missed references. All Section 3.3 follows similar arguments from [1], but re-derived by adding the language term.**
>
> Please note that we mention [1] (LaplacianShot, reference [73] in the paper) in related work (lines 85-87), and provide direct experimental comparisons to it in Table 4 in the paper, as well as in Tables 17 to 21 in the Appendix. We agree with the reviewer that we could emphasize that the derivation of [73] (i.e., linearization of the Laplacian term) is a sub-step of our approach but without the text term (this could be done in sub-section Majorize-Minimize with respect to the z-block in section 3.3).
>
> We clarify, however, that [73] does not have a Block Majorize-Minimize (BMM) optimization procedure (i.e., the additional $\boldsymbol{\mu}$ (prototypes) and $\boldsymbol{\Sigma}$ (covariance) steps in Eqs. (6) and (7) in our section 3.3). The objective function in [73] is quadratic (the term using the fixed prototypes being linear) and has only assignment variables, whereas our objective function is higher-order with three blocks of intertwined variables (prototypes, covariance and assignments). Interestingly, Table 4 shows that, even without the text term, our objective function brings significant improvements over the LaplacianShot one when the number of shots increases (due to learning prototypes $\boldsymbol{\mu}$ and covariance $\boldsymbol{\Sigma}$ from both unlabeled and labeled data).
>
> We would also like note that the general principle of linearizing concave quadratic terms (which provides a majorizing function on the Laplacian term in our case) is quite common in the Majorize-Minimize optimization literature, and has been used in other works in machine learning, even much earlier than LaplacianShot, e.g., the following work in the context of conditional random fields (CRFs): Krahenbuhl and Koltun, Parameter Learning and Convergent Inference for Dense Random Fields, ICML 2013.
> We will add this reference and connect to the LaplacianShot procedure ([73]) in Section 3.3 (e.g., at line 193).
>
>
> **W3 and Q1: GMM argument.**
>
> They are two major differences with the linear term $\mathcal{N}(Y)$ in LaplacianShot [73], in which the class prototypes are fixed (computed from the labeled shots only) and the assignment vector is the only block of variables. First, as mentioned by the reviewer, our formulation enables to learn class prototypes $\boldsymbol{\mu}$ from both the unlabeled and labeled data (both in zero- and few-shot).
> This claim is supported by the results in lines 1 and 4 of Table 6a in the main paper, which show that performance tends to decrease when the class prototypes are fixed.
>
> Second, by introducing a learnable covariance matrix $\boldsymbol{\Sigma}$, we enable TransCLIP to better model the feature variance, which may be unevenly spread across the dimensions of the embedding space. To further support this claim, we add Figures 2a and 2b in the attached PDF document, which show that feature variance varies across dimensions, and that our learnable $\boldsymbol{\Sigma}$ tends to fit this particular distribution. You are correct that learning a covariance matrix is equivalent to learning the weights of the Mahalanobis distance between the data points and their class prototypes, and we will add this discussion to Sec. 3.1, in the GMM-based clustering part. Thank you for pointing that out!
> Interestingly, the learned GMM parameters can then be used to make predictions on new incoming samples, as discussed in Q1.b for Reviewer AvCq.
>
> **Q3: KL Div parametrization.**
>
> We would like to point to Table 6b in the main paper, which studies the impact of $\lambda$ for the zero-shot setting.
>
> However, this Table studies only the zero-shot case. Therefore, we provide here a more thorough study (please refer to Figure 1 in the attached document) for a wider range of settings (zero-, 1-, 4-, and 16-shot), which clearly shows both the importance of this term and its stability around the selected value. Indeed, the average zero-shot performance does not vary significantly between $\lambda=0.5$ and $\lambda=1.6$, and $\lambda=0.5$ is an acceptable value for all few-shot settings. We will add this extension to our current ablation studies.

---

> > ### Comment · Reviewer_aJGT · 2024-08-13
> >
> > Thank you for the clarifications. Since my concerns have been addressed, I have increased my score to 6. I recommend incorporating the discussions on test-time methods into the main paper.

---

### Official Review · Reviewer_NaRY · 2024-07-26

**Soundness:** 3
**Presentation:** 3
**Contribution:** 3
**Rating:** 7
**Confidence:** 2

**Summary:**

In this paper, the authors explore fine-tuning VLMs to specific unlabelled/partially-labelled datasets in a transduction setting. The authors propose an objective function to carry out joint inference of labels for all the test samples simultaneously. The authors then propose an iterative block Majorie minimize procedure to arrive at a local optimum. Empirical results on multiple datasets show clear superiority of the proposed method over other prompt-based fine-tuning and inductive zero-shot and few-shot methods.

**Strengths:**

The paper is generally well-written, easy-to-follow and intuitive. Experiments clearly show big improvement over current state-of-the-art. I believe this method could be very useful for downstream applications of VLMs in different domains.

**Weaknesses:**

Some aspects of the writing should be made more clear.

1. Is Eq 2 did you mean zi^Tz_j since both are vectors?

2. How is the final prediction done, for the ith sample? Is it the argmax of z_i?

3. Design choice: The experiments show great results, but is there apriori any expectation for using a GMM model for the classes? Perhaps this is related to the linear manifold hypothesis for these VLMs and LLMs. Some more discussion on this will be good. As it is written it comes as a random choice which seems to work well with no intuition behind it.

4. Line 167. why is the max function needed for W to be PSD? Even without the max operator, W will be a gram matrix and thus PSD. Am I missing something here?

5. Why does the laplacian regularization enforce visually similar points to have the same assignment? I understand this when z is one-hot, but here z is any vector in the simplex. Is this still true when z belongs to the simplex?

6. One of the baselines used in the paper is a modification of UPL, but this modification is not clearly mentioned in the paper or the appendix. How is the modification done in the transduction setting? Also a small description explaining UPL would be good in the appendix for added context to the readers.

**Questions:**

See weaknesses above.

**Limitations:**

Authors have adequately discussed limitations in the paper.

---

> ### Author Rebuttal · Authors · 2024-08-06
>
> **W1: Eq 2.**
>
>  It's an omission on our side; indeed, it should be $\mathbf{z}_i^\top \mathbf{z}_j$. Thanks for pointing that out! We will update the objective function accordingly.
>
>
> **W2: Final prediction.**
>
> Yes, we take the argmax for the final prediction. We should have made it clearer at the end of Section 3.3 as follows: `'Note that, after convergence, we use the sample-to-class assignment variables $\mathbf{z}_i$ as predictions for each sample $i$ of the query set $\cal Q$, *using the argmax operation for conventional classification*.". We will also update Algorithm 1 in the Appendix to further clarify this (line 13: return argmax($\mathbf{z}$)).
>
>
> **W3: Design choice (using a GMM).**
>
> We agree that further discussions / illustrations on this choice would be helpful to readers. The most basic unsupervised clustering objective would be K-means. This could be viewed as an extension of inductive methods (e.g., CoOp, which also search for new class centroids via prompt tuning) to unlabeled test data. However, K-means makes the assumption that the clusters are 'spherical', which may not be the case for the embedding vectors ensuing from VLMs. Indeed, K-means corresponds to a particular (simplified) case of our formulation, in which the covariance matrix is fixed to the identity matrix, as pointed out in lines 160-161.
>
> The ablation studies in Table 6a show TransCLIP performance when $\boldsymbol{\Sigma}$ is not updated (note that the other regularization terms are present), resulting in a large performance drop. This suggests that the spherical-cluster assumption (i.e., identity covariance matrix), as in K-means, may not be valid for VLM embeddings. Table 6d also shows the impact of having an isotropic $\boldsymbol{\Sigma}$, which could be seen as an intermediate solution between K-means (with fixed $\boldsymbol{\Sigma}$) and the TransCLIP objective function (with learnable $\boldsymbol{\Sigma}$), demonstrating the importance of learning $\boldsymbol{\Sigma}$.
>
> To provide further insights, we added Figures 2a and 2b to this rebuttal (please refer to the attached PDF document), which show that feature variance varies across the dimensions and that our learnable $\boldsymbol{\Sigma}$ tends to fit such 'elliptical' cluster distributions. Additionally, the introduction of a learnable $\boldsymbol{\Sigma}$ can also be seen as learning the weighting factors of the Mahalanobis distance between the class centroids and the data points (as pointed out by Reviewer aJGT).
>
>
> **W4: Affinity matrix.**
>
> You are right; our affinity matrix W can be rewritten as $F^\top F$ and is indeed a Gram matrix, hence PSD. Therefore, the max operator is not necessary to ensure the PSD condition. We will remove this statement. Thanks!
>
>
> **W5: Effect of Laplacian regularization beyond the simplex vertices.**
>
> Yes, the effect of Laplacian regularization holds within the simplex, beyond the vertices. Indeed, for a given, non-vertex $\mathbf{z}_i$ lying within the simplex, the vector $\mathbf{z}_j$ that maximizes $\mathbf{z}_i^\top \mathbf{z}_j$ is the one-hot (vertex) vector where the entry corresponding to the argmax of $\mathbf{z}_i$ is 1 (e.g, for $\mathbf{z}_i = [0.7, 0.2, 0.1]$, $\mathbf{z}_j = [1, 0, 0]$ maximizes the dot product). This function is convex in $\mathbf{z}_j$ over the simplex, and any deviation of $\mathbf{z}_j$ from the maximizing vertex can only cause a decrease in the dot product.
> Of course, this is an intuitive example, but the general behavior of the Laplacian term in our objective function is more complex. Multiple points are linked to each other, with potentially different predictions (and different most confident classes), combined with the KL term penalizing divergence from the zero-shot prediction (potentially deviating from the one-hot solutions).
>
>
> **W6: Modification of UPL.**
>
> As suggested, we will add a small description of UPL and UPL$^*$ in the Appendix (in C.1 Zero-shot, line 570) for better clarity.
>
> UPL: Unsupervised Prompt Learning [25] where N = 16 hard pseudo-labels per class are generated from a training set. Then a cross-entropy loss function on soft tokens is minimized (see UPL$^*$ below).
>
> UPL$^*$: We implement a natural extension of UPL. N = 16 hard pseudo-labels per class are generated from the query (test) set ${\cal P}\subseteq {\cal Q}$ according to the prediction’s confidence. For fairness, we reevaluated the number of pseudo-labels to select and still found that 16 per class yields the best results on average, as seen in Table 22. The following cross-entropy loss function is minimized:
> $$
> \mathcal{L}(\overline{\text{V}}| \\{ \mathbf{x}_i \\} _{i=1} ^{|{\cal P}|})) = \frac{1}{|{\cal P}|} {\sum} _{j \in {\cal P}} \mathcal{L} _{\tiny{UPL}} (\overline{\text{V}}|\mathbf{x}_j)
> $$
> where $\overline{\text{V}}$ denotes the vector of learnable context token embeddings. At inference, the learned tokens are used to generate the text embedding of each class.

---

### Author Rebuttal · Authors · 2024-08-07

We greatly appreciate the reviewers' insightful and constructive comments, and are pleased that four out of the five reviewers voted towards acceptance.

We are also glad that the reviewers found the method useful for various domains and real-world scenarios (Reviewers NaRY and uSPt), the experiments comprehensive and convincing (Reviewers aJGT, bXXg and NaRY), and the work to fill a gap in the literature (Reviewer AvCq). The reviewers also highlighted the interest of our iterative block majorize-minimize (BMM) procedure, which tackles three blocks of intertwined variables (prototypes, covariance and assignments) while guaranteeing convergence.

Below, we address all the questions raised by the reviewers. In particular, we provide additional ablation studies and clarifications, as requested. For the reviewers' convenience, we have included additional illustrations in the attached document.

---

### Decision · Program_Chairs · 2024-09-25

**Decision:**

Accept (spotlight)

**Comment:**

The paper targets a new setting based on transductive learning for CLIP-based models. The paper received five reviews in total and all reviewers voted acceptance (borderline accept or above). The paper is strong in several notable aspects: it is well-written, well-structured, and easy-to-follow; it fills a gap in the literature as this topic is under-studied; the method achieved big improvement over prior SOTA and could be very useful for downstream applications of CLIP; the experiments are comprehensive. In terms of weaknesses, the reviewers pointed out that some concepts are ambiguous and more discussions need to be added for test-time methods. The rebuttal has well addressed most concerns raised by the reviewers. Given the novelty and strong results of the paper, the AC recommends acceptance.